

# Cranial osteology of the pampathere *Holmesina floridanus* (Xenarthra: Cingulata; Blancan NALMA), including a description of an isolated petrosal bone

Timothy J. Gaudin[1] and Lauren M. Lyon[2]

[1] Department of Biology, Geology and Environmental Science, University of Tennessee—Chattanooga, Chattanooga, TN, USA
[2] Department of Geosciences and Don Sundquist Center of Excellence in Paleontology, East Tennessee State University, Johnson City, TN, USA

## ABSTRACT

The present study entails descriptions of several well-preserved skulls from the pampathere species *Holmesina floridanus*, recovered from Pliocene localities in central Florida and housed in the collections of the Florida Museum of Natural History. Bone by bone descriptions have allowed detailed reconstructions of cranial morphology. Cranial foramina are described and illustrated in detail, and their contents inferred. The first ever description of an isolated pampathere petrosal is also included. Cranial osteology of *Holmesina floridanus* is compared to that of Pleistocene species of *Holmesina* from both North and South America (*Holmesina septentrionalis*, *Holmesina occidentalis*), as well as to the other well-known pampathere genera, to closely related taxa among glyptodonts (*Propalaehoplophorus*), and to extinct and extant armadillos (*Proeutatus*, *Euphractus*). This study identifies a suite of apomorphic cranial features that serve to diagnose a putative, progressive series of more inclusive monophyletic groups, including the species *Holmesina floridanus*, the genus *Holmesina*, pampatheres, pampatheres plus glyptodonts, and a clade formed by pampatheres, glyptodonts, and *Proeutatus*. The study highlights the need for further anatomical investigations of pampathere cranial anatomy, especially those using modern scanning technology, and for analyses of pampathere phylogenetic relationships.

## INTRODUCTION

Living armadillos, the only mammals to bear a carapace of dermal bony armor, are the most diverse of the extant groups of Xenarthra, numbering at least 21 of the 31 currently recognized xenarthran species (*Aguiar & Da Fonseca, 2008*—although armadillo taxonomy is currently in flux; e.g., see *Abba et al., 2015*; *Feijó & Cordeiro-Estrela, 2016*; *Billet et al., 2017*; *Hautier et al., 2017*). However, the diversity of extinct armored xenarthrans, i.e., the Cingulata, far surpasses its extant representatives, not only in taxonomic diversity, but in terms of body size, locomotory diversity, and dietary diversity, even including a "horned" taxon *Peltephilus* (*Fernicola, Vizcaíno & Fariña, 2008*;

Corresponding author
Timothy J. Gaudin,
Timothy-Gaudin@utc.edu

*Gaudin & Croft, 2015*; *Croft, 2016*). Regarding diet, it is particularly noteworthy that there are omnivorous extant armadillos, but no herbivores (*McDonough & Loughry, 2008*; *Gaudin & Croft, 2015*), whereas the fossil cingulates include two herbivorous clades, pampatheres and glyptodonts. Both are comprised of large bodied taxa with complex dentitions (lobate teeth composed of multiple dental tissues of differing hardness; *Kalthoff, 2011*). The former numbers only a few genera, whereas the latter encompasses at least 65 genera (*McKenna & Bell, 1997*). Both are understudied, particularly given their conspicuous nature, often bizarre anatomies, and their abundance and ecological importance in late Cenozoic faunas of South and North America.

Pampatheres are a particularly poorly studied group. The oldest undoubted pampathere does not appear until the middle Miocene (*Gaudin & Croft, 2015*; with the possible exception of a very poorly preserved taxon from the late Eocene of Patagonia, *Machlydotherium*, *De Iuliis & Edmund 2002*). The group's basic taxonomy has long been unsettled. *McKenna & Bell (1997)* recognize only four valid pampathere genera, though several new taxa have since been added (*Edmund & Theodor, 1997*; *Góis et al., 2015*). One of their genera, *Pampatherium*, includes as a junior synonym at least one genus that is widely recognized as a separate, valid taxon, *Holmesina*; however, which of the species described in the literature belong in *Holmesina* and which in *Pampatherium* has been uncertain (*Edmund, 1987*; *De Iuliis, Bargo & Vizcaíno, 2000*). In addition, *McKenna & Bell (1997)* recognize the taxon name *Plaina* as a junior synonym of the genus *Kraglievichia*, whereas subsequently, *De Iuliis & Edmund (2002)* synonymize *Plaina* with McKenna and Bell's genus *Vassallia*. Part of the taxonomic difficulties lie with the paucity of fossil material. The majority of preserved pampathere remains consist of isolated osteoderms. *De Iuliis & Edmund (2002*, p. 50*)* note that "taxa based on small samples of osteoderms [are] unreliable," and yet osteoderms have been used extensively in the alpha taxonomy of pampatheres and other cingulates (*Castellanos, 1946*; *Edmund, 1985a*; *Scillato-Yané et al., 2005*; *Góis et al., 2013*).

The nature of the pampathere record has also hindered an understanding of their basic skeletal anatomy. Most of the described postcranial skeletal remains are based on very incomplete material, have received only cursory descriptions, and are poorly illustrated, if at all, by unlabeled photographs showing only one or two views (*Castellanos, 1937*; *James, 1957*; *Robertson, 1976*; *Cartelle & Bohórquez, 1985*; *Edmund, 1985b*; *Edmund & Theodor, 1997*; *Góis et al., 2015*). Despite the fact that several complete skeletal reconstructions have been published (*James, 1957*; *Edmund, 1985b*), the postcranial osteology of pampatheres remains scarcely known.

For the skull, mandible, and dentition, the situation is somewhat better. A fair number of complete, or nearly complete skulls and mandibles are known from a variety of taxa, including *Kraglievichia* (*Castellanos, 1937*), *Vassallia* (*De Iuliis & Edmund, 2002*), *Pampatherium* (*Bordas, 1939*; *De Iuliis, Bargo & Vizcaíno, 2000*), and various species of *Holmesina* (*Simpson, 1930*; *James, 1957*; *Cartelle & Bohórquez, 1985*; *Edmund, 1985b*; *Vizcaíno, De Iuliis & Bargo, 1998*; *Góis et al., 2012*), though several other genera remain incompletely known (e.g., *Scirrotherium*, Edmund and Theodor 1997; *Tonnicinctus*, *Góis et al., 2015*). More detailed examinations of cranial anatomy have been published,

including several studies of the ear region (in *Pampatherium*, *Bordas 1939*; *Guth 1961*; and in *Vassallia*, *Patterson, Segall & Turnbull, 1989*) and a recently published study on brain anatomy based on a digital endocast (*Tambusso & Fariña, 2015*). However, many of these cranial descriptions are fairly cursory, and virtually all are illustrated with unlabeled photographs that leave out many details. Even the ear region studies fail to address or adequately illustrate the detailed anatomy of the petrosal bone, as is common among more modern treatments of mammalian auditory region osteology. To date, there remains no study of the cranial osteology of pampatheres that clearly illustrates suture patterns and provides a bone by bone description of the anatomy, including the cranial foramina and their likely contents.

Fossil pampatheres have been known from the state of Florida, in the extreme southeast of North America, for more than a century (*Simpson, 1930*). Two species in the genus *Holmesina* are currently recognized: a late Pliocene-early Pleistocene (Blancan NALMA) form, *Holmesina floridanus*; and, a middle to late Pleistocene taxon (Irvingtonian and Rancholabrean NALMA), *Holmesina septentrionalis* (*Hulbert & Webb, 2001*). Both are known from extensive material, but the older material is particularly complete, abundant, and well-preserved (see, e.g., the skull illustrated in *Hulbert & Webb (2001*, fig. 10.7*)*, currently on exhibit at the Florida Museum of Natural History), though it remains mostly undescribed. Multiple individuals, including both adults and subadults, are derived largely from two sites: Haile 7G in Alachua County, Florida; and Inglis 1C in Citrus County, Florida. The goal of the present study is to describe the cranial osteology of *Holmesina floridanus*, based on this material. Because of the preservation quality, these fossils will allow us to conduct a thorough, bone by bone analysis of the skull, and to provide a fairly comprehensive view of the cranial foramina and their reconstructed contents. There is even an isolated petrosal among this material, which will allow us to describe the bony anatomy of the auditory region in unprecedented detail. These descriptions are accompanied by a carefully executed series of drawings, including both drawings of the best preserved fossils themselves, as well as reconstructions of the anatomy as we believe it would have appeared in life. The present study will provide the most detailed glimpse yet into the cranial anatomy of pampatheres, and should serve as an important basis for future studies of the paleontology, systematics, and evolution of this enigmatic group of cingulate xenarthrans.

## MATERIALS AND METHODS

Our goal was to base our description on the best preserved specimens of *Holmesina floridanus* available. Unfortunately for our purposes, the best preserved skull, UF 121742 (one of the best preserved fossil skulls we have ever seen!), is currently on exhibit at the Florida Museum of Natural History. Although the museum staff was kind enough to allow our examination of this specimen for an afternoon, it was not possible for us to borrow the skull for more careful study. Therefore, the descriptions below are based largely on three other specimens, UF 191448, UF 224450, and UF 248500, which were also in excellent condition and were available for loan (Fig. 1). UF 191448 is an almost perfectly complete adult skull, with only minor damage in the orbital wall and nasopharyngeal

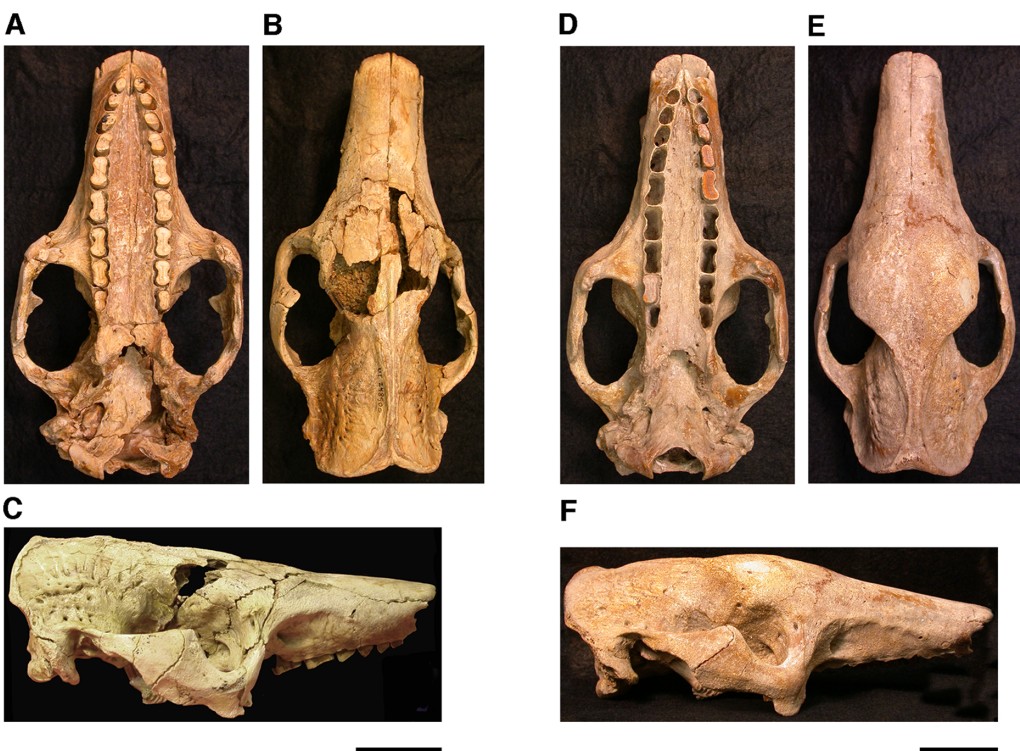

**Figure 1 Photographs of skulls of *Holmesina floridanus*.** Skull of UF 248500: (A) ventral view; (B) dorsal view; (C) right lateral view. Skull of UF 191448: (D) ventral view; (E) dorsal view, (F) right lateral view. Scale bars = 5 cm.                     

roof; but, as an adult, most of the sutures are closed, and the specimen retains only 4 of 18 teeth (left M3–5, and right M8). UF 224450 is an isolated but nearly perfectly preserved left mandible, however it retains only three of nine teeth (M2 and M6–7). UF 248500 is a subadult specimen with some significant damage to the middle portions of the skull, including parts of the skull roof, orbital wall, nasopharynx and left basicranium; but, it retains many if not most of its sutures, all its dentition is intact, and those portions of the skull that are present are very well preserved. In addition, it has a complete, isolated left petrosal that we were able to examine in three dimensions.

In order to examine interspecific variation, including ontogenetic variation, the three specimens that form the primary basis for this description were compared to the other skulls and mandibles of *Holmesina floridanus* in the collections of the Florida Museum of Natural History. Most of these, with the exception of the aforementioned display specimen, are not fully prepared, several are incompletely preserved, and at least one represents a subadult likely even younger than UF 248500; age was estimated based on the level of sutural fusion present in the skull, and the surface texture of the skull bones. These specimens include the following: UF 121742 [exhibit skull]; UF 223813 [skull only]; 248000 [partial mandible]; 275496 [juvenile skull]; 275497 [skull and mandible]; 275498 [skull and mandible]; 278000 [partial skull and mandible]; 285000 [skull and mandible]; 293000 [skull and mandible]. None of the *Holmesina floridanus* material examined preserved any trace of the ectotympanic bone or the auditory ossicles, or showed any trace

of an entotympanic (an element commonly present in other xenarthrans and likely a synapomorphy of Xenarthra; *Patterson et al., 1992*; *Gaudin, 2004*; *Gaudin & McDonald, 2008*), though, as noted above, some specimens are not yet fully prepared.

In order to assess generic level variation within *Holmesina*, the *Holmesina floridanus* material described above was compared to two specimens of the North American Pleistocene species *Holmesina septentrionalis* (UF 889 [partial skull only] and UF 234224 [cast skull only]) and one specimen of the South American Pleistocene species *Holmesina occidentalis* (ROM 3881 [skull], ROM 4955 [mandible]), as well as any literature available on these taxa. Likewise, in order to gain a comparative perspective on pampathere cranial anatomy, our material was compared to one specimen of *Vassallia maxima* (FMNH P14424), as well as the available literature on this and other pampathere skulls. Finally, in order to place this anatomy in a broader context among cingulates, *Holmesina floridanus* was compared to specimens of the basal glyptodont *Propalaeohoplophorus* (YPM VPPU 15007, 15291; FMNH 13205; glyptodonts are the putative sister taxon to pampatheres; *Gaudin & Wible, 2006*; *Porpino, Fernicola & Bergqvist, 2009*; *Billet et al., 2011*), the extinct eutatine armadillo *Proeutatus* (FMNH P13197, P13199; *Proeutatus* is the putative sister taxon to pampatheres and glyptodonts; *Gaudin & Wible, 2006*; *Billet et al., 2011*), and an extant euphractine, the six-banded armadillo *Euphractus sexcinctus* [CM 6339; UTCM 1481, 1486, 1491, 1500; one of the living armadillos that is most closely related to pampatheres in the most comprehensive morphology-based cingulate phylogenies, those of *Gaudin & Wible (2006)* and *Billet et al. (2011)*, but see *Porpino, Fernicola & Bergqvist (2009)* for contrasting view]. In certain specific instances, other comparative taxa have been utilized (e.g., the pampathere *Scirrotherium*, the extinct armadillos *Peltephilus* and *Kuntinaru*, and sloths). In instances in which a specific specimen number has been noted as part of a comparison, the information derives from personal observations made by the authors of the present study. If a literature citation is provided in addition to or in place of a specimen number, the observation derives in part or in whole from the observations of other authors.

Descriptions of the dorsal surface of the petrosal are only available for a small number of cingulate taxa. Therefore, we will be comparing the anatomy of the dorsal surface of our isolated petrosal in *Holmesina floridanus* to the detailed description of *Dasypus novemcinctus* by *Wible (2010)*, to a bisected skull of *E. sexcinctus* (UTCM 1486), and to a specimen of *Vassallia maxima*, FMNH P14424, in which the braincase has been bisected (though its endocranial anatomy was never described; the cut is visible in *De Iuliis & Edmund (2002*, fig. 2*)*). Because these are the only three cingulates for which we have information on the lateral surfaces of isolated petrosals, we shall restrict our comparisons of this surface to these three taxa, *Holmesina floridanus*, *Dasypus novemcinctus* (*Wible, 2010*), and an Eocene dasypodine lacking a specific taxonomic assignment (*Babot, García-López & Gaudin, 2012*).

The mandible is preserved in a number of UF *Holmesina floridanus* specimens, including UF 223813, 248500, 275497, 275498, 285000 and 293000. In all but the first two it remains incompletely prepared and attached to the skull, so that the occlusal surfaces of the teeth are not completely visible and the medial mandibular surfaces are also largely

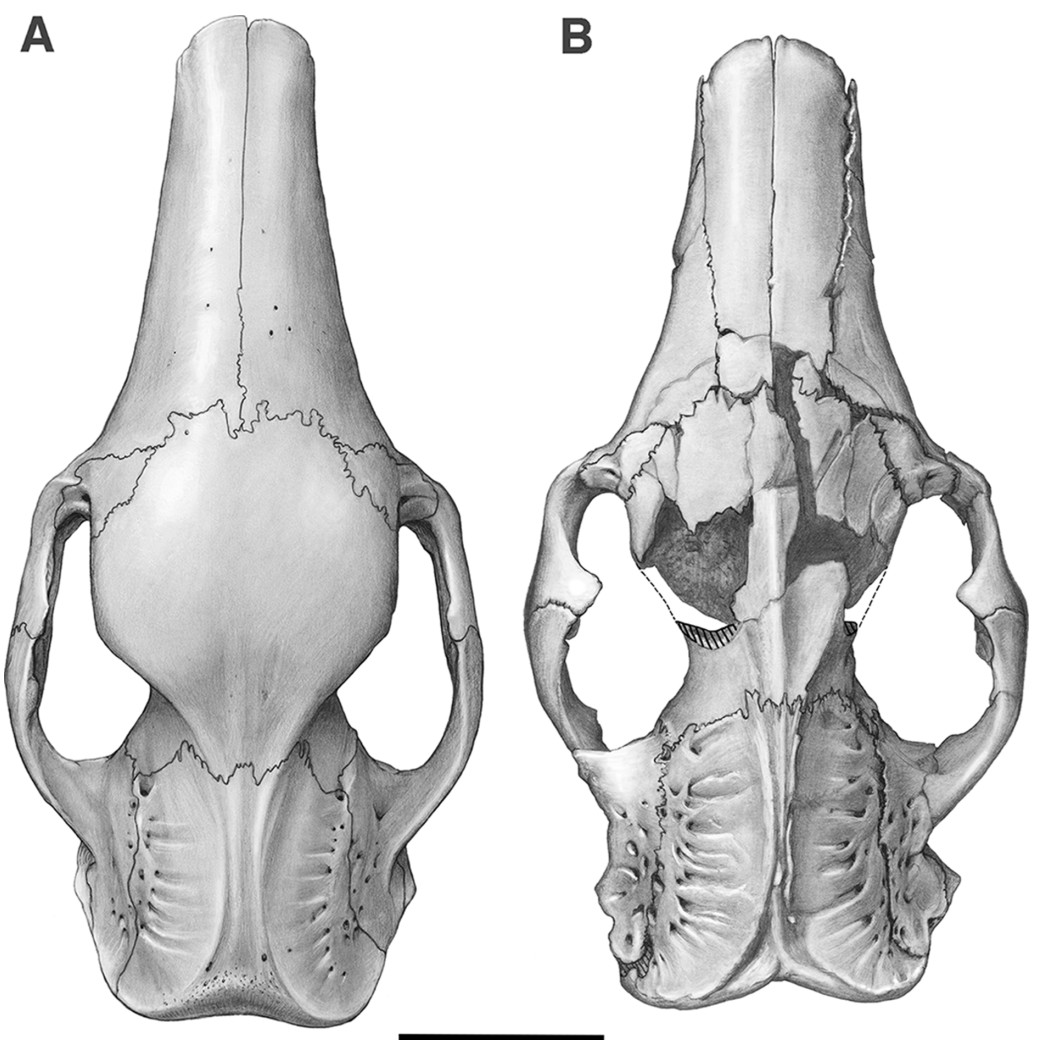

**Figure 2 Skull of *Holmesina floridanus* in dorsal view.** (A) UF 191448; (B) UF 248500. Scale bar = 5 cm.

obscured. The mandible is prepared free in UF 223813 and 248500, but both are damaged to some extent. The left mandible of UF 224450 has also been prepared free. In this specimen the bone is almost perfectly preserved, although it only retains three of nine lower teeth (the second, sixth, and seventh), along with what appears to be a pathological remnant of the fourth. Nevertheless, as the most complete available specimen, it will serve as the primary basis for our description of the mandible.

The pampathere mandible has been described many times in the literature (*Simpson, 1930*; *Castellanos, 1937*; *James, 1957*; *Edmund, 1985b*; *Edmund & Theodor, 1997*; *Vizcaíno, De Iuliis & Bargo, 1998*; *De Iuliis, Bargo & Vizcaíno, 2000*; *De Iuliis & Edmund, 2002*), and, as many of these authors have noted, is broadly similar in its morphology among the various taxa. Since much has already been written about the comparative morphological differences among pampathere mandibles at the generic level, we will focus our comparisons on the species level variation within *Holmesina*.

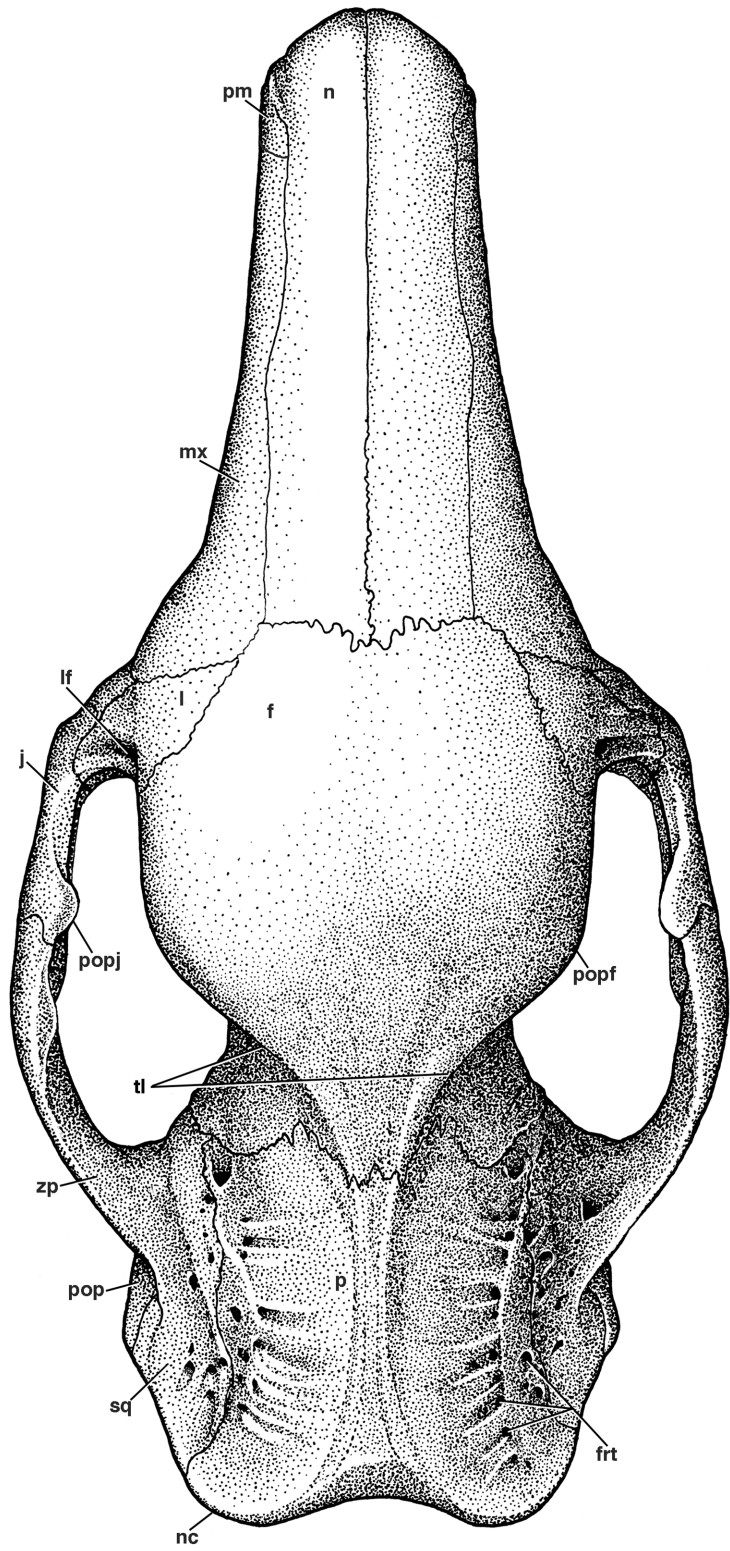

**Figure 3 Reconstruction of the skull of *Holmesina floridanus* in dorsal view. f**, frontal; **frt**, foramina for rami temporalis; **j**, jugal; **l**, lacrimal; **lf**, lacrimal foramen; **mx**, maxilla; **n**, nasal; **nc**, nuchal crest; **p**, parietal; **pm**, premaxilla; **pop**, paroccipital process of petrosal (=mastoid process of *Patterson, Segall & Turnbull (1989)*); **popf**, postorbital process of frontal; **popj**, postorbital process of jugal; **sq**, squamosal; **tl**, temporal lines; **zp**, zygomatic process of squamosal.

**Table 1 Skull measurements for *Holmesina floridanus* and related taxa.**

| Measurement description | *Holmesina floridanus* UF 248500 | *Holmesina floridanus* UF 191448 | *Holmesina septentrionalis* UF 234224 | *Vassalia maxima* FMNH P14424 | *Propalaeho-plophorus australis* YPM VPPU 15007 | *Proeutatus oenophorus* FMNH P13197 | *Euphractus sexcinctus* UTCM 1491 |
|---|---|---|---|---|---|---|---|
| Greatest skull length (GSL) | 227.6* | 249.1 | 293.7 | 248.0 | 158.7 | 117.8* | 119.8 |
| Maximum nasal ln | 89.9 [0.39] | 107.9 [0.43] | 134.0 [0.46] | 117.0 [0.47] | 45[c] [0.28] | 47.9 [0.41] | 42.6 [0.36] |
| Nasal wd at midpoint | 35.6 | 34.9 | 38 | 41 | 23[c] | 10.6 | 15.4 |
| Ratio nasal width to length | 0.40 | 0.32 | 0.28 | 0.35 | 0.51 | 0.22 | 0.36 |
| Rostrum ln (measured from anterior orbital rim) | 110.5 [0.49] | 124.9 [0.50] | 142 [0.48] | 117.0 [0.47] | 45.2 [0.28] | 65* [0.30] | 60.1 [0.50] |
| Premaxilla/nasal suture ln | 19.2 [0.08] | 21.1 [0.08] | – | – | 6.2[c] [0.04] | 13.1 [0.11] | 17.6 [0.15] |
| Mesiodistal ln/max wd of upper molariforms: Mf1 | 7.0/5.5 | – | 10/6 | 6.8/4.5[b] | n | 2.9/1.8 | 4.4/2.3 |
| Mf2 | 7.5/6.1 | – | 13/9 | 8.0/5.5[b] | 3/3.5[d] | 3.4/2.1 | 4.8/2.4 |
| Mf3 | 9.0/6.7 | 9.9/6.4 | 15/8 | 8.5/6.1[b] | 5.5/4[d] | 4.4/2.7 | 4.8/3.1 |
| Mf4 | 10.7/7.1 | 10.3/6.8 | 16/8 | 14.5/6.6[b] | 9/4[d] | 5.5/3.4 | 5.4/3.4 |
| Mf5 | 15.9/8.3 | 16.7/8.6 | 18/10 | 18.5/8.0[b] | 11/4.5[d] | 5.3/4.7 | 5.7/3.9 |
| Mf6 | 16.8/8.7 | – | 22/10 | 19.0/8.6[b] | 12/6[d] | 5.2/5.0 | 6.0/4.5 |
| Mf7 | 15.3/8.1 | 15.0/7.8 | 23/11 | 17.5/8.5[b] | 12.5/7[d] | 4.9/4.6 | 5.6/4.5 |
| Mf8 | 13.3/7.7 | – | 21/9[a] | 16.7/7.5[b] | 12.5/7[d] | 4.3/4.7 | 5.3/4.0 |
| Mf9 | 9.8/5.8 | – | 20/8[a] | 13.7/7.0[b] | 10.5/7[d] | 3.2/3.6 | 4.8/2.9 |
| Mean ratio of upper molariform ln/wd | 1.61 | – | 1.99 | 1.92 | 1.75 | 1.28 | 1.56 |
| Palatal ln (in midline) | 143.6 [0.63] | 163.0 [0.65] | – | 146 [0.59] | 104[d] [0.65] | 64.0 [0.54] | 68.0 [0.57] |
| Min interpterygoid wd | 16.7 [0.07] | 17.8 [0.07] | – | 12 [0.05] | 14 [0.09] | 8.1 [0.07] | 8.1 [0.07] |
| Max zygomatic wd | 121.1 [0.53] | 122.9 [0.49] | – | 138[b] [0.56] | 118 [0.74] | 70.2 [0.60] | 65.6 [0.55] |
| Min interorbital wd | 65.6 [0.29] | 76.2 [0.31] | 89 [0.30] | 79[b] [0.32] | 54 [0.34] | 42.5 [0.36] | 38.5 [0.32] |
| Min postorbital wd | 38.7 [0.17] | 44.3 [0.18] | 56 [0.19] | 52[b] [0.21] | 28 [0.09] | 27.6 [0.23] | 27.5 [0.23] |
| Max wd of glenoid fossa in ventral view (measured along glenoid's long axis) | 23.4 | 23.3 | – | 32 | 31[e] | 8.4 | 9.8 |
| Max anteroposterior ln of glenoid in ventral view | 14.9 | 11.1 | 17 | 12 | 11[e] | 8.0 | 9.8 |
| Ratio of glenoid wd to ln | 1.57 | 2.10 | – | 2.7 | 2.82 | 1.05 | 1.00 |
| Postglenoid skull ln | 43.5 [0.19] | 35.8 [0.14] | 47 [0.16] | 57 [0.23] | 14 [0.09] | 17.1 [0.15] | 20.5 [0.17] |
| Max wd of occipital condyles in ventral view (measured along condyle's long axis) | 21.7 | 24 | 24 | 25 | 61 | 11.4 | 9.6 |
| Max anteroposterior ln of condyles in ventral view | 13.0 | 14.2 | 16 | 15 | 41 | 9.5 | 7.3 |
| Ratio of occipital condyle wd to ln | 1.67 | 1.69 | 1.5 | 1.67 | 1.48 | 1.2 | 1.3 |

Gaudin and Lyon (2017), *PeerJ*, DOI 10.7717/peerj.4022

| Measurement description | Holmesina floridanus UF 248500 | Holmesina floridanus UF 191448 | Holmesina septentrionalis UF 234224 | Vassalia maxima FMNH P14424 | Propalaeho-plophorus australis YPM VPPU 15007 | Proeutatus oenophorus FMNH P13197 | Euphractus sexcinctus UTCM 1491 |
|---|---|---|---|---|---|---|---|
| **Table 1 (continued).** | | | | | | | |
| WD of occiput (measured at base of supraoccipital) | 73.7 | 73.5 | 86 | 97[b] | 63 | 52.1 | 45.6 |
| Max dp of occiput in midline (including ventral edge of foramen magnum) | 72.5 | 70.5 | 83 | 67 | 53 | 36.2 | 32.9 |
| Ratio of wd to dp | 1.02 | 1.04 | 1.04 | 1.44 | 1.19 | 1.44 | 1.39 |

Notes:
All measurements reported in millimeters (mm); those reported to the nearest tenth of a millimeter are direct measurements, those rounded to the nearest integer are taken from literature sources or from photographs taken by TJG. Numbers in square brackets are scaled to greatest skull length (GSL).
–, data unavailable; dp, dorsoventral depth; ln, anteroposterior length; Max, maximum; Min, minimum; n, data not applicable; wd, transverse width.
[*] Estimated due to skull breakage.
[a] Data from UF 889, multiplied by 0.96 to account for size difference between UF 889 and UF 234224.
[b] Data from *De Iuliis & Edmund (2002)*.
[c] Data from YPM VPPU 15291.
[d] Data from *Scott (1903)*, who measured YPM VPPU 15212.
[e] Data from FMNH P13205.

Anatomical terminology, wherever possible, follows that of *Wible & Gaudin (2004)* and *Wible (2010)*. Stereophotographs of UF 248500 were prepared with the assistance of Dr. Stelios Chatzimanolis (University of Tennessee at Chattanooga) in accordance with the procedure outlined in *Gaudin (2011)*.

# RESULTS (DESCRIPTIVE ANATOMY)

## Nasal

The nasals in *Holmesina floridanus* (UF 191448, 248500) consist of two long, transversely convex bones that cover most of the visible surface of the snout in dorsal view (Figs. 2 and 3). The outline of the bones is somewhat variable, with the bones accounting for anywhere between 32–43% of the skull's total length, and the width to length ratio varying from 0.32 to 0.49 (Tables 1 and 2). *E. sexcinctus* (UTCM 1491) falls into the same range for both values, whereas the nasals of *Proeutatus oenophorus* (FMNH P13197) are of similar length but narrower. *Holmesina septentrionalis* (UF 234224) has longer but narrower nasals, *Vassallia maxima* (FMNH P14424) has longer nasals of comparable width, and *Propalaeohoplophorus australis* (YPM VPPU 15007) has nasals that are both shorter and wider (Table 1). In lateral view, the nasals of *Holmesina floridanus* slope gently anteroventrally as in other pampatheres (*Castellanos, 1937*; *Bordas, 1939*; *James, 1957*; *Cartelle & Bohórquez, 1985*; *Edmund, 1985b*; *Edmund & Theodor, 1997*; *Vizcaíno, De Iuliis & Bargo, 1998*; *De Iuliis, Bargo & Vizcaíno, 2000*; *De Iuliis & Edmund, 2002*), as well as in *Propalaeohoplophorus* (*Scott, 1903*), and the extant *E. sexcinctus* (CM 6399, UTCM 1486, 1491). This condition is exaggerated in *Proeutatus oenophorus* (FMNH P13197; *Scott, 1903*), where the posterior half of the nasal bones curve upwards steeply towards the frontal bone.

In dorsal view, the anterior margin of the nasal bones in *Holmesina floridanus* is convex, which is a synapomorphy of Cingulata (*Gaudin & Wible, 2006*; *Gaudin & McDonald,*

**Table 2 Skull measurements for additional specimens of *Holmesina floridanus*.**

| Measurement description | *Holmesina floridanus* UF 223813 | UF 275496 | UF 275497 | UF 275498 | UF 285000 | UF 293000 |
|---|---|---|---|---|---|---|
| Greatest skull length (GSL) | 256* | 237.8 | – | 223* | 239.5 | – |
| Maximum nasal ln | 81 [0.32] | 85.4 [0.36] | 69.7 | 70.0 [0.34] | 85.1 [0.36] | 88.0 |
| Nasal wd at midpoint | 34.4 | 35.0 | 33.8 | 34.2 | 37.8 | 35.0 |
| Ratio nasal width to length | 0.42 | 0.41 | 0.48 | 0.49 | 0.44 | 0.40 |
| Rostrum ln (measured from anterior orbital rim) | 122 [0.48] | 111 [0.47] | 106 | 103 [0.46] | 113 [0.47] | 104 |
| Premaxilla/nasal suture ln | – | 18.0 [0.08] | 19.8 | 22.3 [0.10] | 17.9 [0.07] | 17.4 |
| Mesiodistal ln/max wd of upper molariforms: Mf1 | 7.1/5.4 | 6.0/5.7 | – | 6.8/5.5 | 5.9/5.8 | 6.9/4.3 |
| Mf2 | 8.2/5.8 | – | – | 7.9/5.6 | 7.9/6.1 | 8.0/4.8 |
| Mf3 | 9.5/6.1 | 9.7/6.1 | – | 10.4/6.3 | 9.8/6.1 | 10.2/5.4 |
| Mf4 | 11.7/7.0 | 11.3/7.2 | – | 11.5/7.1 | 12.3/6.7 | 11.7/6.0 |
| Mf5 | 16.0/9.1 | 15.6/8.7 | 13* | 16.6/8.8 | 15.1/8.4 | 16.6/8.0 |
| Mf6 | 16.8/8.5 | 16.9/8.9 | 15.7/7.6 | 18.7/9.0 | 17.2/8.3 | 17.9/7.9 |
| Mf7 | 15.4/8.0 | 15.5/8.4 | 14.9/7.0 | – | 16.6/8.2 | 16.1/7.0 |
| Mf8 | 13.5/7.5 | 13.7/7.9 | – | – | 14.1/8.0 | 15.7/6.6 |
| Mf9 | 10.3/6.0 | 8.6/6.1 | 9.3/6.0 | – | – | 10.2/5.8 |
| Mean ratio of upper molariform ln/wd | 1.68 | – | – | – | – | 1.99 |
| Palatal ln (in midline) | 156* [0.61] | 145* [0.61] | – | – | 149 [0.62] | 155 |
| Min interpterygoid wd | – | – | – | – | – | – |
| Max zygomatic wd | – | – | – | – | – | – |
| Min interorbital wd | – | 57 [0.24] | – | 60 [0.27] | – | 55 |
| Min postorbital wd | – | – | – | 42 [0.19] | – | – |
| Max wd of glenoid fossa in ventral view (measured along glenoid's long axis) | – | – | – | – | – | 29 |
| Max anteroposterior ln of glenoid in ventral view | 12.6 [0.05] | – | 13.6 | 14.5 [0.07] | 14.3 [0.06] | 12.7 |
| Ratio of glenoid wd to ln | – | – | – | – | – | 2.28 |
| Postglenoid skull ln | 46* [0.18] | 44 [0.19] | – | 42 [0.19] | 40 [0.17] | – |
| Max wd of occipital condyles in ventral view (measured along condyle's long axis) | 22.2 | 22.7 | 20.8 | 21.3 | – | 24.2 |
| Max anteroposterior ln of condyles in ventral view | 13.5 | 13.3 | 12.7 | 14.0 | – | 14.0 |
| Ratio of occipital condyle wd to ln | 1.64 | 1.71 | 1.64 | 1.52 | – | 1.73 |
| Wd of occiput (measured at base of supraoccipital) | 69.8 | 73.7 | – | 66.7 | 70.6 | 68 |
| Max dp of occiput in midline (including ventral edge of foramen magnum) | – | 77* | – | 64.0 | – | 64.7 |
| Ratio of wd to dp | – | 0.96 | – | 1.04 | – | 1.05 |

Notes:
All measurements reported in millimeters (mm). Numbers in square brackets are scaled to greatest skull length (GSL).
–, data unavailable; dp, dorsoventral depth; ln, anteroposterior length; Max, maximum; Min, minimum; wd, transverse width.
* Estimated due to skull breakage.

2008). UF 284500 has distinct lateral sutures running the length of the nasals, whereas the sutures with the maxilla and premaxilla are largely fused in UF 191448. Nasal width is uniform from the anterior tip to the maxillo-premaxillary suture, where it then gently narrows posteriorly as it approaches the frontal bone. There appear to be two major

fronto-nasal suture patterns that occur in *Holmesina floridanus*. One of the patterns occurs in UF 191448, as a roughly straight though highly irregular suture (Fig. 2). The other pattern, observed in multiple specimens (UF 223813, 275496, 275497, 275498; 285000) is a shallow V-shaped suture with the apex directed anteriorly. The nasals of UF 248500 are fractured posteriorly, and the bone is clearly incomplete in places, making it hard to discern the course of its fronto-nasal suture. In *Holmesina septentrionalis* (UF 889), the overall shape of the nasal is similar to that of *Holmesina floridanus*. However, the fronto-nasal suture varies in form and may differ substantially from that of *Holmesina floridanus*. In UF 889 it forms a distorted W-shape, due to a large median peak with a posteriorly directed apex. Conversely, in UF 234224 it is roughly straight, but irregular, as in *Holmesina floridanus* (UF 191448). The fronto-nasal suture in *Holmesina occidentalis* (ROM 3881) forms a very shallow, anteriorly concave jagged "U." In *Vassallia* (FMNH P14424) and *Holmesina rondoniensis* (*Góis et al., 2012*) the suture is a shallow V-shape, reminiscent of some *Holmesina floridanus* specimens, except that the apex is directed posteriorly. Similarly, *Pampatherium humboldti* has a W-shaped fronto-nasal suture, but with the median apex directed anteriorly (*Góis et al., 2012*). It is clear from our survey that the shape of the fronto-nasal suture varies widely among pampatheres; large variation in this suture has also been observed in other mammals (e.g., typotherian notoungulates, *Sinclair, 1909*). In our reconstruction of *Holmesina floridanus* we have chosen to illustrate a condition like that in UF 191448 (Fig. 2).

The suture is unknown in *Propalaehoplophorus* (*Scott, 1903*; *Vizcaíno, De Iuliis & Bargo, 1998*). Like some *Holmesina*, the fronto-nasal suture of *Proeutatus* (FMNH P13197) forms a V-shape, with the apex pointing anteriorly. In *Euphractus* it is roughly straight near the lateral edges of the nasal bones, but as it approaches the median suture it too forms an anteriorly directed V-shape, albeit a smaller one than that of *Proeutatus* (*Wible & Gaudin, 2004*).

## Premaxilla

In lateral view, the premaxilla has a broad rectangular facial process, with its dorsoventral height slightly exceeding its anteroposterior length (Figs. 4 and 5). The maxillo-premaxillary suture of the facial process in *Holmesina floridanus* (UF 248500) forms a single posteriorly convex curve. The premaxillary sutures are harder to distinguish in UF 191448, but they appear similar. The dorsal suture between the premaxilla and nasal is relatively short in *Holmesina* (7–10% of GSL in *Holmesina floridanus*; Tables 1 and 2—though not listed in the table, the value for *Holmesina occidentalis* [ROM 3881] is about 7% of GSL) relative to the extant *Euphractus* (15% of GSL, UTCM 1491), though not as short as in glyptodonts (4% of GSL, *Propalaehoplophorus* YPM VPPU 15291; Table 1). *Proeutatus* (11% of GSL, FMNH P13197) is similar in this regard to *Holmesina*.

The free anterior edge of the facial process is vertical but irregularly shaped. The dorsal portion of this edge has a deep and narrow notch in UF 248500 (Figs. 4B and 4C) and UF 121742, which slopes anteroventrally into a large triangular prong. In both UF 191448 (Fig. 4A) and UF 285000 the anterior edge is marked by a shallower, more rounded notch,

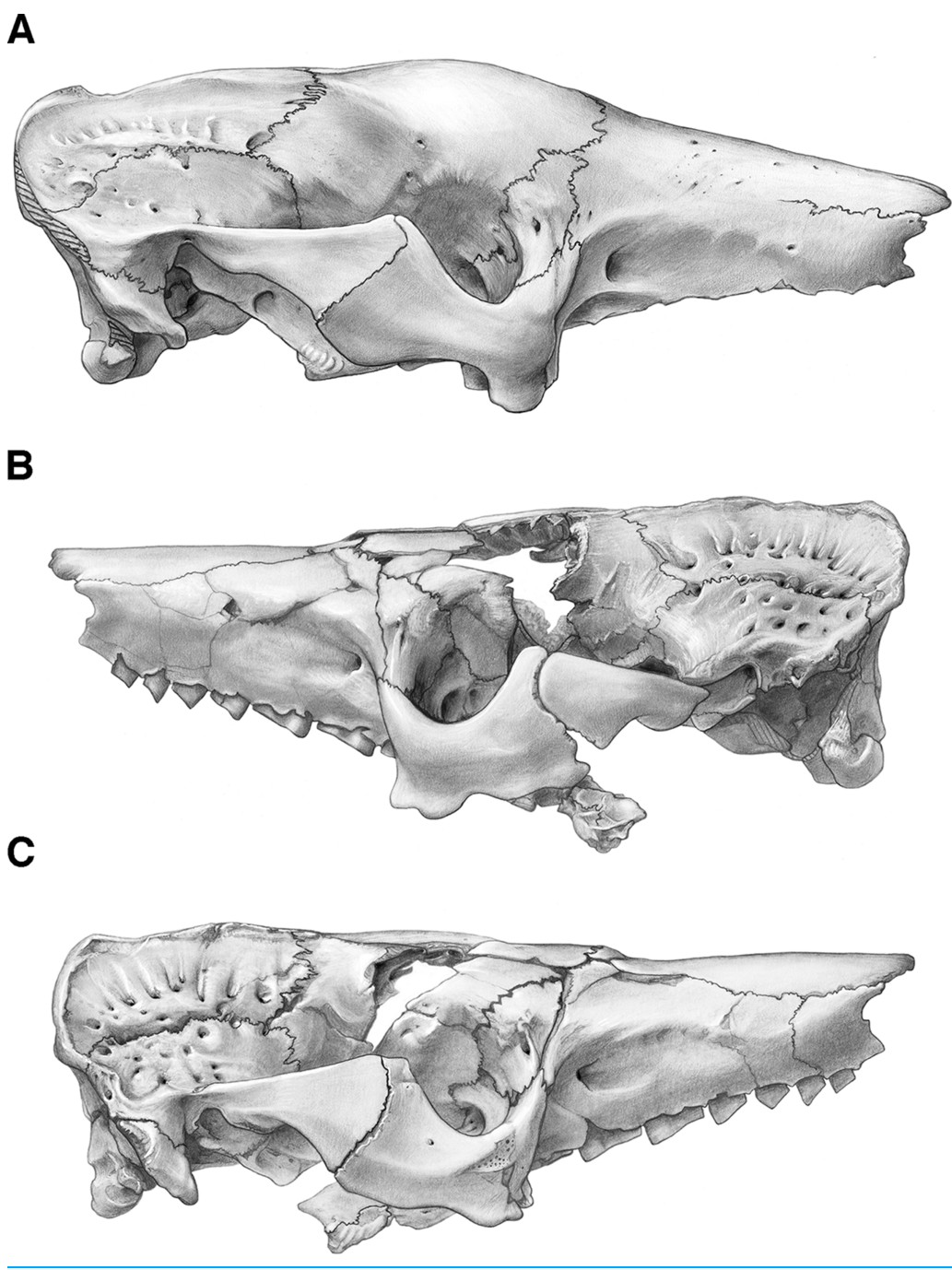

**A**

**B**

**C**

**Figure 4 Skull of *Holmesina floridanus* in lateral view.** (A) UF 191448 in right lateral view; (B) UF 248500 in left lateral view; (C) UF 248500 in right lateral view. Scale bar = 5 cm.

ending in a small bump on its ventral margin. *Holmesina septentrionalis* and *Vassallia maxima* (*Edmund, 1985b*; *De Iuliis & Edmund, 2002*) also have notches that are deep and narrow, as in UF 191448, whereas *Holmesina occidentalis* (ROM 3881) has a shallower C-shaped notch more like UF 248500. *Propalaehoplophorus* has a very shallow C-shaped notch on the anterior edge of its very tall and narrow premaxillary facial process

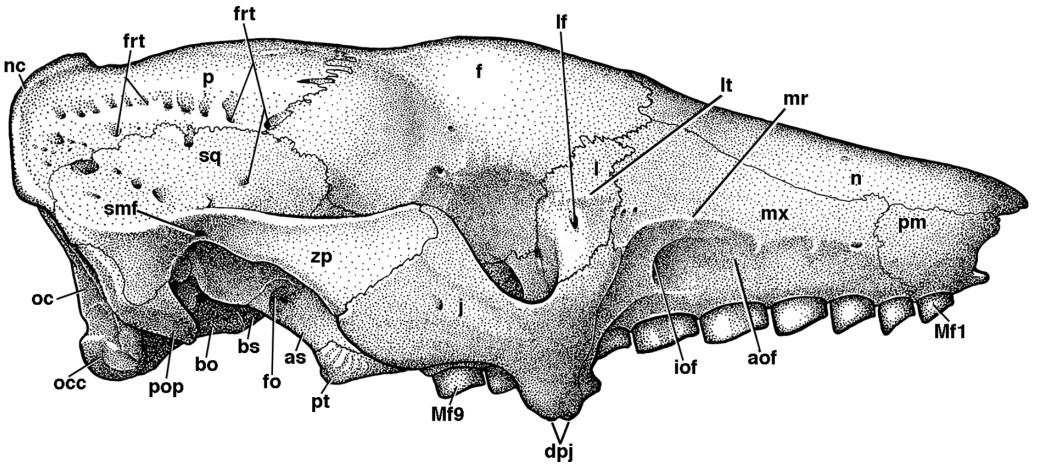

**Figure 5 Reconstruction of the skull of *Holmesina floridanus* in right lateral view. aof**, antorbital fossa; **as**, alisphenoid; **bo**, basioccipital; **bs**, basisphenoid; **dpj**, two projections forming descending process of jugal; **f**, frontal; **fdv**, foramen for frontal diploic vein; **fo**, foramen ovale; **frt**, foramina for rami temporalis; **iof**, infraorbital foramen; **j**, jugal; **l**, lacrimal; **lf**, lacrimal foramen; **lt**, lacrimal tubercle; **Mf1**, first upper molariform tooth; **Mf9**, ninth upper molariform tooth; **mr**, maxillary ridge, i.e., ridge on facial process of maxilla; **mx**, maxilla; **n**, nasal; **nc**, nuchal crest; **oc**, occipital; **occ**, occipital condyle; **p**, parietal; **pm**, premaxilla; **pop**, paroccipital process of petrosal (=mastoid process of *Patterson, Segall & Turnbull (1989)*); **pt**, pterygoid; **smf**, suprameatal foramen; **sq**, squamosal; **zp**, zygomatic process of squamosal.                      

(YPM-VPPU 15291). In *Euphractus*, the anterior margin of the premaxilla is variable in shape—it may be a relatively straight edge sloping posteroventrally (*Wible & Gaudin, 2004*), it may be marked by a wide, shallow, C-shaped notch (e.g., UTCM 1500), or the entire edge may form a single shallow concavity (e.g., UTCM 1486, 1491). The anterior edge of the premaxilla in *Proeutatus* (FMNH P13197) slopes posteroventrally in lateral view, as in *Euphractus*, and it lacks the notch that is present in pampatheres, glyptodonts, and some *Euphractus* (*Wible & Gaudin, 2004*).

The external nares of *Holmesina floridanus* are widest transversely near the nasopremaxillary suture. From there the premaxilla slopes steeply inward ventromedially. In anterior view UF 248500 appears to have an irregularly rounded, upside-down triangular shaped nasal opening. The nares in UF 191448 have a more rounded, inverted pentagonal cross-section, much like that of *Holmesina septentrionalis* (UF 234224). The nasal opening is more ovate and dorsoventrally compressed in both *Proeutatus* (FMNH P13197) and *Euphractus* (CM 6399; UTCM 1486, 1491).

In ventral view, the premaxilla of *Holmesina floridanus* forms a roughly M-shaped palatal suture with the maxilla (Figs. 5 and 6), similar to that of *Holmesina septentrionalis* (UF 889). The maxillo-premaxillary suture exhibits a high degree of variability in other species.

In *Holmesina floridanus*, the anteroventral tip of the premaxilla extends forward in the midline as a rounded prong in UF 191448, though this prong is strongly reduced in UF 248500. *Holmesina septentrionalis* (*Edmund, 1985b*) has a similar, though transversely broader, U-shaped anteroventral prong, and a prong very like that of UF 191448 is also present in *Vassallia* (*De Iuliis & Edmund, 2002*). *Propalaehoplophorus* differs in that the

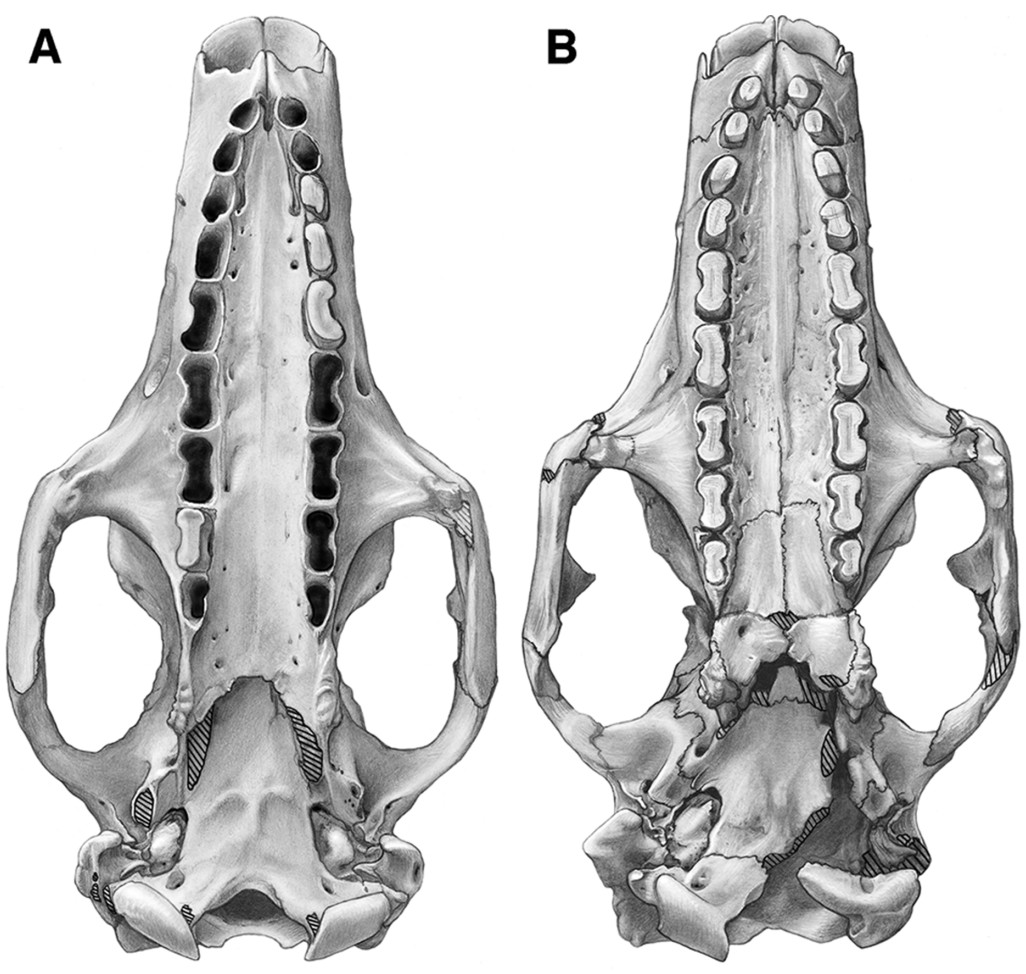

**Figure 6 Skull of *Holmesina floridanus* in ventral view.** (A) UF 191448; (B) UF 248500. Scale bar = 5 cm.

anteroventral edge of the premaxilla forms extensions that project forward to form a distorted M-shape, with long anterolateral edges and a short V-shaped median notch. The premaxillae of *Proeutatus* and *Euphractus* lack anteroventral extensions (*Scott, 1903*; *Wible & Gaudin, 2004*).

The palatal process of the premaxilla in *Holmesina floridanus* is incised by a deep groove that emerges from the front of the incisive foramina (Figs. 6 and 7). The incisive foramen transmits the nasopalatine duct, which connects the oral and nasal cavities with the vomeronasal organ. It also transmits the nasopalatine nerve, artery and vein (*Wible & Gaudin, 2004*). The incisive foramina themselves are deeply recessed posterodorsally, with separate left and right openings that empty into a single midline fossa. This appears to be a general feature of pampatheres, but it is an unusual morphology among cingulates. Other cingulates, such as *Proeutatus* (FMNH P13197) and *Euphractus* (CM 6399; UTCM 1481, 1486), have a common fossa that houses the two separate incisive foramina, and all cingulates (except perhaps glyptodonts; see *Gillette & Ray (1981*, fig. 11c)) have close set

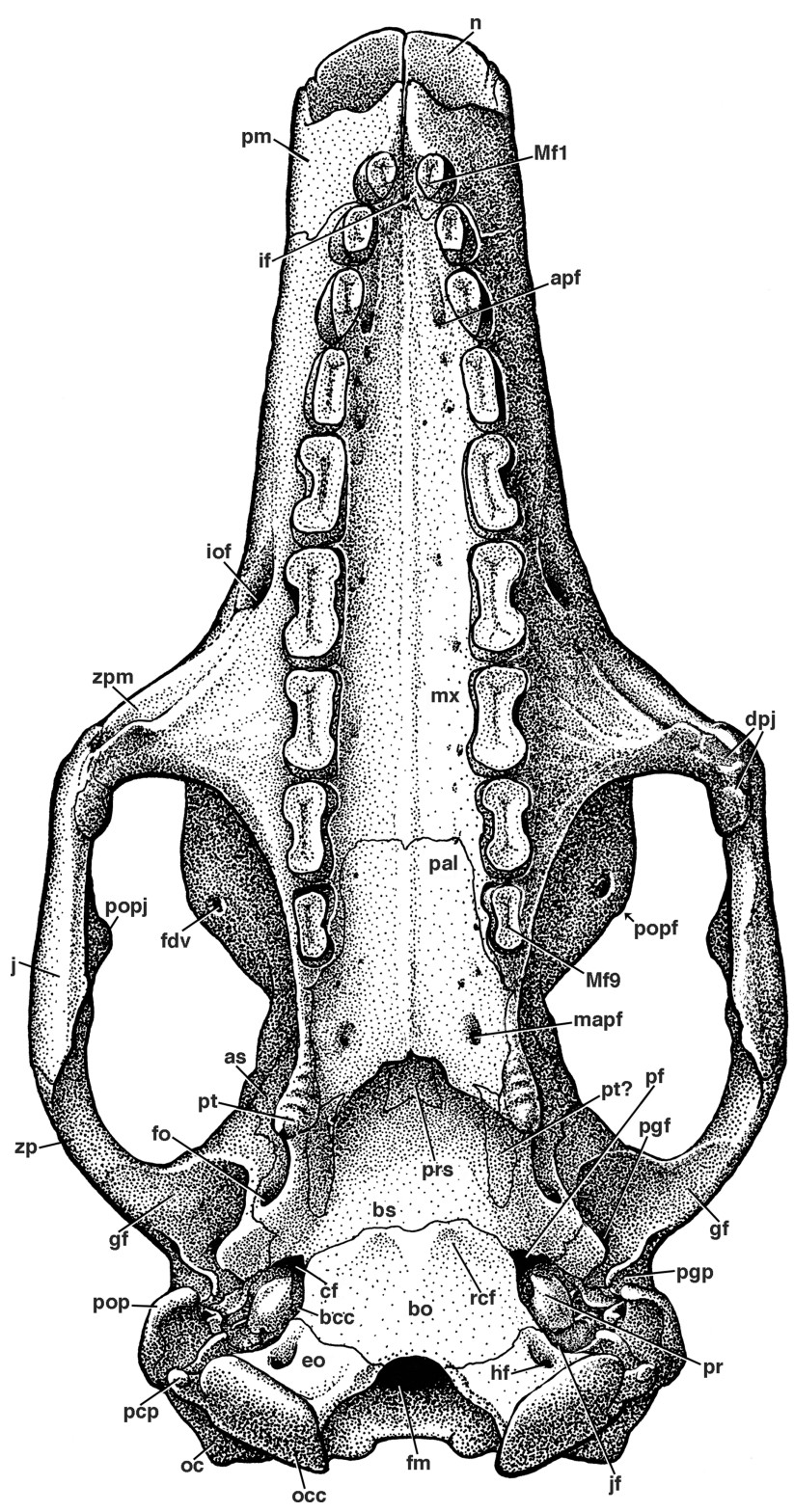

**Figure 7 Reconstruction of the skull of *Holmesina floridanus* in ventral view. apf**, anterior palatal foramen; **as**, alisphenoid; **bcc**, basicochlear commissure; **bo**, basioccipital; **bs**, basisphenoid; **cf**, carotid foramen; **dpj**, two projections forming descending process of jugal; **eo**, exoccipital; **fdv**, foramen for frontal diploic vein; **fm**, foramen magnum; **fo**, foramen ovale; **gf**, glenoid fossa; **hf**, hypoglossal foramen; **if**, incisive foramen; **iof**, infraorbital foramen; **jf**, jugular foramen; **mapf**, major palatine foramen; **Mf1**, first upper molariform tooth; **Mf9**, ninth upper molariform tooth; **mx**, maxilla; **n**, nasal; **oc**, occipital; **occ**, occipital condyle; **pal**, palatine; **pcp**, paracondylar process of exoccipital (=paroccipital process of *Patterson, Segall & Turnbull (1989)*); **pf**, piriform fenestra; **pgf**, postglenoid foramen; **pgp**, postglenoid process; **pm**, premaxilla; **pop**, paroccipital process of petrosal (=mastoid process of *Patterson, Segall & Turnbull (1989)*); **popf**, postorbital process of frontal; **popj**, postorbital process of jugal; **pr**, promontorium of petrosal; **prs**, presphenoid; **pt**, pterygoid; **rcf**, rectus capitis fossa; **zp**, zygomatic process of squamosal; **zpm**, zygomatic process of maxilla.

incisive foramina. However, in no other cingulates are they as deeply recessed, and no other cingulates possess the deep anterior groove found in pampatheres. As in all other cingulates, aside from *Peltephilus* (*Gaudin & Wible, 2006*), the incisive foramina in *Holmesina floridanus* are completely encompassed by the premaxilla.

The premaxilla retains a single tooth near its posterior border with the maxilla. The right maxillary–premaxillary suture runs into the mesial portion of the socket of the second tooth. The premaxilla encompasses the labial half of the second tooth socket, but forms only the front of the socket on the lingual side. The presence of premaxillary teeth is a synapomorphy of euphractine armadillos, glyptodonts, and pampatheres (Node C of *Gaudin & Wible, 2006*), though it is lost secondarily in glyptodonts. The premaxillary tooth of *Holmesina floridanus* is angled anteriorly and slightly medially. It has beveled wear facets on the occlusal surface. The surface area of the mesial facet is greater than that of the distal facet in most specimens, though the distal is larger in UF 293000 and highly reduced in UF 121742 and 275496, and the two facets lie at a 110 degree angle to one another. The fact that UF 275496 appears to be a juvenile based both on its open sutures and the less finished surface texture of its skull bones, whereas UF 293000 and 121742 appear to be adults based on the same criteria, suggests that these differences in wear facet shape are not necessarily age-related. The overall outline of the occlusal surface is ovate, with its mesiodistal length exceeding its transverse width (Table 1). The left premaxillary tooth in UF 248500 possesses a small lenticular island of osteodentine in the center, whereas the right tooth has a narrow linear island of osteodentine. Presence of an elevated core of osteodentine is a synapomorphy of *Proeutatus*, glyptodonts, and pampatheres (Node 7 of *Gaudin & Wible, 2006*), as is the presence of beveled wear facets only in the anterior portion of the tooth row. In both *Holmesina occidentalis* (ROM 3881), and *Proeutatus* (FMNH P13197) the premaxillary tooth has an ovate occlusal surface, similar to *Holmesina floridanus*. In *Vassallia* and *Holmesina septentrionalis* the premaxillary teeth are missing (*Edmund, 1985b*; *De Iuliis & Edmund, 2002*), but it can be ascertained from the shape of the tooth alveoli in these animals that they too had ovate occlusal surfaces, making this a shared trait among cingulate taxa that possess premaxillary teeth. In *Euphractus* (CM 6399; UTCM 1486, 1491) the premaxillary tooth is mostly flat at its tip, with a small discolored island in the center, likely formed from orthodentine (*Ferigolo, 1985*; *Kalthoff, 2011*).

## Maxilla

The facial process of the maxilla contacts the nasal dorsally, the premaxilla anteriorly, and the frontal and lacrimal posteriorly (Figs. 4–7). The large zygomatic process of the maxilla

contacts the jugal posteriorly. The facial process is marked by a ridge that runs anteroposteriorly just below the nasomaxillary suture (Fig. 5). In *Holmesina*, this ridge begins as an indistinct, broad elevation above Mf2/Mf3 (=second and third molariform teeth; note all teeth in pampatheres and glyptodonts are molariform) that becomes a more pronounced, low ridge above Mf4, and finally forms a sharply defined ridge over Mf6. The ridge then curves posteroventrally to become confluent with the maxilla/jugal suture and a large rounded ridge that marks the anterior termination of the jugal and outlines the distinct antorbital fossa (Fig. 5; *Wible & Gaudin, 2004*; =buccinator fossa from *Gaudin (2004)*). A nearly identical lateral maxillary ridge is present in the other *Holmesina* species (*Holmesina occidentalis, Holmesina septentrionalis*). The ridge in *Vassallia*, though present, is less distinct (*De Iuliis & Edmund, 2002*) than it is in *Holmesina*. *Euphractus* (CM 6399; UTCM 1486, 1491) also has a distinct maxillary ridge that begins over Mf3 and marks the dorsal edge of a strong antorbital fossa (*Wible & Gaudin, 2004*). In *Holmesina floridanus*, the antorbital fossa is particularly large and deep posteriorly behind the infraorbital foramen, as well as on the anterior surface of the zygomatic process of the maxilla. This fossa accommodates the nasiolabialis muscles (*Smith & Redford, 1990*; *Vizcaíno, De Iuliis & Bargo, 1998*; *Wible & Gaudin, 2004*). In dorsal view, the maxilla forms a small portion of the roof of the snout as it touches the nasal bone (Figs. 2 and 3). It also comprises the majority of the lateral walls of the snout, which taper anteriorly in both lateral and dorsal views (Figs. 2–5). The antorbital fossa is less well marked in *Proeutatus*, and is absent in *Propalaehoplophorus* (*Scott, 1903*; *Gaudin & Wible, 2006*).

The palatine process of the maxilla is broadly concave anteroposteriorly from Mf1 to Mf7. The palate, including both maxillary and palatine contributions, is convex from Mf7 to the posterior edge of the palate, but concave transversely along its whole length. Both the longitudinal and transverse concavities are especially deep anteriorly, near the junction of the maxilla and premaxilla. The hard palate is marked by numerous foramina (Figs. 6 and 7), as in other xenarthrans (*Gaudin & Wible, 2006*). This is due to the fact that the major palatine arteries, veins, and nerves travel within the palatal process of the maxilla (*Wible & Gaudin, 2004*), rather than on its ventral surface, as in other mammals (e.g., *Canis, Evans & Christiansen 1979*; *Homo, Clemente 1985*). These nerve and vessels finally emerge ventrally from their canal in the maxilla near the front of the palate, through the anterior palatal foramina. Anterior palatal foramina are typically located near Mf4 (e.g., in UF 248500) in *Holmesina floridanus*, but they exhibit some variation in their position in different specimens. For example, in UF 191448, both are near the distal half of Mf3 (Figs. 6 and 7), but on the left side of UF 121742, they are as far back as the mesial half of Mf5. The anterior palatal foramina occupy similar, somewhat varying positions in *Holmesina occidentalis, Holmesina septentrionalis*, and *Vassallia*, showing only slightly greater variation than that found in *Holmesina floridanus* itself—one specimen of *Holmesina septentrionalis* (UF 234224) had the foramina situated a little further forward, at the mesial edge of Mf3 or between Mf2 and Mf3. In all of these species, the foramina open anteriorly into distinct grooves that travel forward, ending just short of the maxillo-premaxillary suture. This anterior palatal foramina and grooves are also present in glyptodonts (*Gaudin, 2004*) and *Proeutatus* (FMNH P13197). The characteristic is

convergent on a similar feature shared by pilosans (*Gaudin, 2004*; *Wible & Gaudin, 2004*; *De Iuliis, Gaudin & Vicars, 2011*).

The median suture of the maxilla is slightly raised from the distal edge of Mf5 posteriorly to the junction with the palatine in *Holmesina floridanus* (Figs. 6 and 7). This trait is also present in *Holmesina occidentalis* (ROM 3881), *Vassallia* (*De Iuliis & Edmund, 2002*), *Propalaehoplophorus* (*Scott, 1903*), *Proeutatus* (FMNH P13197), and *Euphractus* (CM 6399). In *Holmesina floridanus*, the apex of the U-shaped maxillary/palatine suture reaches as far anteriorly as the middle of Mf8. The suture travels posteriorly just medial to the tooth alveoli of Mf8 and Mf9, and then curves laterally behind this last tooth in front of the pterygoid process. A U-shaped maxillo-palatine suture with rounded anterolateral corners is a derived feature of *Proeutatus* and living euphractines (Node 6 of *Gaudin & Wible, 2006*), but this condition also occurs in *Holmesina floridanus* and *Holmesina occidentalis* (ROM 3881). The maxilla/palatine suture is unknown in *Holmesina septentrionalis* and *Propalaehoplophorus*, whereas in *Vassallia*, the suture is M-shaped (*De Iuliis & Edmund, 2002*; *Gaudin & Wible, 2006*).

The zygomatic process of the maxilla is sizeable, and forms most of the anterior wall of the orbit in pampatheres (*Gaudin & Wible, 2006*). In ventral view, the zygomatic process is triangular with a broad base and narrow apex extending laterally at a right angle to the main body of the maxilla (Figs. 6 and 7). The ovate infraorbital foramen in *Holmesina floridanus* is situated above Mf6, and opens anteriorly into a short groove. The maxillary foramen lies above the posterior half of Mf7 (UF 121742, 248500; 285000) or the anterior half of Mf8 (UF 191448). It is triangular in shape, and serves as the posterior entrance to a long infraorbital canal that perforates the base of the zygomatic process. This canal is riddled with many smaller foramina along its medial wall, as occurs in *Euphractus* (*Wible & Gaudin, 2004*). In *Holmesina occidentalis, Holmesina septentrionalis* and *Vassallia*, the infraorbital canal also extends from Mf8–Mf6 (ROM 3881; UF 234224; *Edmund, 1985b*; *De Iuliis & Edmund, 2002*); thus, this appears to be a characteristic of pampatheres in general. In contrast, *Propalaehoplophorus* has a more dorsally situated infraorbital canal than that of pampatheres. The canal is relatively short, its entire length located above Mf6–Mf5 (*Scott, 1903*). *Proeutatus* also has a short, dorsally positioned infraorbital canal that begins above Mf7 and exits above Mf5/Mf6, and lies above the antorbital fossa. In *Euphractus* (CM 6399), the canal is intermediate in length between that of *Proeutatus* and *Holmesina*, beginning over the posterior half of Mf7 and exiting over the anterior half of Mf6. The infraorbital canal transmits the infraorbital nerves and vessels from the orbit to the snout (*Wible & Gaudin, 2004*).

Sutures are fused or poorly marked in the orbit of UF 191448, and large portions of the orbital process of the maxilla are missing or heavily fractured in UF 248500, though the sutures are more clearly visible in the latter specimen. That said, the orbital process of the maxilla appears to comprise the anteroventral part of the medial wall of the orbit (Fig. 8) as in most cingulates, with the exception of dasypodine armadillos (*Gaudin & Wible, 2006*). The orbital exposure of the maxilla borders the lacrimal anterodorsally, the frontal posterodorsally, and the alisphenoid, pterygoid (or palatine; see description of palatine below), and orbitosphenoid posteriorly. Atypical of other mammals and even

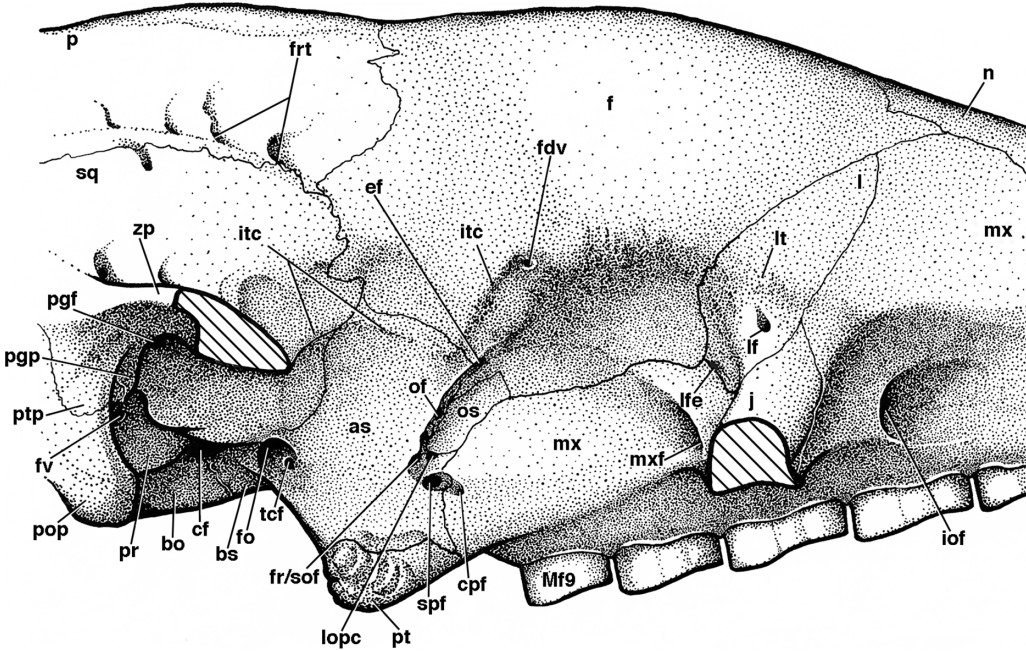

**Figure 8 Reconstruction of right orbital wall of *Holmesina floridanus* in lateral view.** Cross-hatched surfaces indicate where zygomatic arch is "cut." **as**, alisphenoid; **bo**, basioccipital; **bs**, basisphenoid; **cf**, carotid foramen; **cpf**, caudal palatine foramen; **ef**, ethmoid foramen; **f**, frontal; **fdv**, foramen for frontal diploic vein; **fo**, foramen ovale; **fr/sof**, fused foramen rotundum and sphenorbital fissure; **frt**, foramina for rami temporalis; **fv**, fenestra vestibuli; **iof**, infraorbital foramen; **itc**, infratemporal crest; **j**, jugal; **l**, lacrimal; **lf**, lacrimal foramen; **lfe**, lacrimal fenestra; **lopc**, lateral opening of pterygoid canal; **lt**, lacrimal tubercle; **Mf9**, ninth upper molariform tooth; **mx**, maxilla; **mxf**, maxillary foramen; **n**, nasal; **of**, optic foramen; **os**, orbitosphenoid; **p**, parietal; **pgf**, postglenoid foramen; **pgp**, postglenoid process; **pop**, paroccipital process of petrosal (=mastoid process of *Patterson, Segall & Turnbull (1989)*); **pr**, promontorium of petrosal; **pt**, pterygoid; **ptp**, post-tympanic process of squamosal; **spf**, sphenopalatine foramen; **sq**, squamosal; **tcf**, transverse canal foramen; **zp**, zygomatic process of squamosal.

other cingulates, pampatheres and glyptodonts possess a sphenopalatine foramen that is housed in a common fossa with the sphenorbital fissure, though this fossa in *Holmesina floridanus* is partially walled laterally by an anterior bridge of the alisphenoid that contacts the maxilla (Fig. 8). The opening of the sphenopalatine foramen is directed cranially (UF 121742). Within the orbit the maxilla forms the anterior edge of the sphenopalatine foramen, whereas the alisphenoid (or palatine; see description of palatine below) forms the posterior edge. In *Euphractus*, the sphenopalatine foramen lies between the maxilla and palatine (*Wible & Gaudin, 2004*).

The presence of nine upper teeth is the primitive condition in *Proeutatus*, euphractine armadillos, and pampatheres (Node 3 of *Gaudin & Wible, 2006*), with all but the first (Mf2–Mf9) housed in the maxilla. *Propalaehoplophorus* has only eight teeth, since it is missing the premaxillary tooth, as noted above. Therefore, we believe the first tooth in *Propalaehoplophorus* is homologous with Mf2 in pampatheres (though see *González-Ruiz et al. (2015)* for contrasting interpretation), and we will label it as such for comparative purposes. UF 248500 preserves a complete dentition (Fig. 6B), whereas in UF 191448 there are only four teeth remaining (the left Mf3–Mf5, and the right Mf8; Fig. 6A). Among

other *Holmesina floridanus*, UF 121742 also has a complete dentition, whereas at least partial dentitions are visible in the incompletely prepared specimens UF 223813, 275496, 285000, and 293000. The upper molariforms in *Holmesina floridanus* are relatively short and broad compared to those in other pampatheres or glyptodonts (Tables 1 and 2). The occlusal surfaces of Mf2 and Mf3 are ovate in outline. The occlusal surface of Mf4 is ovate in UF 191448 and almost rectangular in UF 293000, but reniform in UF 248500 and most other specimens, with an occlusal surface that is concave lingually and convex labially. In UF 191448, Mf5 is reniform and concave labially, and Mf5 is bilobate in UF 285000 and 275498, whereas in UF 248500 and the other *Holmesina floridanus* specimens, Mf5–Mf7 are trilobate on the lingual side, and bilobate on the labial side of the tooth, though the middle lingual lobe is often poorly marked. This causes these teeth to retain a bilobate gestalt, as is typical for pampatheres (*Hoffstetter, 1958*; *Edmund, 1985b*; *Edmund & Theodor, 1997*; *De Iuliis & Edmund, 2002*). Mf8 and Mf9 are bilobate on both sides of the jaw. The presence of reniform occlusal surfaces on the anterior teeth and bilobate occlusal surfaces on the posterior teeth appears to be a characteristic of pampatheres. *Holmesina septentrionalis* has occlusal surfaces that are reniform from Mf2 to Mf4, but bilobate from Mf5 to Mf9, as in *Holmesina floridanus* (*Edmund, 1985b*). *Holmesina occidentalis* (ROM 3881) differs from *Holmesina floridanus* and *Holmesina septentrionalis* in that Mf3–Mf4 are more ovate in outline, and the posterior lobes are displaced slightly laterally in Mf6–Mf9, whereas in other pampatheres the lobes are linearly arranged. *Vassallia* is missing most of its teeth, but the occlusal surface of the left Mf6 appears to be similar in shape to that of *Holmesina floridanus*, albeit with deeper lateral lobes (*De Iuliis & Edmund, 2002*). *Scirrotherium*, *Kraglievichia*, and *Pampatherium* appear to differ mainly in the size and shape of Mf4, with the tooth smaller and more ovate in *Scirrotherium* (*Edmund & Theodor, 1997*), and relatively larger than *Holmesina* and bilobate in shape in the latter two genera (*Simpson, 1930*; *De Iuliis, Bargo & Vizcaíno, 2000*). In *Propalaehoplophorus*, the anterior teeth are reniform, or weakly lobate in the case of Mf4, reminiscent of the condition in pampatheres. However the posterior teeth are distinct in outline, with Mf5–Mf6 irregularly shaped, weakly bilobate labially and trilobate lingually, whereas Mf7–Mf9 are strongly trilobate on both sides. This trilobate pattern is a defining feature of glyptodonts (*Hoffstetter, 1958*; *Gillette & Ray, 1981*). *Proeutatus* possesses anterior teeth with ovate cross-sections as in *Euphractus*, whereas the back teeth are shaped like tear drops with the apex pointing anteriorly and lingually (*Scott, 1903*). *Euphractus* has ovate or circular occlusal surfaces on all its teeth, as in other armadillos (*Wible & Gaudin, 2004*; *Gaudin & Wible, 2006*).

In UF 248500, Mf2 possesses an oval island of osteodentine in the center of the tooth, which becomes narrow and linear in Mf3–Mf4 and Mf9. Mf5 through Mf8 have a line of osteodentine that is either Y-shaped or triangular at either end (Fig. 6B). This osteodentine pattern was consistently present among the other *Holmesina floridanus* specimens that were examined and appears in other pampatheres as well. In *Propalaehoplophorus*, each lobe of the molariforms has a branched central ridge of osteodentine, as in other glyptodonts (*Scott, 1903*; *Gillette & Ray, 1981*; *Ferigolo, 1985*; *Kalthoff, 2011*). In *Proeutatus* the posterior teeth also possess an osteodentine core like

glyptodonts and pampatheres, but this core forms a loop rather than a linear or branched structure (FMNH P13197; *Scott, 1903*). In *Euphractus* (CM 6399; UTCM 1486, 1491), as in other armadillos, there is no osteodentine in the teeth. There is only an ovate region of modified dentine in the center of each tooth (*Ferigolo, 1985*; *Gaudin & Wible, 2006*; *Kalthoff, 2011*).

Mf2 and Mf3 both have beveled crowns, with a mesial facet that is much larger than the distal facet. The angle between the mesial and distal facets on Mf2 is more acute than that of Mf1, whereas in Mf3 the two facets form nearly a right angle. Mf4 and all of the remaining teeth have but one flat occlusal surface. The long axis of the tooth crowns in UF 248500 are all angled anteroventrally in lateral view (Figs. 4 and 5). Additionally, Mf2 and Mf3 are lingually oriented in anterior view, Mf5–Mf7 are vertical, and Mf8–Mf9 are tilted labially. The corresponding occlusal surfaces form a gently rolling planar surface that faces slightly ventrolaterally in the posterior teeth, and faces progressively more ventromedially near the front of the toothrow. This is similar to the condition occurring in glyptodonts, where the upper teeth slant lingually anteriorly and labially posteriorly (*Gaudin, 2004*). The posterior molariforms take on a stairstep appearance in lateral view, with the occlusal surfaces slanting posteroventrally (Figs. 4 and 5). In ventral view, the anterior left and right toothrows bend inward to form a nearly closed dentition in both Holmesina and Vassalia (Figs. 6 and 7). This is also the case in *Kraglievichia* and (to a lesser extent) *Pampatherium* (*Simpson, 1930*; *Bordas, 1939*; *De Iuliis, Bargo & Vizcaíno, 2000*), and likely represents a derived trait of pampatheres. This feature is unusual among cingulates, but it is also present in *Macroeuphractus* (*Vizcaíno & De Iuliis, 2003*). This differs from the condition that occurs in the extinct "horned" armadillo *Peltephilus*, where the dentition is fully closed anteriorly (*Scott, 1903*; *Vizcaíno & Fariña, 1997*; *Gaudin & Wible, 2006*).

## Palatine

The palatine bone consists in part of a large horizontal process that forms the back of the hard palate, with the left and right bones separated medially by a raised suture (Figs. 6 and 7). This elongated median ridge is a synapomorphy among euphractine armadillos, *Eutatus*, *Proeutatus*, glyptodonts, and pampatheres (Node A, *Gaudin & Wible, 2006*). However, the median palatine ridge in both *Euphractus* and *Proeutatus* is more sharply defined than that of *Holmesina floridanus*. As noted above, the anterior apex of the maxillo-palatine suture in *Holmesina floridanus* (UF 248500) lies opposite the midpoint of M8. The ventral surface of the horizontal process has a few small perforations that appear to accommodate branches of the major palatine arteries, veins, and nerves. The posterior-most region of the palatal surface may have one or two minor palatine foramina of varying size (size and number vary both bilaterally and among specimens; these are identified as minor palatine foramina because their openings are directed posteriorly, toward the soft palate), and the posterior margin in some specimens is marked (on one side or both right and left) by a deep notch that presumably served the same purpose (Fig. 9), accommodating the minor palatine nerves and vessels that service the soft palate (*Wible & Gaudin, 2004*). The minor palatine foramen in UF 248500 opens into a caudal palatine foramen that is situated in the floor of the sphenopalatine canal, just medial and

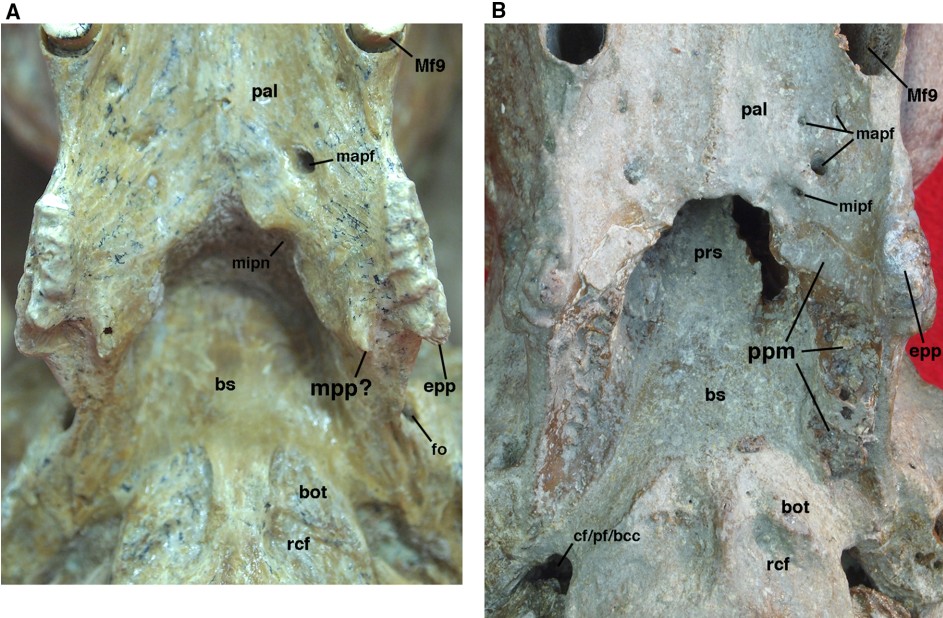

**Figure 9 Posterior palate, pterygoid processes, and choanae of *Holmesina floridanus* in ventral view.**
(A) UF 121742 (exhibit skull); (B) UF 191448. **bot**, basioccipital tuber; **bs**, basisphenoid; **cf/pf/bcc**, confluent carotid foramen, piriform fenestra and basicochlear commissure; **epp**, entopterygoid process (=hamulus or pterygoid process of other cingulates); **fo**, foramen ovale; **mapf**, major palatine foramina; **Mf9**, ninth upper molariform tooth or alveolus; **mipf**, minor palatine foramen; **mipn**, notch for minor palatine nerve and vessels; **mpp?**, neomorphic medial pterygoid process; **pal**, palatine; **ppm**, pneumatized mass of bone that may pertain to the pterygoid; **prs**, presphenoid; **rcf**, rectus capitis fossa.

anterior to the aperture of the sphenopalatine foramen. This suggests that the caudal palatine foramen accommodated both the major and minor palatine nerves and vessels, as in other xenarthrans (*Wible & Gaudin, 2004*).

The posterior edge of the palatine, which forms the anteroventral margin of the choanae, takes on a narrow U-shape. This configuration is a synapomorphy of glyptodonts and pampatheres (*Gaudin & Wible, 2006*). Moreover, the palatine extends only a short distance beyond the toothrow posteriorly, which is a synapomorphy among *Tolypeutes*, euphractine armadillos, *Eutatus*, *Proeutatus*, glyptodonts, and pampatheres (Node 5 of *Gaudin & Wible, 2006*).

In several *Holmesina floridanus* specimens examined, there was a transverse crack present behind M9 but anterior to the minor palatine foramina. Although it is more or less symmetrical on the right and left sides in UF 248500 (Fig. 6B), and a similar crack is present in roughly the same place in a couple of other specimens (UF 223813, 275496 [juvenile]), we have ultimately decided that it is just a crack in the palatine, and not a suture. The posterolateral corner of the palatine's horizontal process curves ventrally to form a large triangular flange. This flange covers the robust pterygoid process on its anterior, ventral, medial surface. In *Holmesina floridanus* (UF 248500) this flange forms distinct sutures laterally and posteriorly with the pterygoid bone.

In lateral view, there is typically no exposure of the palatine in the orbit (UF 191448, UF 121742; Fig. 8). In the juvenile specimen, UF 248500, there is a narrow portion of the

palatine's perpendicular process visible as a vertical splint lying between the maxilla anteriorly, and the alisphenoid and pterygoid posteriorly. As noted above, this may be a temporary condition, and the alisphenoid may have grown over it to cover the maxilla later in life. The dorsal edge of the palatine bone is broken in UF 248500, and the orbital sutures are fused in UF 191448. Thus the connections with the orbitosphenoid are unclear, though there is clearly no contact with the squamosal. The lack of an orbital palatine exposure is likely an autapomorphy of *Holmesina*, since an exposure is present in *Vassallia* (*De Iuliis & Edmund, 2002*), glyptodonts (*Guth, 1961*) and *Proeutatus* and *Euphractus* (*Wible & Gaudin, 2004*). The vertical process of the palatine forms the anterolateral wall of the nasopharynx, contacting the presphenoid, basisphenoid, and probably the vomer dorsally, although sutural fusion in UF 191448 and UF 121742 and damage to UF 248500 make it difficult to determine the posterior extent of this part of the palatine.

## Pterygoid

The pterygoid in cingulates is generally a small bone that forms the posteroventral margin of the orbit's medial wall, extending posteroventrally into a short pterygoid process or hamulus. It typically forms a somewhat larger portion of the posterolateral wall of the nasopharynx (*Wible & Gaudin, 2004*). Although the sutures in this region of the skull are difficult to interpret in the various specimens of *Holmesina floridanus*, it would appear the pterygoid bone occupies a similar position in this taxon. Its small, rectangular lateral surface contacts the alisphenoid dorsally and the maxilla (and perhaps the palatine) anteriorly (Figs. 4, 5 and 8). There is no contact between the pterygoid and squamosal bones, which is designated a derived feature of Cingulata by *Gaudin & Wible (2006)*, though it is likely a primitive feature of eutherian mammals (*Novacek, 1986*; *Wible, Novacek & Rougier, 2004*; *Wible et al., 2009*). Therefore, among xenarthrans, the presence of a pterygoid/squamosal contact should be considered a derived feature of pilosans instead.

The pterygoid of *Holmesina floridanus* forms a blunt, triangular, and quite rugose pterygoid process. This kind of blunt, rough, thickened pterygoid process is a synapomorphy of glyptodonts and pampatheres (*Gaudin & Wible, 2006*; albeit an ambiguous synapomorphy, due largely to the absence of preserved pterygoids in *Proeutatus* and a number of other fossil armadillos closely allied to this clade). In *Holmesina floridanus, Holmesina occidentalis* (ROM 3881; *Vizcaíno, De Iuliis & Bargo, 1998*) and *Vassallia* (FMNH P 14424; the relevant area in *Holmesina septentrionalis* is not preserved in the specimens we examined), the lateral surface of the pterygoid is covered with a variable number of rugose ridges, typically around six, which are slanted in a generally anterodorsal to posteroventral orientation. These ridges are also present in *Propalaehoplophorus* (*Scott, 1903*) although the pterygoid is much more dorsoventrally elongate in this genus. These ridges serve as an attachment point for the robust medial pterygoid muscle in these herbivorous cingulates. There are similar ridges on the lateral surface of the pterygoid of some sloths, although they are less densely packed and organized somewhat differently (*Gaudin, 2004, 2011*). The pterygoid process is positioned

lateral to the toothrow in ventral view (Figs. 6 and 7), which is also a synapomorphy of pampatheres and glyptodonts (*Gaudin & Wible, 2006*).

In ventral view, the pterygoid of UF 248500 forms an L-shaped exposure that contributes to the posterolateral corner of the hard palate, with a narrow portion comprising the pterygoid process/lateral exposure of the pterygoid extending anteroposteriorly, and a narrow transverse portion that extends medially (Fig. 6B). A similar morphology is probably present in UF 121742, though the sutures are not always clear, whereas in some specimens (e.g., UF 191448, UF 223813, UF 275496) there is no evidence of a suture between the pterygoid process and palatine, though we suspect that this is the result of fusion. A palatal exposure of the pterygoid is an unusual feature among cingulates (and among placental mammals in general; *O'Leary et al., 2013*), but is a synapomorphy of the dasypodine armadillos *Dasypus* and *Stegotherium* (*Gaudin & Wible, 2006*). At least the pterygoid process contribution to the palate may be more widespread among pampatheres and glyptodonts. Though it is not mentioned in *De Iuliis & Edmund (2002)*, such a contribution is visible in *Vassallia* (FMNH P14424), and Guth illustrates a similar morphology in *Glyptodon* (*Guth, 1961*, fig. 123).

The dorsal portion of the pterygoid in UF 248500, which normally forms much of the posterolateral wall of the nasopharynx in cingulates (*Wible & Gaudin, 2004*), is strongly reduced, extending dorsally as a triangular wedge only a short distance. In UF 121742, the dorsal and medial exposure of the pterygoid appears larger, but still does not reach the roof of the nasopharynx. Because of suture closure, it is unclear whether the area dorsal to the pterygoid in the latter specimen is formed by palatine extending posterodorsally, or basisphenoid extending ventrally.

We have observed several unusual morphologies associated with the pterygoid region in individual specimens of *Holmesina floridanus*. UF 121742 possesses two pterygoid processes—a large, more laterally situated process that is clearly homologous to the pterygoid process of the other *Holmesina floridanus* specimens and other cingulates, and a smaller, more medially situated process extending posteriorly from the back margin of the hard palate (Fig. 9A). The presence of two pterygoid processes or crests, an entopterygoid process/crest and an ectopterygoid process/crest, is a feature that is widely observed among primitive eutherians [e.g., *Zalambdalestes* (*Wible, Novacek & Rougier, 2004*); *Lepticitis* (*Novacek, 1986*)] and many extant placental mammals [e.g., *Atelerix* (UTCM 727, 1553; *Frost, Wozencraft & Hoffmann, 1991*); *Tupaia* (UTCM 1980; *Wible, 2011*); *Elephantulus* (UTCM 1482, 1512)]. The ectopterygoid process/crest is typically formed mostly by the alisphenoid, so for those taxa with a single pterygoid process or hamulus formed by the pterygoid, it is generally homologized with the entopterygoid process/crest, as has been done for the armadillo *Euphractus* by *Wible & Gaudin (2004)*. If the lateral pterygoid process of UF 121742 is indeed the entopterygoid process, as seems almost certain, the more medial process represents a neomorph. We suspect this process represents an attachment point for enlarged pharyngeal or masticatory muscles. If the muscular anatomy of *Canis* (*Evans & Christiansen, 1979*) can be used as a model, the pterygopharyngeus seems a likely candidate.

In UF 191448, there is an unusual, vertical mass of cancellous pneumatized bone that lies at the junction between the medial wall of the orbit and the lateral wall of the choanae. This mass may be part of the pterygoid, due to its position in the skull, and the fact that it has a small palatal exposure along the posterior margin of the palate that appears to match the medial, transverse portion of the pterygoid palatal exposure in UF 248500 and other *Holmesina floridanus* specimens (Fig. 9B). On the other hand, this mass appears to be completely surrounded by sutures (including the palatal exposure), which would suggest that it too is a neomorphic feature. Pneumatization of the pterygoid is rare among cingulates. However, it is commonplace among pilosans, where inflated, often bullate pterygoids are known among myrmecophagid anteaters, *Megalocnus, Mylodon*, some nothrotheriid sloths, the three-toed sloth *Bradypus torquatus*, and the two-toed sloth genus *Choloepus* (*Stock, 1925*; *Guth, 1961*; *Patterson et al., 1992*; *Gaudin, 2004*; *De Iuliis, Gaudin & Vicars, 2011*). This separate pneumatized mass of bone is only present in UF 191448, but other *Holmesina floridanus* specimens did display pneumatized bone around the posteromedial edge of the choanae. This mass of bone in UF 191448 forms a discrete suture with the palatine and basisphenoid anteriorly and dorsally, the palatine anteriorly and ventrally, and the pterygoid and alisphenoid bones laterally.

## Lacrimal

The lacrimal is shaped roughly like a parallelogram, with its long axis tilted anterodorsally (Figs. 4, 5, and 8). It contacts the maxilla anteriorly and posteroventrally, the frontal posterodorsally, and the jugal ventrally. The lacrimal consists of a facial and orbital process; the boundary between these two processes is not particularly distinct. In *Wible & Gaudin (2004)*, the low ridge that runs from the postorbital process of the frontal ventrally onto the jugal, the antorbital ridge, was used as a rough boundary between the facial and orbital processes. The antorbital ridge exhibits some variation in its development among *Holmesina floridanus* specimens. The position of the lacrimal foramen also varies among pampatheres. In the majority of pampathere specimens examined in this study, the lacrimal foramen is located on the antorbital ridge, that is, on the boundary between the facial and orbital processes, as it is in *Proeutatus* (FMNH P13199) and *Euphractus* (*Wible & Gaudin, 2004*). In *Holmesina septentrionalis* (UF 889, 243224) and *Vassallia* (P 14424), however, the lacrimal foramen is located anterior to the antorbital ridge; therefore, it is clearly situated on the facial process. This is apparently also the condition in primitive glyptodonts (*Scott, 1903*). In *Euphractus, Proeutatus*, and most of the pampatheres and glyptodonts examined, the lacrimal foramen is relatively small. However, in *Holmesina septentrionalis* (UF 889) the lacrimal foramen is situated within a much larger, circular depression. A similar, but more dorsoventrally ovate depression appears to be present in *Holmesina septentrionalis* (UF 243224), as well as in *Propalaehoplophorus* (YPM VPPU 15007), although in this specimen the depression opens posteriorly. The lacrimal foramen transmits the nasolacrimal duct from the eye to the nasal cavity (*Wible & Gaudin, 2004*). Just dorsal to the lacrimal foramen is a rugose area, the lacrimal tubercle (Figs. 4, 5 and 8). In UF 191448, the tubercle is small, and continuous with a crest that extends ventrally onto the zygoma anterior to the lacrimal foramen (*Wible & Gaudin,*

*2004*). The lacrimal tubercle is much larger in UF 248500, and contacts not only this anterior crest, but the antorbital ridge as well. A lacrimal tubercle is present in all cingulates, with the exception of *Dasypus* and *Stegotherium* (*Gaudin & Wible, 2006*), and is distinct from the rest of the lacrimal surface, which is generally smooth.

The facial process of the lacrimal bone in *Holmesina floridanus* (UF 191448, et al.), and other pampatheres (*Holmesina occidentalis*; *Vassallia*), is typically triangular in shape (Figs. 4, 5 and 8). The shape is more variable in *Holmesina septentrionalis*. In UF 889, it is triangular as in other pampatheres, but the anterodorsal apex of the triangle is elongated with a rounded tip, whereas in UF 234224 the facial process is more ovate than triangular, elongated dorsoventrally. *Euphractus* has a quadrangular facial process (*Wible & Gaudin, 2004*; *Gaudin & Wible, 2006*). According to *Gaudin & Wible (2006)*, a quadrangular facial process is a synapomorphy of the clade including *Eutatus*, euphractine armadillos, *Proeutatus*, glyptodonts, and pampatheres (Node B of *Gaudin & Wible, 2006*), although the latter revert to the triangular shape characteristic of dasypodine and tolypeutine armadillos.

The orbital process of the lacrimal bone in *Holmesina floridanus* is also triangular, but it is somewhat smaller than the facial process (Figs. 4, 5 and 8). The lacrimal contributes to a small portion of the anterior orbital wall, where it contacts the jugal anterolaterally, and the frontal posteriorly. There is also a small lacrimal contact with the maxilla posteroventrally, on the orbital side of the jugal in *Holmesina floridanus* (UF 191448, 248500), as in *Euphractus* (UTCM 1486, 1491; *Wible & Gaudin, 2004*). This trait, the presence of lacrimal contact with the orbital process of the maxilla, is a synapomorphy of *Tolypeutes*, *Eutatus*, euphractine armadillos, *Proeutatus*, pampatheres, and glyptodonts (Node 4 of *Gaudin & Wible, 2006*). The lacrimal fenestra, which perforates the lower edge of the orbital process of the lacrimal, serves as the site of origin for the inferior oblique muscle, and is present at the intersection of the lacrimal, frontal, and maxilla in *Holmesina floridanus* (*Gaudin & Wible, 2006*; *Wible & Gaudin, 2004*). This condition is primitive, and occurs in all cingulates with the exception of *Dasypus, Stegotherium, Zaedyus,* and *Chlamyphorus* (*Gaudin & Wible, 2006*).

## Jugal

The jugal forms the anterior portion of the zygomatic arch. In *Holmesina floridanus* (UF 248500, UF 191448) the dorsal edge of the jugal is U-shaped, whereas the ventral edge is irregular (Figs. 4 and 5). The jugal can be divided into two processes, facial and zygomatic. Roughly half of the anterior root of the zygoma is comprised of the transversely broad facial process of the jugal bone, which contacts the lacrimal dorsally, the maxilla anteriorly, ventrally, and medially. The zygomatic process is oriented almost perpendicular to the facial process, and is strongly compressed mediolaterally and deep dorsoventrally. It has a dorsoventrally convex surface laterally, and is concave medially. In lateral view it broadens posteriorly toward its posterior contact with the squamosal, near the middle of the zygomatic arch. The jugal–squamosal suture in UF 248500 is asymmetrically concave posteriorly, with the anterodorsally oriented ventral portion more elongate than posterodorsally sloped dorsal portion (Figs. 4B and 4C). In UF 191448, the

junction between these dorsal and ventral portions is more angular (Fig. 4A). In UF 248500, the posterodorsal edge of the zygomatic process is extended into a sharp, triangular postorbital process. In UF 191448, the postorbital process is more rounded, and formed jointly by the jugal and squamosal. The jugal/squamosal contact in *Holmesina occidentalis* (ROM 3881) and *Vassallia* (*De Iuliis & Edmund, 2002*) shows a similar pattern, though in the latter the postorbital process is carried largely by the squamosal rather than the jugal. In contrast to the pampathere condition, in both *Propalaehoplophorus* and *Proeutatus* (*Scott, 1903*) there is a substantial posterior extension of the jugal underneath the zygomatic process of the squamosal, so that much of the jugal/squamosal suture is horizontal, as in euphractine armadillos (*Wetzel, 1985*; *Wible & Gaudin, 2004*). The postorbital process on the zygomatic arch is also less well developed in *Euphractus* (but not *Chaetophractus* or *Zaedyus*; *Wetzel 1985*; *Wible & Gaudin 2004*), *Proeutatus* (FMNH 13197; *Scott, 1903*), and some specimens of *Propalaehoplophorus* (e.g., FMNH P13205, *Propalaehoplophorus* sp.; *Propalaehoplophorus australis*, *Scott 1903* plate 23; but not YPM VPPU 15007, *Propalaehoplophorus australis*, or *Propalaehoplophorus minor*, *Scott 1903* plate 27).

The facial process extends ventrally and slightly laterally into a prominent ventral (or descending) process of the zygomatic arch. This ventral process is in fact an anteromedial to posterolaterally extended, cresecent-shaped complex, comprised of a variable number of strong rugose bumps or transverse ridges. In UF 248500, there are only two bumps/ridges (Figs. 4B, 5 and 7), with the more anterior being formed in part by the jugal and in part by the maxilla. In other specimens, there may be as many as four (e.g., UF 275498, 285000 on L only). In some specimens, this ventral zygomatic process (or complex of processes) appears worn, although it is unclear if this is reflective of the age of the specimen (they do seem less "worn" in juvenile specimens) or due to some sort of post-mortem abrasion.

*Holmesina occidentalis* and *Vassallia* have ventral zygomatic processes quite similar to those in *Holmesina floridanus*, with three bumps or ridges that are heavily worn in the *Vassallia* specimen [FMNH P14424; *De Iuliis & Edmund (2002)*, who also report a similar morphology in *Pampatherium*; *Vizcaíno, De Iuliis & Bargo (1998*, pp. 297–298*)* note that the ventral process is "narrower and less rugose" in *Holmesina occidentalis* than in *Vassallia*]. The ventral zygomatic process of pampatheres is comparable in position to the small boss present in *Euphractus* (*Wible & Gaudin, 2004*; *Gaudin & Wible, 2006*) and *Proeutatus* (FMNH P13197; *Gaudin & Wible, 2006*), but is much larger in size. *Propalaehoplophorus* and other glyptodonts possess a gigantic descending process (*Hoffstetter, 1958*; *Gaudin & Wible, 2006*) that forms a greatly elongated, anteroposteriorly compressed plate of bone, but unlike *Holmesina,* this process is primarily formed by the maxilla (YPM VPPU 15007; *Gillette & Ray, 1981*), the jugal forming only a small portion of the dorsolateral margin. This descending process is greatly enlarged in order to accommodate the bulky masseter muscle in glyptodonts (*Gillette & Ray, 1981*), and this is likely the case in pampatheres, though the masseter would have been enlarged to a lesser degree.

## Frontal

The frontal bone in *Holmesina floridanus* forms slightly less than a third of the total skull length, including the anterior half of the braincase. It is shaped roughly like a pentagon in dorsal view, broadening dramatically in its anterior reaches (Figs. 2 and 3). This is due to the presence of enlarged sinuses beneath the frontal bone, a feature present in many other cingulates (*Gaudin & Wible, 2006*; *Billet et al., 2017*). The median interfrontal suture is fused in all the adult and subadult specimens of *Holmesina floridanus*, with the sole exception of the youngest specimen, UF 275496. The interfrontal suture is also fused at least posteriorly in *Holmesina occidentalis* (ROM 3881), and along its entire length in *Vassallia* (FMNH P14424) and *Propalaehoplophorus* (YPM-VPPU 15007; although apparently not in *Scott's (1903)* illustration of *Propalaehoplophorus australis*, pl. XXIII, fig. 3), but not in *Holmesina septentrionalis* (UF 889), *Proeutatus* (FMNH P13197), or *Euphractus* (UTCM 1486, 1491; *Wible & Gaudin, 2004*). The frontal bone contacts the nasal, maxilla, and lacrimal bones anteriorly and the parietal posteriorly on the skull roof. It dips ventrally and laterally into the orbit to form a sizeable portion of the medial orbital wall (Figs. 4, 5 and 8). The orbital portion of the frontal likely contacts the maxilla, orbitosphenoid and alisphenoid ventrally, and the squamosal posteroventrally, creating a triangular exposure in lateral view that is similar to that of *Euphractus* (UTCM 1491; *Wible & Gaudin, 2004*). The fronto–parietal suture is a very irregular and jagged line that travels slightly anterodorsally across the top of the braincase from a position even with the anterior edge of the glenoid fossa, as in *Proeuphractus*, *Proeutatus*, other pampatheres, and glyptodonts (Node E of *Gaudin & Wible, 2006*). This differs from *Euphractus* in which the most lateral part of fronto-parietal suture lies posterior to the anterior edge of the glenoid fossa (*Wible & Gaudin, 2004*; *Gaudin & Wible, 2006*).

The frontal bone in *Holmesina floridanus* has very distinct temporal lines curving postermedially from the large, blunt postorbital processes (Figs. 2 and 3). The posterior half of the fused interfrontal suture is elevated by a prominent midline crest in UF 248500, that extends unbroken between the temporal lines back along the midline of the parietal, all the way to the nuchal crest. A ridge of similar extent is present in UF 191448, but it is much more weakly developed. *Wible & Gaudin (2004)* describe a weakly developed crest in a similar position in *Euphractus*, where it serves as a site of origin for the orbito-auricularis muscle. The crest is also present in *Holmesina occidentalis* and *Proeutatus* (FMNH P1319; *Scott, 1903*). It is present on the frontal only in *Holmesina septentrionalis* (UF 889) and *Vassallia* (FMNH P14424), being replaced posteriorly by a true sagittal crest. It is missing entirely in *Propalaehoplophorus*, where again there is a strong sagittal crest (FMNH P13205; YPM VPPU 15007; *Scott, 1903*). It is likely that the presence of a strong ridge in this position is related to the presence of large pinnae for the ears.

As is typical of euphractine armadillos, *Proeutatus*, pampatheres and glyptodonts, there are numerous small nutritive foramina in UF 191448 that coalesce around the midline of the frontal dorsally, just anterior to the frontal–parietal suture, in a depression between the temporal lines and behind the frontal sinuses (Node C of *Gaudin & Wible, 2006*).

These foramina are less evident in UF 248500. In lateral view, within the temporal fossa, there are also foramina along the posterolateral region of the frontal bone in eutatine armadillos, euphractine armadillos, *Proeutatus*, *Vassallia* and glyptodonts (Node A of *Gaudin & Wible, 2006*). These appear to be absent in *Holmesina floridanus*, though they are present in *Holmesina occidentalis* (ROM 3881).

In addition to these foramina, the frontal is marked by two other types of foramina within the orbit (Fig. 8). UF 191448 has a pair of asymmetrical foramina for the frontal diploic vein (sensu *Wible & Gaudin, 2004*; =supraorbital foramina of *Gaudin (2004)* and others). On the left, there is a single opening situated ventral and posterior to the broad, low, rugose area that marks the postorbital process. On the right, there are two foramina, one mirroring the opening on the left, the other, smaller opening situated further anterior and dorsal, virtually on the process itself. The left side of UF 248500 is damaged in the region of the postorbital process, but the right side has a single foramen like that described for UF 191448. In UF 121742, the foramen is more anteriorly situated, lying in front of a strong infratemporal crest that extends posteroventrally from the postorbital process, a crest that is only weakly developed in UF 191448. The morphology of UF 121742 is also found in *Holmesina septentrionalis* (UF 889, UF 234224) and *Vassallia* (UF P14424). The foramen for the frontal diploic vein also occurs in a similar position in glyptodonts (*Gaudin, 2004*), whereas in *Proeutatus* it is situated more posteriorly (FMNH P13197).

The ventral portion of the orbital wing in UF 121742 is marked by a ventrally directed foramen that lies between the infratemporal crest and a rounded ridge that marks the dorsal margin of the optic foramen (Fig. 8). Given the position of this opening, anterodorsal to the optic foramen, and its connection to an anteriorly directed canal, we identify it as the ethmoid foramen (sensu *Wible & Gaudin, 2004*; transmits the ethmoidal nerve and vessels). Other cingulates may have as many as three ethmoid foramina (*Gaudin & Wible, 2006*). Although sutures are not unambiguous in this area, the opening appears to be contained entirely within the frontal, in contrast to some cingulates in which there is orbitosphenoid participation in the rim (*Gaudin & Wible, 2006*). In UF 248500, there appears to be a second ethmoid foramen, just dorsal to the first and separated from it by the infratemporal crest.

## Parietal

The parietal bone is roughly rectangular and forms the posterior half of the braincase (Figs. 2–5). It contacts the frontal anteriorly, the squamosal ventrolaterally, and the supraoccipital posteriorly. As in most cingulates, with the exception of *Peltephilus*, there is no contact between the parietal and the alisphenoid bones (Fig. 8) due to an extensive contact between the frontal and squamosal bones (Node 2 of *Gaudin & Wible, 2006*; see also *Novacek & Wyss, 1986*; *Rose & Emry, 1993*; *Gaudin et al., 1996*). Although the parietal tends to be relatively flat transversely in eutatine and euphractine armadillos and in glyptodonts (*Gaudin & Wible, 2006*), in *Holmesina floridanus* and other pampatheres (*Cartelle & Bohórquez, 1985*; *De Iuliis, Bargo & Vizcaíno, 2000*; *De Iuliis & Edmund, 2002*) it is strongly convex transversely, giving the braincase a much more tubular appearance. The parietals are marked by strong temporal lines, which approach one another, but do

not unite to form a midline sagittal crest. As noted above in the description of the frontal, the parietals do carry a midline crest for the extrinsic ear muscles between the temporal lines. This morphology, which also characterizes *Holmesina occidentalis* (ROM 3881), is very reminiscent of the pattern in *Proeutatus* (FMNH 13197; *Scott, 1903*) and some specimens of *Euphractus* (*Wible & Gaudin, 2004*). A true sagittal crest is present on the parietals in *Holmesina septentrionalis* (UF 889), *Vassallia* (*De Iuliis & Edmund, 2002*), and *Propalaehoplophorus* (*Scott, 1903*) and other glyptodonts (*Gillette & Ray, 1981*), as noted above. Both the temporal lines and the midline crest unite posteriorly with a robust nuchal crest. The nuchal crest is of uniform thickness along the posterior edge of the skull, as is characteristic of *Tolypeutes*, eutatine armadillos, euphractine armadillos, *Proeutatus*, pampatheres, and glyptodonts (Node 5 of *Gaudin & Wible, 2006*). It is strongly convex posteriorly, overhanging the dorsal potion of the occiput.

Within the temporal fossa in *Holmesina floridanus,* the parietal surface is heavily pitted with a large but variable number of foramina (12–16 in UF 191448 and 248500), especially in the ventrolateral half of the bone. The more dorsally located foramina open into distinct grooves, traveling at various angles, through which the rami temporales emerge. The presence of so many temporal foramina (greater than five) is a synapomorphy of *Priodontes*, *Tolypeutes*, eutatine armadillos, euphractine armadillos, *Proeutatus*, pampatheres, and glyptodonts (Node 3 of *Gaudin & Wible, 2006*).

## Squamosal

The squamosal consists of two broad regions, the squamous part and the zygomatic process (Figs. 2–8). The squamous part comprises a roughly rectangular, vertical exposure in the lateral wall of the braincase, contacting the frontal anteriorly, the alisphenoid and petrosal ventrally, and the parietal dorsally. It also has a lappet that wraps around the nuchal crest to form a small, triangular exposure on the occiput, contacting the occipital exposure of the mastoid petrosal ventrally, and the supraoccipital dorsally. *Euphractus* has a very similar occipital exposure of the squamosal (*Wible & Gaudin, 2004*), and according to *Gaudin & Wible (2006)*, this feature is a synapomorphy of euphractine aramadillos, pampatheres and glyptodonts (Node C of *Gaudin & Wible, 2006*). The anterior portions of the squamosal/parietal suture and the dorsal portions of the squamosal/frontal suture form a slightly raised ridge, as they do in some *Euphractus* (*Wible & Gaudin, 2004*) and in *Vassallia* (*De Iuliis & Edmund, 2002*) and *Propalaehoplophorus* (YPM VPPU 15007). Like the parietal, the squamous region's posterior and dorsal surface is marked by a variable number (5–12 in UF 191448 and UF 248500) of foramina for the rami temporales. This is a common feature in cingulates. The squamous part of the squamosal is crossed horizontally by a crest that connects the dorsal edge of the zygomatic process to the nuchal crest, marking the lower limit of the temporal fossa (Figs. 4 and 5). This is also a feature in *Holmesina occidentalis* (ROM 3881) and *Holmesina septentrionalis* (UF 234224), as well as *Proeutatus* (FMNH P13197) and *Propalaehoplophorus* (YPM VPPU 15007; *Scott, 1903*), whereas in *Vassallia* (*De Iuliis & Edmund, 2002*) the ventral end of the nuchal crest passes lateral to the crest extending posteriorly from the dorsal edge of the zygoma,

so that the two approach but do not contact. The latter is similar to the condition in *Euphractus* (*Wible & Gaudin, 2004*).

The region of the squamosal immediately ventral to the lower ridge of the temporal fossa is strongly convex anteroposteriorly, forming a porus acousticus that would have accommodated the external auditory meatus. The posterior wall of the porus is formed by a flat, roughly ovate ventral projection that abuts the anterior base of the petrosal's paroccipital process (=mastoid process of *Patterson, Segall & Turnbull (1989)* and *Gaudin (1995)*). This projection is the post-tympanic process of the squamosal (Fig. 7). In UF 191448, it has a somewhat thickened ventral edge that may have participated in the facet for the posterior crus of the ectotympanic. The lower anterior wall of the porus is formed by a freestanding ridge, the postglenoid process (Fig. 10). As in *Euphractus* (*Wible & Gaudin, 2004*) and a few other eutherians (e.g., *Zalambdalestes*, *Wible, Novacek & Rougier 2004*), the postglenoid process lies posterior to the postglenoid foramen in *Holmesina floridanus*. The postglenoid process and post-tympanic process approach one another medially at roughly a 60°–75° angle in ventral view (it is more acute in UF 248500 than in UF 191448), with the porus narrowing accordingly (Figs. 7 and 10). The morphology of this region of the skull in *Vassallia* (FMNH P14424) is very similar to that of *Holmesina floridanus*. *Propalaehoplophorus* (YPM VPPU 15007) is also similar, though the porus is narrower, with a much more acute angle (<20°) between the postglenoid and post-tympanic process, and the former is much larger that it is in *Holmesina floridanus*. *Proeutatus* (FMNH P13197) has a very odd morphology in this region of the skull. The glenoid is situated so far posteriorly that it approaches the ventral end of the nuchal crest. As a consequence, the porus acousticus is reduced to a narrow vertical groove, and the process identified as the post-tympanic process by *Patterson, Segall & Turnbull (1989)* is actually two processes, the post-tympanic process and the immediately adjacent postglenoid process (the two distinct tips of these processes are visible in the lateral view of the skull in *Patterson, Segall & Turnbull (1989*, fig. 13A*)). In *Holmesina floridanus* 191448, there is a bilateral foramen just lateral and dorsal to the anterior end of the postglenoid process (Figs. 4A and 5). This is likely homologous to the suprameatal foramen found in some *Euphractus* specimens (*Wible & Gaudin, 2004*; it transmits a ramus temporalis of the stapedial artery system). Like *Euphractus*, the presence of this foramen may be variable in *Holmesina*, because it is absent in UF 248500 and UF 121742.

Anterior and medial to the postglenoid process is a small, circular depression that represents the squamosal contribution to the epitympanic recess, accommodating the mallear/incudal articulation (Fig. 10). The squamosal forms roughly 2/3 of this depression, the remainder formed by the lateral reaches of the petrosal. In *Propalaehoplophorus* (YPM VPPU 15007), the epitympanic recess is ovate rather than circular, elongated along a posterolateral to anteromedial axis.

Anterior and medial to the epitympanic recess is a massive process that extends as a broad ridge laterally and ventrally, forming the anterolateral wall to the tympanic cavity. This is the entoglenoid process, which extends across the squamosal/alisphenoid suture and onto the alisphenoid behind the foramen ovale (Fig. 10). The posterior surface of this process is marked by a circular depression that almost certainly represents the facet for the

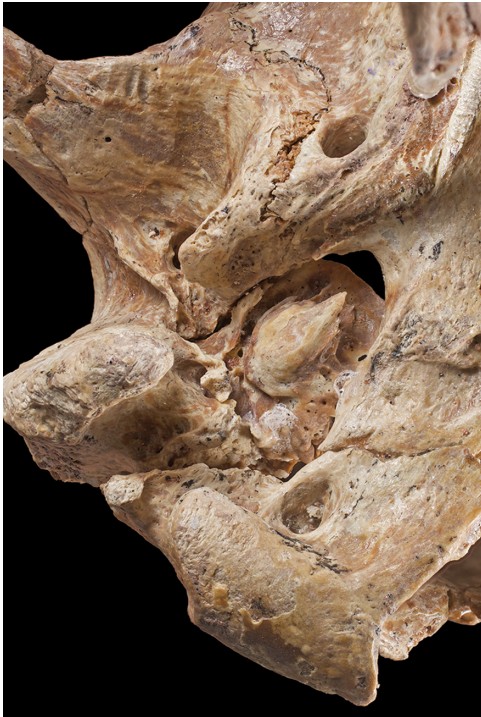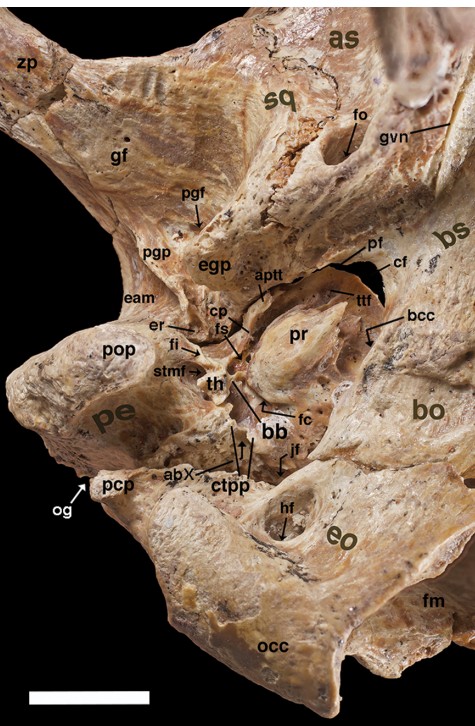

**Figure 10 Stereophotographs of right auditory region of *Holmesina floridanus* (UF 248500) in ventral view. abX**, groove for auricular branch of vagus nerve (c.n. X); **aptt**, anteroventral process of tegmen tympani (=processus crista facialis); **as**, alisphenoid; **bb**, bony bridge between tympanohyal and crista interfenestralis; **bcc**, basicochlear commissure; **bo**, basioccipital; **bs**, basisphenoid; **cf**, carotid foramen; **ci**, crista interfenestralis; **cp**, crista parotica; **ctpp**, caudal tympanic process of petrosal; **eam**, external auditory meatus; **egp**, entoglenoid process; **eo**, exoccipital; **er**, epitympanic recess; **fc**, fenestra cochleae; **fi**, ridge immediately ventral to fossa incudis; **fm**, foramen magnum; **fo**, foramen ovale; **fs**, facial sulcus; **gf**, glenoid fossa; **gvn**, groove for vidian nerve; **hf**, hypoglossal foramen; **jf**, jugular foramen; **occ**, occipital condyle; **og**, groove for occipital artery; **pcp**, paracondylar process of exoccipital (=paroccipital process of *Patterson, Segall & Turnbull (1989)*); **pe**, petrosal; **pf**, piriform fenestra; **pgf**, postglenoid foramen; **pgp**, postglenoid process; **pop**, paroccipital process of petrosal (=mastoid process of *Patterson, Segall & Turnbull (1989)*); **pr**, promontorium of petrosal; **sq**, squamosal; **stmf**, stylomastoid foramen; **th**, tympanohyal; **ttf**, tensor tympani fossa on epitympanic wing of petrosal; **zp**, zygomatic process of squamosal. Scale bar = 1 cm. Photos by S. Chatzimanolis and T. Gaudin.

anterior crus of the ectotympanic. At its posterior, dorsal and medial extremity, the entoglenoid process abuts the small anteroventral process of the tegmen tympani on the petrosal (=processus crista facialis of *Patterson, Segall & Turnbull (1989)*, *Gaudin (1995)*, *Wible & Gaudin (2004)*), which may have a small contribution to the ectotympanic facet. A similar entoglenoid process is present in *Vassallia* (FMNH P14424), *Propalaehoplophorus* (YPM VPPU 15007), and *Proeutatus* (FMNH P13197). In the latter two, it appears to be somewhat inflated.

In ventral view, the root of the zygomatic process is triangular, extending (and narrowing) laterally, as it does in all pampatheres and glyptodonts (Node 8 of *Gaudin & Wible, 2006*). Its dorsal surface is deeply concave transversely to house the temporalis muscles. On its ventral surface, it carries the glenoid articulation for the mandible. Just beyond the lateral edge of the glenoid, the process curves anteriorly in a graceful arc. In

lateral view, it deepens considerably in a dorsoventral plane as it approaches its anterior contact with the jugal, with which it forms the zygomatic arch. As noted above, it may or may not contribute to the postorbital process on the zygoma. *Propalaehoplophorus* (*Scott, 1903*), *Proeutatus* (*Scott, 1903*), and *Euphractus* (*Wible & Gaudin, 2004*) and other euphractine armadillos (*Wetzel 1985*) all lack the anterior broadening of the zygomatic process seen in pampatheres (*Cartelle & Bohórquez, 1985*; *De Iuliis, Bargo & Vizcaíno, 2000*; *De Iuliis & Edmund, 2002*). Like the zygomatic portions of the jugal, the zygomatic region of the squamosal is convex laterally and concave medially.

The glenoid articular surface on the ventral side of the zygomatic root is convex in both transverse and anteroposterior planes, as it is in most eutatine and euphractine armadillos and in glyptodonts and other pampatheres (Node B of Gaudin & Wible, 2004; *Vizcaíno, De Iuliis & Bargo, 1998*). Its shape is somewhat more unusual however, forming a rounded triangle, narrowing laterally, with its transverse width much greater than its anteroposterior length (Tables 1 and 2). Glyptodonts show similar transverse extension of the glenoid, though the shape of the facet is generally more rectangular and even narrower anteroposteriorly (*Scott, 1903*; *Gaudin, 2004*; *Gaudin & Wible, 2006*), whereas in eutatine and euphractine armadillos the glenoid is more U-shaped, and as long or longer in the anteroposterior as opposed to the transverse dimension (*Gaudin & Wible, 2006*). As in other cingulates, there is a postglenoid foramen in *Holmesina floridanus*. Like other pampatheres and glytopdonts, this foramen is clearly visible in ventral view (Figs. 7 and 10), because the external auditory meatus is positioned well behind the glenoid, exposing the postglenoid fossa in which the foramen is situated (*Gaudin & Wible, 2006*). In euphractine and eutatine armadillos this area tends to be obscured by the nearby ectotympanic. The postglenoid foramen transmits the capsuloparietal emissary vein in *Euphractus* (*Wible & Gaudin, 2004*).

## Petrosal

The petrosal bone is preserved in situ in UF 191448 and on the right side of UF 248500 (Figs. 6, 7 and 10), whereas a nearly complete, isolated left petrosal is available in the latter specimen (Figs. 11 and 12). This will allow us to describe in detail not only the ventral exposure of the bone, but also its dorsal and lateral surfaces. The petrosal, which houses the inner ear, is bordered by the squamosal laterally, the exoccipital posteromedially, the supraoccipital dorsally, and the basioccipital and basisphenoid medially. It is comprised of two primary regions, the pars canicularis housing the semicircular canals and vestibular apparatus, and the pars cochlearis housing the cochlea (*MacIntyre, 1972*). In ventral and lateral view, these are represented most notably by the mastoid region and promontorium, respectively.

The promontorium of *Holmesina floridanus* is globose, and lacks any clear grooves for arteries or nerves, as is typical for cingulates (*Guth, 1961*; *Bugge, 1979*; *Patterson, Segall & Turnbull, 1989*; *Wible & Gaudin, 2004*; *Wible, 2010*). At its anterior pole, it is marked by a distinctive, elongate triangular process, the rostral tympanic process of the petrosal (Figs. 7, 10, 11 and 12). *Wible (2010)* describes a small blunt rostral tympanic process that is present to a varying degree in some *Dasypus novemcinctus*, but no such process is

Gaudin and Lyon (2017), *PeerJ*, DOI 10.7717/peerj.4022                                                                33/73

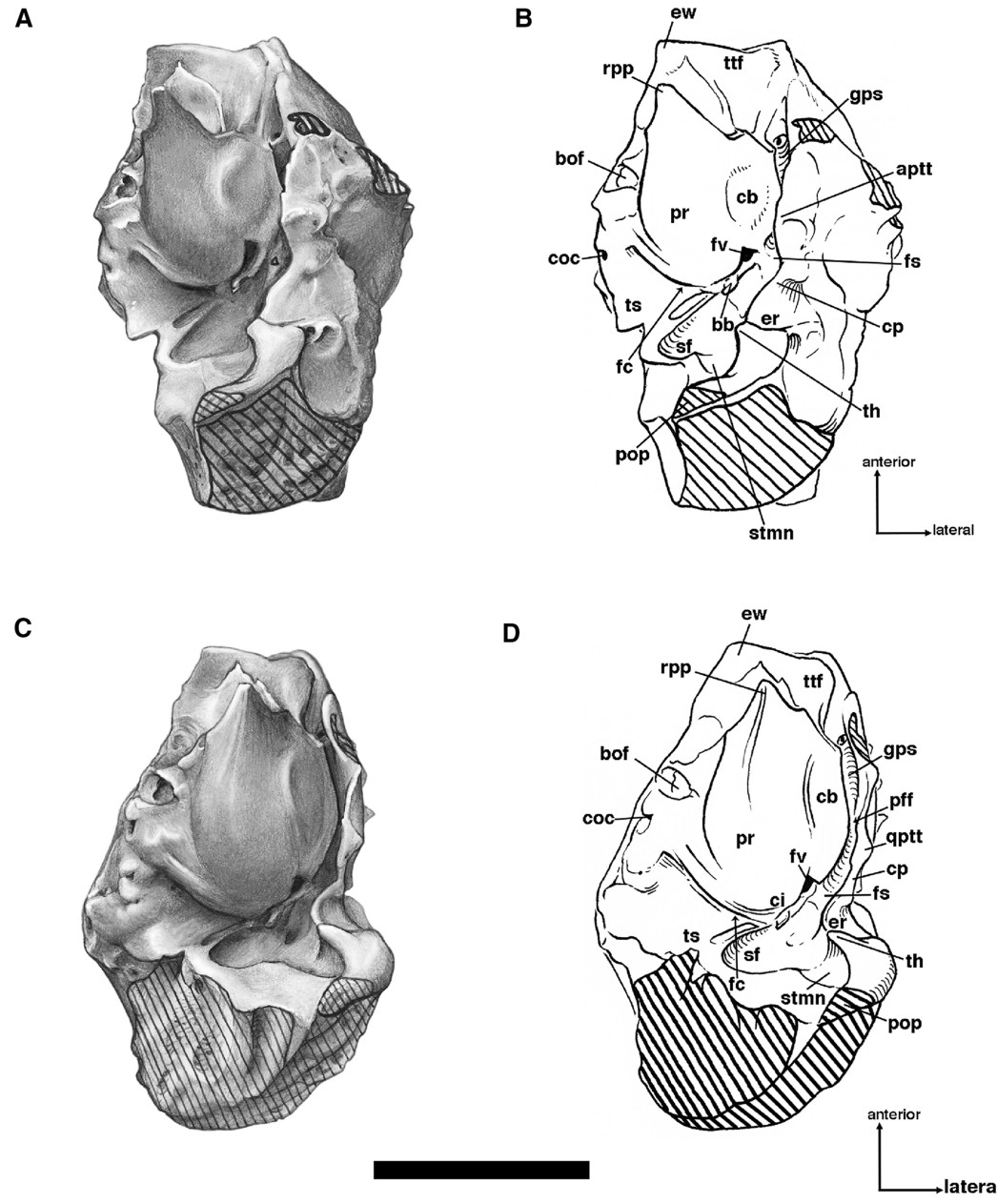

**Figure 11 Isolated left petrosal of *Holmesina floridanus* (UF 248500) in (A, B) ventrolateral; (C, D) ventral; (E, F) lateral. aptt**, anteroventral process of tegmen tympani (= processus crista facialis); **bb**, bony bridge between tympanohyal and crista interfenestralis; **bof**, basioccipital facet; **cb**, circular boss on ventrolateral surface of promontorium; **ci**, crista interfenestralis; **coc**, cochlear canaliculus; **cp**, crista parotica; **er**, epitympanic recess; **ew**, epitympanic wing; **fc**, fenestra cochleae; **fs**, facial sulcus; **fv**, fenestra vestibuli; **gps**, sulcus for greater petrosal nerve; **pff**, primary facial foramen; **pop**, paroccipital process of petrosal (= mastoid process of *Patterson, Segall & Turnbull (1989)*); **pr**, promontorium of petrosal; **rpp**, rostral process of petrosal; **sf**, stapedius fossa; **stmn**, stylomastoid notch; **th**, tympanohyal; **ts**, triangular shelf (= roof of post-promontorial sinus). Scale bar = 1 cm.

present in *Euphractus* (*Wible & Gaudin, 2004*). In *Vassallia*, *Propalaehoplophorus* and other glyptodonts, and *Proeutatus* (*Guth, 1961*; *Patterson, Segall & Turnbull, 1989*), the entire promontorium is elongated anteromedially, giving the promontorium a teardrop

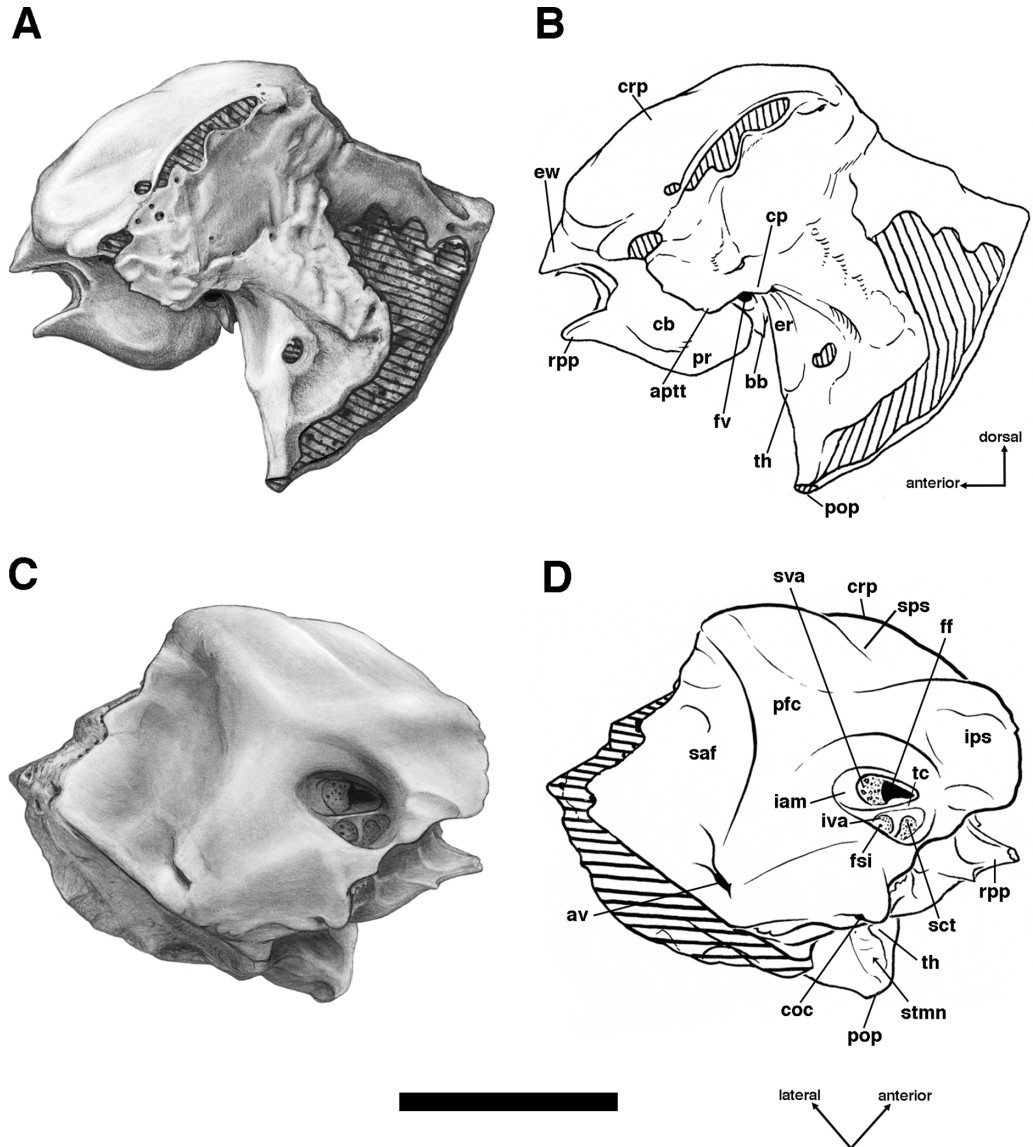

**Figure 12 Isolated left petrosal of *Holmesina floridanus* (UF 248500) in (A, B) lateral; and (C, D) medial views.** **aptt**, anteroventral process of tegmen tympani (= processus crista facialis); **av**, aqueductus vestibuli; **bb**, bony bridge between tympanohyal and crista interfenestralis; **cb**, circular boss on ventrolateral surface of promontorium; **coc**, cochlear canaliculus; **cp**, crista parotica; **crp**, crista petrosa; **er**, epitympanic recess; **ew**, epitympanic wing; **fsi**, foramen singulare; **fv**, fenestra vestibuli; **iam**, internal acoustic meatus; **ips**; fossa/groove for inferior petrosal sinus; **iva**, inferior vestibular area; **pfc**, prefacial commissure; **pop**, paroccipital process of petrosal (= mastoid process of *Patterson, Segall & Turnbull (1989)*); **pr**, promontorium of petrosal; **rpp**, rostral process of petrosal; **saf**, subarcuate fossa; **sct**, spiral cribriform tract; **stmn**, stylomastoid notch; **sps**, groove for superior petrosal sinus; **sva**, superior vestibular area; tc, transverse crest; **th**, tympanohyal. Scale bar = 1 cm.

shape in ventral view. It seems likely that the anteromedial elongation of the promontorium in the pampathere/glyptodont/*Proeutatus* clade is homologous to the rostral tympanic process in *Holmesina floridanus;* i.e., the rostral process broadens posteriorly in the former group, so that it blends in with the contour of the

promontorium, whereas the process in *Holmesina floridanus* is substantially reduced in length and breadth, especially posteriorly, but this makes it look like a more distinct, separate process than in the related taxa. The Eocene dasypodine described by *Babot, García-López & Gaudin (2012)* also has an anteromedially elongate petrosal, but this is attributed to the presence of a short epitympanic wing and elongate tensor tympani fossa rather than the presence of a rostral tympanic process. The anterolateral surface of the promontorium in UF 248500 is marked by a large, slightly raised circular boss of unclear function (Figs. 11A–11D). This feature is also present in UF 223813, but is less clear in UF 191448, and is not at all evident in *Vassallia* (FMNH P14424).

Extensive shelving surrounds the promontorium of *Holmesina floridanus* in ventral view. This includes not only the lateral facial sulcus and crista parotica typical of mammalian petrosals (*MacIntyre, 1972*; *Wible et al., 2009*), but also an extensive epitympanic wing anteriorly and a medial flange medially (Figs. 11A–11D). The epitympanic wing is separated by a sizable gap from the underlying rostral tympanic process. In its anterolateral corner, it carries a fossa, particularly well developed in UF 191448 (also UF 223813, 275496), just medial to the groove for the greater petrosal nerve (Figs. 11A–11D; see description of groove below). A small epitympanic wing is present is *Dasypus* (*Wible, 2010*) and Eocene dasypodines (*Babot, García-López & Gaudin, 2012*), but is much better developed in *Euphractus* (*Wible & Gaudin, 2004*). An epitympanic wing is also present in *Vassallia* (FMNH P14424), though it is somewhat less extensive anteriorly. Like *Holmesina*, there is a depression between it and the anteromedial extension of the promontorium, lying just medial to a groove for the greater petrosal nerve. The epitympanic wing is evidently absent in *Proeutatus* (*Patterson, Segall & Turnbull, 1989*) and *Propalaehoplophorus* (YPM VPPU 15007; FMNH P13205).

In extant *Dasypus* (*Wible, 2010*) and *Euphractus* (*Wible & Gaudin, 2004*), the tensor tympani muscle attaches to the tegmen tympani, lateral to the greater petrosal nerve. However, the tegmen is much reduced in *Holmesina* and *Plaina*, and thus we think the tensor tympani likely attached someplace medial to the greater petrosal nerve, as in the Eocene dasypodine described by *Babot, García-López & Gaudin (2012)* and the basal eutherian *Zalambdalestes* (*Wible, Novacek & Rougier, 2004*), where it attaches to the anterolateral surface of the promontorium. This attachment site in pampatheres could be either the fossa in the epitympanic wing, or the boss on the anterolateral surface of the petrosal, both described above (Figs. 10 and 11A–11D). Given that the fossa is present in *Vassallia* and *Holmesina*, whereas the boss is present in only some specimens of *Holmesina floridanus*, we believe the fossa the more likely location, but we cannot rule out the other. In *Proeutatus* (*Patterson, Segall & Turnbull, 1989*) and *Propalaehoplophorus* (YPM VPPU 15007; FMNH P13205), which have a reduced tegmen tympani like pampatheres, but lack an epitympanic wing of the petrosal, the tensor tympani likely attaches to the anterolateral surface of the promontorium, as in Eocene dasypodines and *Zalambdalestes* (*Wible, Novacek & Rougier, 2004*; *Babot, García-López & Gaudin, 2012*).

The medial flange of the petrosal in *Holmesina floridanus* is quite extensive both transversely and anteroposteriorly (especially in the juvenile specimen UF 275496) when compared to that of *Dasypus* (*Wible, 2010*) or Eocene dasypodines (*Babot, García-López &*

*Gaudin, 2012*). In contrast to both of the latter forms, it extends as far forward as the epitympanic wing, creating a squared off anterior edge for the petrosal, and it is covered by a variety of pits and low ridges. The most prominent of these is a pit that is situated at roughly the midpoint of the medial flange, which serves as a point of attachment for the basioccipital (Figs. 11A–11D). In *Dasypus*, a patent basicochlear fissure is maintained into adulthood, so that there is no medial connection between the petrosal and the floor of the basicranium (*Wible, 2010*). Immediately behind this basioccipital facet is a prominent foramen, the cochlear canaliculus (for the perilymphatic duct—see *Clemente, 1985*; *Evans & Christiansen, 1979*; *Wible, 2010*). The medial flange in *Holmesina floridanus* also differs from that of *Dasypus* and Eocene dasypodines in that it extends posteriorly beyond the cochlear canaliculus, reaching the region termed the "triangular shelf" in *Dasypus* (*Wible, 2010*), that is, the roof of the post-promontorial sinus. In so doing it forms a shallow jugular notch, i.e., the anteromedial edge to the jugular foramen. The medial flange of the petrosal is difficult to observe in *Vassallia*, *Propalaehoplophorus*, *Proeutatus,* and even the extant *Euphractus,* because of the lack of preserved, isolated petrosals in these taxa. However, it is clear the latter has an extensive contact between petrosal and basicranium, whereas only a small basioccipital/petrosal contact is present in *Vassallia* (FMNH P14424) and *Proeutatus* (FMNH P13197), as in *Holmesina*.

*Holmesina floridanus* has three prominent foramina in the ventral exposure of the pars cochlearis. The most anterior of these is the laterally directed primary facial foramen, which is hidden in ventral view by a low ridge at the base of the promontorium, and in lateral view by the anteroventral process of the tegmen tympani. This opening transmits the facial nerve (c.n. VII) to the middle ear space (*Wible & Gaudin, 2004*). In some *Dasypus* (as in most therians—*Wible 1990*, *2003*), the space immediately lateral to the primary facial foramen, the cavum supracochleare, has a bony floor, creating a discrete hiatus Fallopii and secondary facial foramen anterior and posterior to the cavum, respectively (*Wible, 2010*). This floor is not present in any *Holmesina floridanus* specimen, nor is it known to occur in Eocene dasypodines (*Babot, García-López & Gaudin, 2012*), *Euphractus* (*Wible & Gaudin, 2004*), *Proeutatus* (FMNH P13197), *Vassallia* (FMNH P14424), or any glyptodont (e.g., *Propalaehoplophorus*, YPM VPPU 15007, FMNH P13205; see also *Patterson, Segall & Turnbull, 1989*).

The second foramen in the ventral pars cochlearis is a larger, laterally directed opening posterior to the primary facial foramen, the fenestra vestibuli, which accommodates the footplate of the stapes (Figs. 11 and 12). As in *Dasypus* and *Euphractus* (*Wible & Gaudin, 2004*; *Wible, 2010*), the opening of the fenestra vestibuli is somewhat recessed, and surrounded by a narrow rim of bone. The opening is rounder in *Holmesina floridanus* than in the extant forms, with a stapedial ratio (sensu *Segall, 1970*; Length/width) of ~1.4, whereas it is 1.9 in *Dasypus* and 1.9–2.0 in *Euphractus. Proeutatus* (FMNH P13197) and *Propalaehoplophorus* (FMNH P13205) also resemble the living taxa in this regard, with ratios of 2.4 and 1.8, respectively. The Eocene dasypodine described by *Babot, García-López & Gaudin (2012)* is intermediate in this regard, with a stapedial ratio of 1.6.

The third opening in the pars cochlearis' ventral surface is a posteriorly directed foramen separated from the rim of the fenestra vestibuli by a broad bar of bone, the crista

interfenestralis, This opening is generally called the fenestra cochleae (we follow *Patterson et al., 1992*; *Gaudin, 1995*; *Wible & Gaudin, 2004* in using this widely recognized term), although, as *Wible (2010)* points out, the latter is actually a separate hole recessed within the cochlear fossula, and this more superficial, posteriorly facing foramen is actually the external aperture of the cochlear fossula. The "fenestra cochleae" of *Holmesina floridanus* is unusual in several respects. First, it is very wide and low, with a ratio of width to depth of approximately 3.4 (Fig. 13B). In *Dasypus* (*Wible, 2010*) and *Propalaehoplophorus* (FMNH P13205), the ratio is closer to 2, whereas in *Euphractus* (UTCM 1491) and *Proeutatus* (FMNH P13197, P13199) the ratio is between 1.0 and 1.2. Although we could not obtain measurements for *Vassallia*, it appears similar in dimensions to *Holmesina*. The fenestra cochleae is also unusual in *Holmesina* in that it is shielded in ventral view by a prominent ridge, and, in UF 191448, it is divided by a ventral process into two separate openings. Neither feature is known to occur in other cingulates.

The crista interfenestralis, between the fenestrae vestibuli and cochleae, also exhibits unusual characteristics in *Holmesina floridanus*. For one, it is quite broad, its maximum width clearly exceeding the maximum diameters of either of the openings flanking it. This is a feature that also occurs in *Vassallia* (FMNH P14424), but not in extant armadillos (*Wible & Gaudin, 2004*; *Wible, 2010*), *Proeutatus* (FMNH P13197, 13199), or *Propalaehoplophorus* (YPM VPPU 15007, FMNH P13205), though it is also fairly broad in the Eocene dasypodine described by *Babot, García-López & Gaudin (2012)*. In addition, the crista is connected laterally by a bony bridge to the base of the tympanohyal, the bridge forming a partial floor to the facial sulcus. This bridge is broken in most specimens of *Holmesina floridanus*, as it is in the isolated left petrosal of UF 248500 (Figs. 11A and 11B), but is intact on the right side of that specimen (Fig. 10), as well as in UF 275496. This appears to be a unique apomorphy of *Holmesina*, although there are low ridges on the crista interfenestralis of *Vassallia* (FMNH P14424), and low tubera in *Euphractus* (UTCM 1491) and *Proeutatus* (FMNH P13197).

There is a narrow elongate groove that runs lateral to the promontorium along its entire length (Figs. 11C and 11D). Anterior to the primary facial foramen, this groove accommodates the greater petrosal nerve. The portion posterior to the primary facial foramen is the facial sulcus for the facial nerve (c.n. VII). The sulci are bordered laterally by a well-developed, sharp edged crista parotica. The latter forms a rounded, U-shaped ventral extension immediately opposite the primary facial foramen. This extension is somewhat rugose and broadened mediolaterally relative to the rest of the crista, and likely represents the anteroventral process of the tegmen tympani (Figs. 11C–11D and 12). The anteroventral process, which is termed the processus crista facialis by *Patterson, Segall & Turnbull (1989)* and others (*Gaudin, 1995*; *Wible & Gaudin, 2004*), is much better developed in extant armadillos, where it typically forms a mediolaterally expanded, cup-shaped depression (*Patterson, Segall & Turnbull, 1989*; *Wible & Gaudin, 2004*; *Wible, 2010*). It may also contact the squamosal, malleus, ectotympanic, entotympanic or alisphenoid bones in living armadillos (*Wible & Gaudin, 2004*; *Wible, 2010*), whereas in *Holmesina floridanus* it is much reduced, and only contacts the squamosal. The anteroventral process is also small in *Vassallia* (FMNH P14424), *Proeutatus* (FMNH

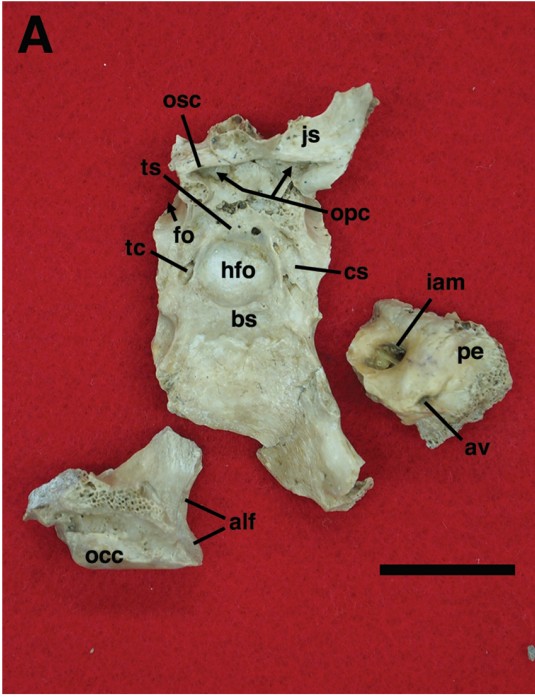
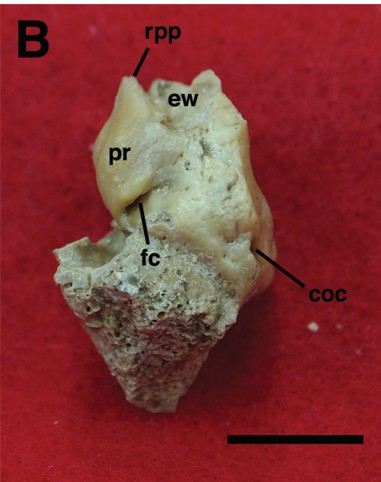

**Figure 13 Isolated right petrosal, left occipital condyle, and basicranium of *Holmesina floridanus*, UF 223813.** (A) Petrosal, basicranium and condyle in dorsal view; (B) petrosal in posterior view. **alf**, fossa for alar ligament of atlas; **av**, aqueductus vestibuli; **bs**, basisphenoid; **coc**, cochlear canaliculus; **cs**, carotid sulcus for internal carotid artery; **ew**, epitympanic wing; **fc**, fenestra cochleae; **hfo**, hypophyseal fossa; **iam**, internal acoustic meatus; **js**, jugulum sphenoidale; **occ**, occipital condyle; **opc**, internal openings of optic canals; **osc**, orbitosphenoid crest; **pe**, petrosal; **pr**, promontorium of petrosal; **rpp**, rostral process of petrosal; **tc**, opening accommodating veins that connect to transverse canal foramen; **ts**, tuberculum sellae. Upper scale bar (A) = 2 cm. Lower scale bar (B) = 1 cm.

P13197), and *Propalaehoplophorus* (FMNH P13205), lacking any concavity and contacting only the squamosal.

Just posterior to the fenestra vestibuli, the facial sulcus traverses the ventral surface of the petrosal pars cochlearis, becoming confluent medially with a large, ovate depression, the stapedius fossa for the stapedius muscle (Figs. 11C and 11D). The sulcus then turns laterally and ventrally, terminating at a shallow stylomastoid notch in the isolated left petrosal of UF 248500. However, on the right, the facial sulcus passes posterior to the tympanohyal, which abuts the large caudal tympanic process of the petrosal posteriorly, enclosing a stylomastoid foramen for the emerging facial nerve (c.n. VII; Fig. 10). An enclosed stylomastoid foramen is also present in *Dasypus* (*Wible, 2010*) and *Euphractus* (*Wible & Gaudin, 2004*). *Patterson, Segall & Turnbull (1989*, fig. 15B) illustrate a very similar morphology for *Vassallia*, though our inspection of their specimen (FMNH P14424) reveals the tympanohyal has subsequently broken off. An enclosed stylomastoid foramen is lacking, however, in both *Proeutatus* and *Propalaehoplophorus*, where the tympanohyal and the paroccipital and caudal tympanic processes of the petrosal frame only ⅔ or ¾ of an opening. The tympanohyal of UF 248500 is broken through its

base on the left, at the posterior terminus of the crista parotica, but on the right it is straight, elongated ventrally and expanded distally, forming a concave, ovate stylohyal fossa similar to, but much smaller and simpler than the structure of the same name so characteristic of sloths (*Patterson et al., 1992*; *Gaudin, 1995*, *2004*). A similar anatomy was apparently present in *Vassallia* (*Patterson, Segall & Turnbull, 1989*; though, as noted above, the tympanohyal in this specimen is now broken), but not in *Dasypus* (*Wible, 2010*), *Proeutatus* (FMNH P13197, P13199) or *Propalaehoplophorus* (YPM VPPU 15007, FMNH P13205). In *Euphractus* (*Wible & Gaudin, 2004*), the circular depression that *Wible & Gaudin (2004)* label a stylohyal fossa has a small tympanohyal exposure in its center, but is formed largely by the ectotympanic anteriorly, and the mastoid region of the petrosal posteriorly and laterally. The tympanohyal is typically not straight in other cingulates, as it is in *Holmesina*. It curves medially and posteriorly at its distal end in *Dasypus* (*Wible, 2010*) and *Proeutatus* (*Patterson, Segall & Turnbull, 1989*, figure 13C), it is posteroventrally directed in *Euphractus* (*Wible & Gaudin, 2004*), and it bends laterally at its distal end in *Propalaehoplophorus* (YPM VPPU 15007, FMNH P13205).

Although it is not evident in the isolated petrosal (due to postmortem breakage), the right petrosal in UF 248500 and both left and right petrosals in UF 191448 are characterized by a massive caudal tympanic process of the petrosal on the ventral pars cochlearis (Fig. 10). The process is concave posteriorly in both specimens, apparently articulating posteriorly with a small elevation on the anterior edge of the exoccipital, although the petrosal is anteriorly displaced from its suture with the exoccipital in both specimens. The caudal tympanic process forms the lateral half of the posterior wall to the stapedius fossa, and lies well lateral to the fenestra cochleae. It occupies a similar position in *Vassallia* (FMNH P14424) and *Dasypus* (*Wible, 2010*), though it is less massive in both taxa. In *Proeutatus* (FMNH P13197) and *Propalaehoplophorus* (YPM VPPU 15007, FMNH P13205) it is both smaller and more medially placed. The caudal tympanic process of *Holmesina floridanus* is separated laterally by a deep notch from the massively enlarged paroccipital process of the petrosal (=mastoid process of *Patterson, Segall & Turnbull, 1989*; *Gaudin, 1995*; and others). This huge paroccipital process is slightly hooked medially and angled anteriorly at its distal extremity, and extends ventral to the level of the basicranial plate (Figs. 4–7 and 10). Though almost cylindrical in appearance, its transverse diameter is in fact a good deal larger than its anteroposterior diameter, and it tapers distally to a rounded tip. The great enlargement of the paroccipital process is evidently a feature of pampatheres in general, because it is present in *Holmesina septentrionalis* (UF 234224; *Edmund, 1985b*), *Holmesina occidentalis* (ROM 3881), *Vassallia* (*Patterson, Segall & Turnbull, 1989*; *De Iuliis & Edmund, 2002*), and *Pampatherium* (*Bordas 1939*; *Guth, 1961*; *Cartelle & Bohórquez, 1985*). The paroccipital process of glyptodonts is massive, but not as elongated as that of pampatheres (YPM PU 15007; *Patterson, Segall & Turnbull, 1989*), whereas the process is much smaller, though still sizable, in *Proeutatus* (labeled as "mastoid process" in *Patterson, Segall & Turnbull, 1989*) and *Euphractus* (*Wible & Gaudin, 2004*). It is also flattened in the latter two taxa, anteroposteriorly in *Euphractus* and obliquely in *Proeutatus* (in an anterolateral/

posteromedial plane). The notch separating the caudal tympanic and paroccipital processes of the petrosal in *Holmesina floridanus* is saddle-shaped, separating the stylomastoid foramen anteriorly from the sulcus for the occipital artery posteriorly (Fig. 10).

The caudal tympanic process is also separated by a medial notch from a small process attached to the back of a broad shelf of bone that lies behind the promontorium. The notch is likely for the auricular branch of the vagus nerve (c.n. X; Fig. 10) based on comparisons with *Dasypus* (*Wible, 2010*). The broad shelf, which is trapezoidal in shape, widening anteriorly (Figs. 11A–11D), is the roof of the postpromontorial sinus, the structure *Wible (2010)* terms the "triangular shelf" in *Dasypus*. This shelf is considerably broader in *Holmesina floridanus*, as it is in *Vassallia* (FMNH P14424) and *Propalaehoplophorus* (YPM VPPU 15007, FMNH P13205). The shelf is semicircular but similar in size to that of *Dasypus* in *Proeutatus* (FMNH P13197), whereas in *Euphractus* (UTCM1491) it remains triangular but is larger and extends further anterolaterally than in *Dasypus*.

As a final aspect visible in ventral view, we note that the area of the petrosal lateral to the crista parotica in *Holmesina floridanus* is a concavity that forms the medial half of the epitympanic recess, which accommodates the mallear and incudal heads in mammals. The lateral half of the recess is formed by squamosal, and is bisected transversely by the postglenoid process. The petrosal portion of the recess has a small divot in the lateral portion of its posterior wall that presumably represents the fossa incudis. The fossa lies immediately above a low ridge that extends anteromedially from the base of the tympanohyal. In *Dasypus*, the lateral wall of the anterolaterally facing fossa incudis is formed by the squamosal (*Wible, 2010*), as was likely true in the Eocene dasypodine described by *Babot, García-López & Gaudin (2012)*, but this does not appear to be the case in *Holmesina*, nor in *Vassallia* (FMNH P14424), *Propalaehoplophorus* (FMNH P13205), or *Proeutatus* (FMNH P13197), where the fossa is more anteriorly oriented. *Euphractus* also lacks squamosal participation in the fossa incudis, but in this case it is due to the presence of an open epitympanic sinus above the ossicles (*Wible & Gaudin, 2004*), as is typical for euphractine armadillos (*Patterson, Segall & Turnbull, 1989*; *Gaudin, 1995*; Node 6 of *Gaudin & Wible, 2006*).

Because of the presence of an isolated petrosal, we are able to describe and illustrate (Figs. 12C and 12D) details of the dorsal surface (i.e., the cerebellar face) of the petrosal that have never been described before in pampatheres. The most distinctive feature visible in a dorsal view of the *Holmesina floridanus* petrosal is a large opening in the anteroventral region of the endocranial exposure (in the pars cochlearis), the internal acoustic meatus. This opening is much deeper than that of *Dasypus* (*Wible, 2010*) and is ventrally displaced, so that it is separated from the endocranial roof of the basicranial plate by only a thin, sharp crest. This arrangement also differs from that in *Vassallia*, *Euphractus*, and Eocene dasypodines (*Babot, García-López & Gaudin, 2012*), in which the meatus is equally deep but more dorsally positioned.

At the bottom of the internal acoustic meatus is a series of openings that have been identified (Figs. 12C and 12D) based on *Wible's (2010)* description of *Dasypus*. The

openings are clustered into two groups, separated by a sharp transverse crest. In *Dasypus*, the transverse crest is broad and rounded, whereas in *Euphractus* it is broad but with a sharp medial edge. The anatomy in *Vassallia* (FMNH P14424) and Eocene dasypodines (*Babot, García-López & Gaudin, 2012*) is much like that of *Holmesina*. The two openings above (i.e., dorsal and lateral to) the transverse crest are the facial foramen for the facial nerve (c.n. VII), and posterior to that and roughly equivalent in size, the superior vestibular area. Below the transverse crest there are three openings: anteromedially, the large spiral cribriform tract, separated by a strong crest from two smaller, more posterior openings in a common fossa, a more posteromedial foramen singulare and a more anterolateral inferior vestibular area. The arrangement of these openings is very similar in *Vassallia*, whereas in *Dasypus* there is no real crest separating the spiral cribriform tract from the more posterior foramina. Moreover, the posterior foramina are clearly visible in medial view in pampatheres, whereas in *Dasypus* they face more anteriorly (foramen singulare) or ventrally (inferior vestibular area; *Wible, 2010*; UTCM 801[isolated petrosal]). *Euphractus* also lacks the septum between the spiral cribriform tract and the two posterior foramina, which are quite small, and located in close proximity along the posterior wall of the lower opening of the internal acoustic meatus (UTCM 1486). At the medioventral edge of the petrosal's endocranial surface, slightly posterior to the internal acoustic meatus, lies the opening of the cochlear canaliculus. It occupies the same position in *Vassallia*, *Dasypus* (*Wible, 2010*), and Eocene dasypodines (*Babot, García-López & Gaudin, 2012*), whereas in *Euphractus*, where the ventromedial edge of the petrosal contacts the basicranial plate along its whole length, the cochlear canaliculus occupies a more dorsal, endocranial position.

Anterior to the internal acoustic meatus is a distinct concavity, which may have accommodated the inferior petrosal sinus (Figs. 12C and 12D). A similar concavity is present in *Vassallia* and *Euphractus*, but is absent in *Dasypus* (*Wible, 2010*). In the Eocene dasypodine described by *Babot, García-López & Gaudin (2012)*, the sulcus for the inferior petrosal sinus is clearly marked along the ventral edge of the petrosal. The region immediately dorsolateral to the meatus, the so-called prefacial commissure, is broad and swollen in both pampatheres. In *Dasypus* (and apparently in Eocene dasypodines; *Babot, García-López & Gaudin, 2012*) it is a narrow bar of bone (*Wible, 2010*), whereas in *Euphractus* is is broad like the pampatheres, but flat rather than swollen. The prefacial commissure in *Holmesina* is surmounted by a rounded crista petrosa that at its posterodorsal end is divided into medial and lateral ridges by a vascular groove (Figs. 11G and 11H). This groove is situated too far medially to carry the postglenoid vein described by *Wible (2010)* in *Dasypus*, and so we suspect it carried the superior petrosal sinus. This groove is also present in *Vassallia* and Eocene dasypodines (*Babot, García-López & Gaudin, 2012*), though it is missing in both *Dasypus* and *Euphractus*, both of which have a much sharper crista petrosa. Indeed, in *Euphractus* (UTCM 1486), the crista petrosa is so elevated that it resembles a low ossified tentorium, like that of pangolins and carnivorans (*Gaudin, Emry & Morris, 2016*), extending a short distance dorsally between the cerebrum and cerebellum. *Euphractus* also has a very large, concave cerebral surface of the petrosal,

whereas in *Dasypus* (*Wible, 2010*), Eocene dasypodines (*Babot, García-López & Gaudin, 2012*), and pampatheres this surface is much smaller.

The endocranial exposure of the pars canalicularis is occupied by a broad, deeply concave subarcuate fossa in all the cingulates examined in this report. It is narrower anteroposteriorly in *Euphractus* than in *Dasypus* or pampatheres (whereas the subarcuate fossa is very incompletely preserved in the *Babot, García-López & Gaudin (2012)* specimen). In the latter forms it takes on a rounded triangular shape, with its apex pointing ventromedially. In pampatheres and *Euphractus*, it is divided by a low, rounded, roughly transverse ridge into upper and lower concavities. The upper concavity is further divided by a low ridge into anterior and posterior concavities in *Holmesina* and *Vassallia*. The first, more horizontal ridge is almost certainly created by the lower portion of the posterior semicircular canal, whereas the second, more vertical ridge is created by the crus commune of the anterior and posterior semicircular canals. The aqueductus vestibuli, which transmits the endolymphatic duct, takes the form of a vertical slit opening into the ventromedial corner of the subarcuate fossa (Figs. 12C and 12D). It has the same shape and position in *Vassallia*. In *Euphractus*, this opening is quite close to the exoccipital bone posteriorly, in contrast to pampatheres, and in *Dasypus* (*Wible, 2010*) and Babot et al.'s Eocene dasypodine (2012), it is located just outside the subarcuate fossa, in a slightly more ventral, medial and anterior position. *Billet, Hautier & Lebrun (2015)* note that the vestibular aqueduct displays a derived orientation in euphractines, perpendicular rather than subparallel to the crus commune. The position of this aperture suggests that pampatheres share with dasypodines the more plesiomorphic orientation. A small opening into the recessus angularis, like that described for *Dasypus* by *Wible (2010)*, is present on the dorsolateral rim of the subarcuate fossa in *Holmesina floridanus*. On the right side of the UTCM 1486 specimen of *Euphractus* there is a similar opening; however, on the left side, there are three or four small vascular foramina in this area, some within and some outside the subarcuate fossa, the middle opening on the rim the largest. As noted by *Wible (2010)*, the recessus angularis opening may or may not lie within the subarcuate fossa in *Dasypus*.

We have illustrated the isolated left petrosal of *Holmesina floridanus* (UF 248500) in lateral view (Figs. 12A and 12B), much as *Wible (2010)* has done for *Dasypus*. As in *Dasypus*, there are three broad regions of the petrosal of *Holmesina* recognizable in lateral view. There is a cerebral surface, exposed in the floor of the middle cranial fossa of the endocranium. Like *Dasypus* this surface is elongated along an anteroventrolateral to posterodorsomedial axis, and is relatively narrow transversely, though it is less triangular and more ovate in *Holmesina*. The tympanic exposure includes the promontorium, with its prominent elongated rostral tympanic process and large lateral, circular boss of unknown function. The fenestra vestibuli is also visible laterally, but not the primary facial foramen, which is hidden by a distinct ventral, semicircular ventral extension, the anteroventral process of the tegmen tympani. This process is present in *Dasypus* (*Wible, 2010*), but does not extend ventrally to the same degree. Like *Dasypus*, this tympanic exposure also includes portions of the epitympanic recess situated lateral to the crista parotica. The petrosal contribution to the fossa incudis lies at the posterior and dorsal

extremity of this surface, as in *Dasypus* (*Wible, 2010*), but is less clearly marked. The tympanohyal is prominently exposed in *Dasypus* in lateral view, but is broken off in UF 248500. The remainder of the lateral exposure in UF 248500 is comprised of a posterodorsal contact surface for the squamosal, and the broken remains of the paroccipital process. Because the latter is so much larger in *Holmesina* than in *Dasypus*, it accounts for a much larger portion of this lateral surface, despite the fact that most of the process is broken off in the illustrated specimen.

The mastoid exposure of the petrosal is largely missing from the isolated petrosal, due to postmortem breakage, and so this region of the petrosal will be described based on the in situ right petrosal from UF 248500, and on UF 191448. In lateral view, the dominant feature of the mastoid exposure in *Holmesina floridanus* is the gigantic paroccipital process (Figs. 4, 5 and 8), which, as noted above, has a slight medial hook and is angled anteriorly at its distal extremity, extends ventral to the level of the basicranial plate, and is slightly compressed anteroposteriorly with a rounded tip. It has a clear, sigmoid suture dorsally with the squamosal (and its post-tympanic process) in both specimens, extending in a posterodorsal to anteroventral direction. The lateral edge of the paroccipital process is continuous dorsally with the nuchal crest. As previously observed, the morphology of this region is similar in all pampatheres (*Bordas, 1939*; *Guth, 1961*; *Cartelle & Bohórquez, 1985*; *Edmund, 1985b*; *Patterson, Segall & Turnbull, 1989*; *De Iuliis & Edmund, 2002*), whereas the shape and size of the paroccipital process is variable in other cingulates (*Patterson, Segall & Turnbull, 1989*; *Wible & Gaudin, 2004*; *Wible, 2010*).

In posterior view, the mastoid region has a broad, rectangular (UF 191448) or rhomboid (UF 248500) exposure on the occiput (Fig. 14). In UF 191448, the transversely elongated exposure is marked by two narrow vertical depressions. The deeper and more medial of these is the sulcus for the occipital artery, which arises as a deep notch between the paracondylar process of the exoccipital and the paroccipital process, and terminates dorsally at the posttemporal foramen (the posterior opening of the posttemporal canal for the arteria diploetica magna—see *Wible & Gaudin, 2004*). This opening lies just below the suture between the mastoid and the occipital exposure of the squamosal. The second, more lateral and much shallower vertical depression represents the attachment surface for the digastric muscle, travelling along the inside edge of the nuchal crest. This depression does not reach the tip of the paroccipital process ventrally, but dorsally it extends beyond the mastoid, crossing the occipital surface of the squamosal onto the supraoccipital. It terminates just below a large muscular boss on the nuchal crest. The morphology of UF 248500 differs from that of UF 191448 in several respects. Most importantly, the shape of the occipital exposure is different—it is more rhomboid than rectangular, with its dorsal border sloping ventrolaterally. Additionally, the digastric fossa is shallower, and has a sigmoid shape. In *Holmesina septentrionalis* (UF 234224), the digastric fossa takes on a shape similar to that in UF 248500, and the occipital artery sulcus is bowed medially. The mastoid occipital exposure is even broader mediolaterally in *Vassallia* than in *Holmesina*, taking on a "Y-shape" as indicated by *De Iuliis & Edmund (2002*, p. 56), with medial and lateral extensions passing dorsal to the posttemporal foramen (="mastoid foramen" of *De Iuliis & Edmund, 2002*). In *Propalaehoplophorus* (YPM VPPU 15007), *Proeutatus*

(FMNH P13197), and *Euphractus* (UTCM 1491), the digastric fossa is much shorter vertically than in pampatheres, restricted to the posterior surface of the paroccipital process, and not extending dorsally onto the squamosal and supraoccipital. *Holmesina floridanus, Holmesina septentrionalis* (UF 234224), and *Vassallia* (FMNH P14424) all have a groove for the occipital artery extending dorsally from the posttemporal foramen across the squamosal and onto the supraoccipital. This condition was also described in *Euphractus* by *Wible & Gaudin (2004)*, and is optimized as a cingulate synapomorphy by *Gaudin & Wible (2006)*.

### Ectotympanic, entotympanic, ear ossicles

To our knowledge, no remnant of the ectotympanic or ear ossicles has ever been recovered in any pampathere, and our specimens, well-preserved though they are, have proven no exception (*Guth (1961)* described partial stapes elements in several glyptodonts, but not any portion of the ectotympanic or other ossicles). There appear to be facets for the attachment of the anterior and posterior crura of the ectotympanic preserved in UF 248500, on the ventromedial surface of the squamosal's entoglenoid process, and on the anterior surface of the tympanohyal and the portion of the petrosal forming the anterior wall of the stylomastoid foramen, respectively. This suggests that the ectotympanic formed a loosely attached, dorsally incomplete ring. There is also no indication of the presence of an entotympanic element—indeed, none has ever been described in any pampathere or glyptodont, despite its occurrence in *Euphractus* (*Wible & Gaudin, 2004*) and many other cingulates (*Patterson, Segall & Turnbull, 1989*; *Wible, 2010*).

### Vomer

The vomer of *Holmesina floridanus* is only partially visible in two places. It can be seen anteriorly through the external narial opening of UF 248500, as an elongate ridge extending dorsally from the roof of the maxillary palatine processes into the nasal cavity. Here it is Y-shaped in cross section, with the base in the midsagittal plane and the dorsal arms of the "Y" supporting the ossified portion of the median nasal septum. It appears to come to an abrupt anterior termination well behind the internal openings of the incisive foramina, therefore it likely did not contact the premaxilla, in contrast to *Vassallia, Propalaehoplophorus* and most other cingulates (*Gaudin & Wible, 2006*). The vomer is also visible looking posteriorly through the choanae of UF 191448, as a pair of nearly vertical alae extending along the lateral edge of the presphenoid, converging anteriorly until they meet in the midline, perhaps covering the anteriormost tip of the presphenoid ventrally. Much of the posterior and ventral reaches of these alae are broken, but they likely contacted the maxilla and perhaps the palatine ventrally along the lateral walls of the nasal passage.

### Presphenoid/orbitosphenoid

There is a clear suture between the presphenoid and basisphenoid in UF 248500, and the posterior portion of the presphenoid is visible in ventral view extending a short distance

posterior to the choanae, although most of the anterior presphenoid is missing (Figs. 6 and 7). The entire presphenoid is preserved in UF 191448, though it is fused into the surrounding elements, so that its precise boundaries are no longer evident (Fig. 9B). Nevertheless, it can be inferred from the two specimens that the presphenoid takes the form of a narrow, elongate triangle that tapers anteriorly until disappearing beneath the vomer within the nasal cavity. As noted above, the anterior presphenoid connects laterally with the vomerine alae inside the nasal cavity, and likely contacts the palatine and pterygoid posterolaterally, although UF 248500 has ventrolateral flanges of the basisphenoid that extend lateral to the posteriormost parts of the presphenoid, and could preclude contact with the pterygoid. The presphenoid has a very similar form in other cingulates. In *Vassallia*, there is a ventrolateral projection of the basisphenoid that extends forward to preclude pterygoid/presphenoid contact, as in *Holmesina floridanus*.

The lateral portions of the orbitosphenoid, i.e., the areas where it would normally be exposed at the surface along the medial orbital wall, are badly damaged in UF 248500. There is also some damage in this area in UF 191448, and the orbital sutures are all closed in this specimen, making it difficult to assess orbitosphenoid anatomy. However, two additional specimens of *Holmesina floridanus*, UF 121742 and UF 223813, provide better insight. The former is an exquisitely preserved display specimen and shows the surface exposure in the orbital wall, the latter a fragmentary specimen that preserves the endocranial portion of the orbitosphenoid (which can also be glimpsed though breaks in UF 248500). The specimens taken together show that the optic nerve (c.n. II) is completely enclosed in a canal formed by the orbitosphenoid bone (Fig. 8), as is typical for placental mammals (*Novacek, 1993*). The lateral wall of this canal forms the medial wall of a combined sphenorbital fissure (transmitting c.n. III, IV, $V_1$, and VI, as well various orbital blood vessels) and foramen rotundum (transmitting c.n. $V_2$). In nearly all cingulates, these two openings are fused.

The endocranial surface of the presphenoid/orbitosphenoid (Fig. 13A) is marked by a strong, continuous orbitosphenoid crest surmounting the internal apertures of the left and right optic canals, but the jugulum sphenoidale (i.e., the surface of the presphenoid/orbitosphenoid rostral to the orbitosphenoid crest—using terminology of *Evans & Christiansen (1979)*, *Wible (2008)*) is only weakly convex in the midline. In many cingulates, including *Euphractus*, there is a strong midline crest in this area (*Gaudin & Wible, 2006*—note *Euphractus* is coded as lacking this feature, but should be coded as variably present, because a sharp crest is present in UTCM 1491, and a weaker, rounded crest is present in UTCM 1500). As in *Holmesina*, the midline crest itself is only weakly developed in *Vassallia*, but the entire jugulum singulare is swollen and convex, quite unlike the condition in *Holmesina*.

The surface exposure of the orbitosphenoid in the medial orbital wall is relatively small and ovate or rectangular in UF 121742, and elongated along a posteroventral to anterodorsal axis (Fig. 8). It contacts the frontal anteriorly and dorsally, the maxilla and alisphenoid ventrally, and is separated by a gap from the lateral wall of the common opening for the optic foramen and sphenorbital fissure. The orbitosphenoid forms the medial wall of this common opening. In contrast to *Euphractus* (*Wible & Gaudin, 2004*)

and *Proeutatus* (*Gaudin & Wible, 2006*, fig. 6.6a), there does not appear to be contact between the palatine and orbitosphenoid, although it is possible that there is a connection at the base of the medial wall for the common fossa that holds the optic foramen/ sphenorbital fissure and sphenopalatine canal. The optic foramen transmitted the optic nerve (c.n. II), whereas the sphenorbital fissure likely accommodated the oculomotor (c.n. III), trochlear (c.n. IV), abducens (c.n. VI), and first and second branches of the trigeminal nerve (c.n. $V_1$, the ophthalmic, and c.n. $V_2$, the maxillary), along with a variety of vessels (*Wible & Gaudin, 2004*).

The orbitosphenoid does not participate in the rim of either the sphenopalatine foramen or the ethmoid foramen in UF 121742. Both conditions are known to occur at least variably in euphractine armadillos (*Gaudin & Wible, 2006*), but *Gaudin & Wible (2006)* code both as absent in *Vassallia*, as they are in *Holmesina*. Like *Vassallia*, *Propalaehoplophorus*, and *Proeutatus* (*Gaudin & Wible, 2006*—an unambiguous synapomorphy of Node 7), the optic foramen (i.e., the lateral opening of the optic canal) is hidden in lateral view by the lateral wall of the fossa housing the combined optic foramen/sphenorbital fissure, unlike *Euphractus* and most other extant armadillos (*Gaudin & Wible, 2006*), in which the optic foramen is visible laterally. The small opening to the pterygoid canal lies on the suture between the orbitosphenoid and alisphenoid, just anterior to the optic foramen/sphenorbital fissure common opening, and at the base of a bony bridge that connects the alisphenoid and maxilla and forms a partial lateral wall to the common fossa for the sphenopalatine canal and the optic foramen/sphenorbital fissure (Fig. 8). The position of the pterygoid canal foramen is similar in *Vassallia*, *Proeutatus*, and *Euphractus* (*Wible & Gaudin, 2004*; *Gaudin & Wible, 2006*), whereas in *Propalaehoplophorus*, this foramen lies within the common fossa for the optic foramen and sphenorbital fissure (*Gaudin & Wible, 2006*). The lateral surface of the orbitosphenoid is marked by a rounded, anterodorsally directed ridge in UF 121742. This ridge lies ventral to a groove emerging from the optic foramen; a similar ridge is formed by the frontal bone above this groove, separating it from the ethmoid foramen.

## Alisphenoid

The alisphenoid is apparently quite large in *Holmesina floridanus,* with a shallow bowl-like surface contour (Fig. 8). It has sutural connections dorsally with the orbitosphenoid, frontal, and squamosal, the first being the shortest and most anterior. It is roughly horizontal. The middle section is positioned more dorsally, and travels posterodorsally, meeting at a point with the squamosal suture, which sweeps posteriorly and ventrally in a great semicircular curve, crossing the entoglenoid process at its posteriormost extremity, so that the alisphenoid forms roughly the anterior third of this process. The alisphenoid has a generally horizontal suture ventrally with the pterygoid, taking part in the dorsalmost lateral rugosities of this element. As noted above, it contacts either a thin sliver of palatine or the maxilla anteriorly, and forms the posterior half of the rim for the sphenopalatine foramen. There is no contact between alisphenoid and parietal, as noted above.

The large foramen ovale (for the mandibular branch of the trigeminal nerve, c.n. $V_3$; *Wible & Gaudin, 2004*) is housed completely within the alisphenoid, as in most cingulates

(*Gaudin et al., 1996*), though the squamosal does closely approach its dorsal margin (Fig. 8). There is a small transverse canal foramen (for a vein from the cavernous sinus—see *Wible & Gaudin, 2004*) anteroventral to the foramen ovale in UF 121742, and on the left side of UF 191448. On the right side of UF 191448, and in UF 275496, there are two small foramina in this position, whereas the foramen appears to be absent in UF 248500. This feature is present in most cingulates (it is an ambiguous synapomorphy of Node 3 in *Gaudin & Wible, 2006*). There is no separate foramen rotundum in *Holmesina floridanus*. As in all known cingulates except *Stegotherium*, this opening has become confluent with the sphenorbital fissure (*Gaudin & Wible, 2006*). The alisphenoid also likely forms at least the lateralmost parts of the piriform fenestra's anterior edge (Fig. 10), though it is difficult to be certain of the contribution because of fusion between the alisphenoid and basisphenoid posteromedially.

The alisphenoid has a prominent, rounded posterior edge that forms the terminus for the lateral wall of the nasopharynx. Just below its suture with the frontal, it is traversed by a sharp crest that originates on the anteromedial corner of the glenoid articular surface and extends anteriorly across the squamosal and alisphenoid. This is a posterior section of the infratemporal crest (Fig. 8). It terminates anteriorly at a large boss, where it joins the anterior portion of the infratemporal crest described above in connection with the frontal bone. This boss likely serves as the site of origin for most of the extrinsic eye muscles, and would therefore be homologous with the ossified ala hypochiasmatica described by *Wible & Gaudin (2004)* in *Euphractus*, though it is carried by the alisphenoid rather than the orbitosphenoid. The anatomy of these crests is very similar to *Holmesina floridanus* in *Holmesina septentrionalis* (UF 234224), *Vassallia* (FMNH P14424), and *Propalaehoplophorus* (YPM VPPU 15007). The alisphenoid terminates anteriorly in a thin, freestanding crest that marks the lateral margin of the fossa housing the optic foramen/sphenorbital fissure and the sphenopalatine canal. As noted above, it also forms a bony bridge lateral to this fossa that connects anteriorly with the maxilla. The entoglenoid portion of the alisphenoid in UF 248500 bears a shallow groove that runs anteroventromedially toward the foramen ovale, which likely accommodated the chorda tympani nerve.

### Basisphenoid

The basisphenoid and basioccipital are fused in all the *Holmesina floridanus* specimens available to us, so we cannot determine the boundary between the two with certainty. In other cingulates for which the suture is known (*Gillette & Ray, 1981*; *Patterson, Segall & Turnbull, 1989*; *Wible & Gaudin, 2004*; *Wible, 2010*), the boundary lies anterior to the basioccipital tubera, roughly at the level of the carotid foramina. We will assume a similarly positioned boundary here (Fig. 7).

The main body of the basisphenoid has a flat ventral surface contour and is trapezoidal in outline, tapering anteriorly. It contacts the presphenoid anteriorly and the basioccipital posteriorly. Along its lateral margins, it bears a prominent, ventrally curving flange. In UF 248500, this flange has a sutural outline anteriorly, although, due to damage in this area it is unclear if the bone to which it is sutured is palatine, pterygoid, or perhaps even

alisphenoid. More posteriorly, this flange, if present, is fused to the alisphenoid—there are vague indications of a basisphenoid/alisphenoid contact emerging from the piriform fenestra, crossing the anteriormost region of the entoglenoid process and extending anteriorly onto the nasopharyngeal wall in UF 121742. The ventral basisphenoid flange is visible in UF 275496 (a juvenile specimen), but is not visible in UF 191448 due to sutural fusion. The ventral flange is also present in *Vassallia* (FMNH P14424). In both *Holmesina* and *Vassallia* this flange has a triangular anterior extension that reaches forward beyond the level of the presphenoid/basisphenoid suture, presumably separating the vomerine alae from the palatine and/or pterygoid. Although it is not illustrated by *Wible & Gaudin (2004)*, at least three specimens of *Euphractus* (UTCM 1486, 1491, and 1500) examined for this study have a small ventral flange of the basisphenoid. It is much smaller than in pampatheres, restricted anteriorly and triangular in shape. It extends laterally between the nasopharyngeal exposures of the pterygoid and palatine.

The posterolateral corner of the basisphenoid bears a concave, semicircular indentation for the carotid foramen (Figs. 6, 7, 9 and 10), the opening transmitting the internal carotid artery into the braincase (*Wible & Gaudin, 2004*). As noted in our description of the alisphenoid, these two sphenoid elements also form the anterior margin of the piriform fenestra (along with the entoglenoid process of the squamosal), though their relative contributions are unclear due to sutural fusion in this area. A short distance anterior to this indentation, a narrow longitudinal groove forms in both UF 248500 and UF 191448. It travels anteriorly across the basisphenoid, beginning near the junction of its ventral flange and body, but shortly thereafter curving ventrally and then traveling straight for the remainder of its course across the ventral flange. This is the groove for the vidian nerve (Fig. 10). Its anterior terminus is missing in UF 248500, but in UF 191448 it terminates at the medial opening for the pterygoid canal, located at the junction of the ventral basisphenoid flange and the perpendicular plate of the palatine. This open groove for the vidian nerve is nearly identical in form in *Vassallia* (FMNH P14424), and an open groove of somewhat different form is preserved in *Proeutatus*, whereas in some cingulates, like *Euphractus*, the nerve is partially enclosed by a canal, and in others, e.g., *Propalaehoplophorus*, it is fully enclosed by a canal (*Gaudin & Wible, 2006*).

The dorsal surface of the basisphenoid is exposed in UF 223813. It is marked by a large, deep, circular hypophyseal fossa, flanked laterally by prominent grooves for the internal carotid arteries (Fig. 13A). In the roof of the internal carotid sulci are bilaterally symmetrical openings—small breaks in the basisphenoid show that these are connected to a canal within the tuberculum sellae (i.e., the eminence in front of the hypophyseal fossa), and are likely part of the cavernous sinus system, accommodating the veins that open at the transverse canal foramen anteroventral to the foramen ovale.

## Basioccipital

The basioccipital forms the remainder of the basicranial surface, accounting for over half its length (if we are reconstructing the position of the basisphenoid/basioccipital suture correctly). It has straight lateral margins that converge only slightly anteriorly in *Holmesina floridanus* (Fig. 7). The basioccipital is considerably shorter and wider in

*Holmesina septentrionalis* (UF 234224) and *Holmesina occidentalis* (ROM 3881), with lateral margins that are more steeply inclined, whereas the proportions of the basioccipital in *Vassallia* (*De Iuliis & Edmund, 2002*), *Propalaehoplophorus* (*Scott, 1903*), and *Proeutatus* (*Patterson, Segall & Turnbull, 1989*) are more like those of *Holmesina floridanus*. The basioccipital lateral margins are largely separate from the petrosal in *Holmesina floridanus*, although, as noted above, there is an articulation between the two bones, with a knob forming on the dorsal edge of the basioccipital's lateral margin, fitting into a depression in the medial flange of the petrosal and interrupting the otherwise open basicochlear commissure. At its posterior limit, the lateral margin of the basioccipital curves laterally, forming the anterior half of the notch for the jugular foramen. The jugular foramen provides passage outside the cranium to the glossopharyngeal (c.n. IX), vagus (c.n. X), and accessory (c.n. XI) nerves and the sigmoid sinus, but likely lies too far medially to be involved with the occipital artery as it is in *Euphractus* (*Wible & Gaudin, 2004*). UF 248500 retains the suture between the exoccipital and basioccipital, showing it as a nearly horizontal contact that extends from the medial margin of the jugular foramen to the anterior portion of the ventral rim of the foramen magnum (Fig. 6). In *Euphractus* (*Wible & Gaudin, 2004*), this suture runs more diagonally, contacting the rim of the jugular foramen further anteriorly and the foramen magnum further posteriorly.

The ventral surface of the basioccipital is convex transversely and highly irregular, marked by several prominent elevations and depressions. The anteriormost of these include two prominent lateral tubercles flanking an even taller median crest (Figs. 6, 7 and 9). These represent the basioccipital tubera and pharyngeal tubercle, respectively [based on comparison with *Canis* (*Evans & Christiansen, 1979*) and *Homo* (*Clemente, 1985*)], the former serving as the site of attachment for the m. longus capitis. Behind the basioccipital tubera are large, shallow depressions, elongated along a posterolateral to anteromedial axis that accommodated the insertion of the m. rectus capitis ventralis. In *Holmesina septentrionalis* (UF 234224) and *Holmesina occidentalis* (ROM 3881), the pharyngeal tubercle and rectus capitis fossae are less well-developed, whereas only the latter is reduced in *Vassallia* (*Gaudin & Wible, 2006*). *Proeutatus* resembles the morphology in *Holmesina floridanus*, but the basioccipital tubera are more elongated along an oblique axis, whereas the basioccipital surface relief is much reduced in both *Propalaehoplophorus* and *Euphractus* (*Wible & Gaudin, 2004*; *Gaudin & Wible, 2006*).

**Exoccipital/supraoccipital**

The occiput is a single fused plate in UF 191448, as is typical among adult mammals, but in the subadult UF 248500 the demarcations among its constituent elements are still visible, including the contact between just described basioccipital and the exoccipital elements on the skull base, as well as the junction between the exoccipitals and supraoccipital on the posterior surface of the skull.

The paired exoccipitals have two primary sections: a horizontal moiety on the skull base; and a vertical portion that forms part of the occipital surface. The former joins the basioccipital at its anteroventral extremity, at a suture that passes medially from the

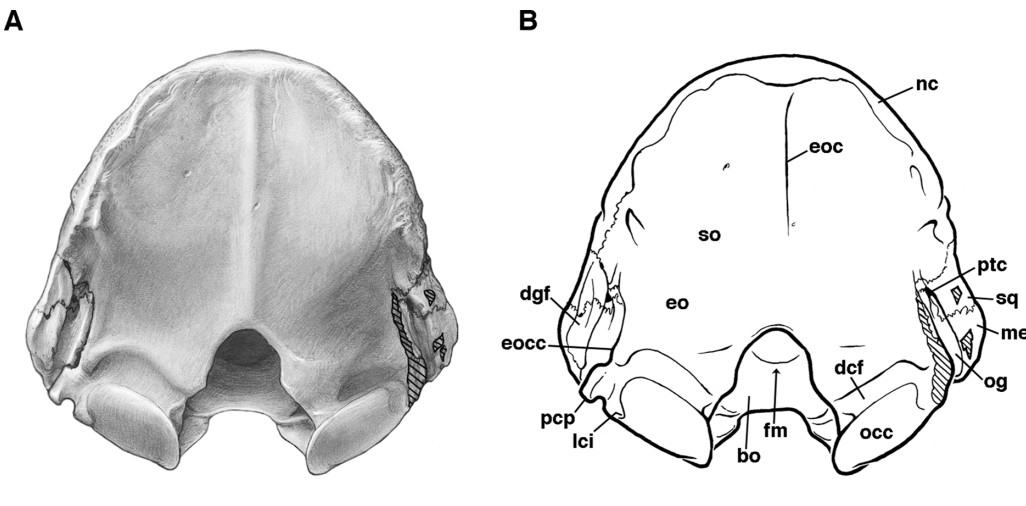

**Figure 14 Skull of *Holmesina floridanus* in posterior view.** (A) UF 191448; (B) Reconstruction. **bo**, basioccipital; **dcf**, dorsal condyloid fossa; **dgf**, digastric fossa; **eo**, exoccipital; **eoc**, external occipital crest; **eocc**, exoccipital crest; **fm**, foramen magnum; **lci**, lateral condyle indentation, i.e., indentation on lateral edge of occipital condyle; **nc**, nuchal crest; **occ**, occipital condyle; **og**, groove for occipital artery; **me**, mastoid exposure of petrosal; **pcp**, paracondylar process of exoccipital (=paroccipital process of *Patterson, Segall & Turnbull (1989)*); **ptc**, posttemporal canal; **so**, supraoccipital; **sq**, squamosal. Scale bar = 5 cm.               

jugular foramen. It is not clear if the suture enters the rim of the foramen magnum, or meets its opposite anterior to the rim of the foramen magnum. Damage to the medial portions of both the left and right exoccipitals of UF 248500 leaves a sizable gap in this area (Fig. 6B). The posterior, vertical segment of the exoccipital shares a lateral suture with the mastoid part of the petrosal. This suture extends from the base of the paracondylar process dorsally to the base of the supraoccipital. Any connection between the occipital exposure of the squamosal and the exoccipital is precluded by a dorsal contact between the mastoid petrosal and the supraoccipital (Fig. 14). The crack that we interpret as the exoccipital/supraoccipital suture in UF 248500 is not perfectly symmetrical, and so may not represent the actual suture, but it occupies almost an identical position as that of the extant armadillo *Euphractus* (*Wible & Gaudin, 2004*, fig. 5), extending ventromedially from the supraoccipital/mastoid suture to the dorsal rim of the foramen magnum. A specimen of *Propalaehoplophorus*, YPM VPPU 15007, has a nearly identical suture on the left side only. Lastly, there is an asymmetrical crack in roughly the same area of the occiput in *Vassallia* (FMNH P14424), though it is oriented at a shallower angle and so does not appear to enter the dorsal rim of the foramen magnum, which would then be formed entirely by the exoccipital.

  The lateral edge of the exoccipital's basicranial segment is marked by a distinct concavity that represents the jugular notch, i.e., the medial edge of the jugular foramen. As noted above, the anterior portion of the jugular notch is formed by the basioccipital. Extending more laterally than posteriorly from this notch is a sutural contact between exoccipital and mastoid. This suture is broadly open in both UF 191448 and UF 248500

(Fig. 10), but this is presumably due to postmortem displacement of the petrosal. At the lateral extremity of this contact surface, the exoccipital bears a strong, free-standing ventral projection, the paracondylar process (=paroccipital process of *Patterson, Segall & Turnbull (1989)*, *Gaudin (1995)*; =jugular process of *Wible & Gaudin (2004)*). In posterior view, the paracondylar process has a convex lateral border and a concave medial border, giving it a hooked appearance, and it is separated by a distinct notch from the lateral edge of the occipital condyle (Figs. 6, 7, 10 and 13). This morphology is apparently a general feature of pampatheres, because it is also present in *Holmesina septentrionalis* (UF 234224), *Holmesina occidentalis* (ROM 3881), *Holmesina paulacoutoi* (*Cartelle & Bohórquez, 1985*), *Vassallia* (FMNH P14424), and *Pampatherium* (*Bordas, 1939*; *Guth, 1961*). In *Propalaehoplophorus* (YPM VPPU 15007) the paracondylar process is well developed, but more blunt, and neither hooked medially nor separated by a notch from the occipital condyle. The process is dramatically reduced by comparison in both *Proeutatus* (FMNH P13197) and *Euphractus* (*Wible & Gaudin, 2004*). Just medial to the jugular notch is a strong fossa that houses the hypoglossal foramen at its base. In UF 248500, there are two hypoglossal foramina, each connecting to a corresponding opening just inside the foramen magnum within the cranial cavity, and providing passage for the hypoglossal nerve (c.n. XII) and vein (*Wible & Gaudin, 2004*). In UF 191448, there appears to be a single hypoglossal foramen. This mirrors the variation noted for *Euphractus* by *Wible & Gaudin (2004)*, whereas *Gaudin & Wible (2006*, char. 153*)* record only a single hypoglossal foramen in *Vassallia*, *Propalaehoplophorus*, and *Proeutatus*.

The hypoglossal fossa of *Holmesina floridanus* sits at the medial edge of a second, broader and shallower fossa that lies just anterior to the occipital condyle, the ventral condyloid fossa of *Wible & Gaudin (2004)*. Medial to these two depressions, the ventral surface of the exoccipital is transversely convex, and terminates at a strong, rounded ridge, which is the lateral edge of the foramen magnum. The transverse convexity of the exoccipital's basicranial exposure is another general feature of pampatheres, present in *Holmesina septentrionalis* (UF 234224), *Holmesina occidentalis* (ROM 3881), and *Vassallia* (FMNH P14424); but not in *Propalaehoplophorus* (YPM VPPU 15007), *Proeutatus* (FMNH P13197), or *Euphractus* (*Wible & Gaudin, 2004*), where the basicranial portion of the exoccipital is flat.

Prominent occipital condyles join the vertical and horizontal segments of the exoccipital (Figs. 6, 7 and 13). The condyles are cylindrical (="roughly rectangular" in ventral view, char. 155[1] in *Gaudin & Wible (2006)*) in shape, an unambiguous synapomorphy of Cingulata according to *Gaudin & Wible (2006)*. The lateral edge bears a distinct indentation that is present in all cingulates except *Peltephilus* (Node 2 of *Gaudin & Wible, 2006*). The portion of the condyle anterior and ventral to this indentation extends much further laterally than the more dorsal and posterior portion. This is also a feature of in *Holmesina septentrionalis* (UF 234224), *Holmesina occidentalis* (ROM 3881), and *Propalaehoplophorus* (YPM VPPU 15007), whereas in *Proeutatus* (FMNH P13197) and *Euphractus* (UTCM 1486, 1491, 1500) the condyle is more symmetrical about this indentation, and in *Vassallia* (FMNH P14424) the indentation itself is dramatically reduced. In ventral view, the condyle appears to be somewhat wider transversely in

pampatheres and glyptodonts than in armadillos (as represented by *Proeutatus* and *Euphractus*). The measurements reflect this, with the ratio of width to length greater than or equal to 1.5 in *Holmesina floridanus*, *Holmesina septentrionalis*, *Holmesina occidentalis*, *Vassallia*, and *Propalaehoplophorus*, and substantially less in *Proeutatus* and *Euphractus* (Tables 1 and 2).

The surface of the exoccipital immediately medial to the condyles is deeply impressed by a fossa that extends anteromedially almost to the front of the foramen magnum's ventral rim (Fig. 13A). Based on comparison with other placental mammals (see *Homo*, *Clemente 1985*; *Canis*, *Evans & Christiansen 1979*; in which the condyles are much smaller and shallower) we identify this depression as the site of insertion for the alar ligaments extending forward from the dens of the axis. It is not at all clear why these ligaments would be so large in *Holmesina floridanus*, but they appear similarly enlarged in other pampatheres, based on the presence of this fossa in *Holmesina septentrionalis* (UF 234224), *Holmesina occidentalis* (ROM 3881), and *Vassallia* (FMNH P14424). No such depression is described in Wible and *Gaudin (2004)*, but we have subsequently examined specimens of *Euphractus* (UTCM 1486, 1491, 1500) in which a small, circular depression is present in this area. A similar circular depression is also observed in *Proeutatus* (FMNH P13197), whereas *Propalaehoplophorus* (YPM VPPU 15007) appears to have fossa similar in size to that of pampatheres, but much shallower.

The vertical portion of the exoccipital bears a strongly marked, transversely elongated depression immediately dorsal to the occipital condyle (Fig. 14). This is the dorsal condyloid fossa of *Wible & Gaudin (2004)*. Dorsal to this depression, the exoccipital is nearly flat. As noted above, the exoccipital forms nearly the entire rim of the foramen magnum, the supraoccipital only contributing a small exposure on the dorsalmost point of the opening. The rim is irregularly shaped due to a small convexity located at roughly the midpoint of its height, the nuchal tubercle. The nuchal tubercle is developed to a similar degree in *Proeutatus* (FMNH P13197) and *Euphractus* (*Wible & Gaudin, 2004*), but is less prominent in *Propalaehoplophorus* (YPM VPPU 15007). The latter also has a broader, transversely ovate foramen magnum, in contrast to the taller, more triangular shaped foramen in *Proeutatus* and *Holmesina floridanus*.

The supraoccipital is a broad, hemispherical plate that extends from its ventral contacts with the squamosal, mastoid and exoccipital to its dorsal termination at the nuchal crest, where it is presumably fused to the parietal, as in *Euphractus* (*Wible & Gaudin, 2004*). As in both *Euphractus* (*Wible & Gaudin, 2004*) and *Proeutatus* (*Scott, 1903*), the nuchal crest is posteriorly convex laterally and posteriorly concave in the midline. This shape is broadly shared among euphractine and eutatine armadillos, pampatheres, and early glyptodonts, extending all the way back to the oldest known cingulate skull, that of the Eocene taxon *Utaetus* (Barrancan SALMA; *Simpson, 1948*; *Gaudin & Croft, 2015*). In *Holmesina floridanus*, there are prominent, raised tubercles just behind the most posterior point of curvature on the nuchal crest. Low, broadly rounded ridges extend ventromedially from the tubercles toward the foramen magnum. The central region of the supraoccipital between these elevations has a gently concave surface, interrupted in the midline by a very weakly developed external occipital crest (Fig. 14). The supraoccipital is very similar in

*Vassallia* (FMNH P14424). In *Holmesina septentrionalis* (UF 234224) and *Propalaehoplophorus* (YPM VPPU 15007), the nuchal crest is more rugose, and the external occipital crest is more prominent, the latter also the case in *Euphractus* (*Wible & Gaudin, 2004*). *Proeutatus* lacks the raised tubercles present in the other taxa, it has a large pair of mastoid foramina that perforate the supraoccipital, and it has a characteristic nuchal crest that is very tall to the point of being slightly recurved anteriorly in lateral view (*Scott, 1903*; FMHH P13197).

The overall shape of the occiput in *Holmesina floridanus* is rather tall and narrow, almost triangular, with its maximum depth and transverse width (measured at the base of the supraoccipital) nearly equivalent (Tables 1 and 2). This is also the case in *Holmesina septentrionalis* (UF 234224), whereas in *Vassallia* (FMNH P14424), *Propalaehoplophorus* (YPM VPPU 15007), *Proeutatus* (FMHH P13197), and *Euphractus* (UTCM 1491), the occiput is lower, broader and more semicircular in shape, with a width/depth ratio $\geq 1.2$.

## Mandible

The mandible of *Holmesina floridanus* (Fig. 15; MML = 182–200 mm; Tables 3 and 4) is smaller than that of *Holmesina septentrionalis* [both *Simpson (1930)* and *James (1957)* report MML of 240 mm] and *Holmesina occidentalis* (MML > 268 mm in ROM 4955; Table 3). Proportions are very similar to *Holmesina occidentalis*, with a very similar relative depth of the horizontal ramus (Table 2), whereas the horizontal ramus appears slightly deeper in *Holmesina septentrionalis* (*Simpson, 1930*; *James, 1957*; *Edmund, 1985b*). Like *Holmesina occidentalis* (ROM 4955), UF 224450 has two mental foramina that open on the external surface of the horizontal ramus in the symphyseal region (ventral to mf3 and mf4, respectively; the mental foramen accommodates the mental nerves and vessels of the rostral lower jaw area, *Evans & Christiansen, 1979*). Unlike *Holmesina occidentalis*, both mental foramina are associated with grooves in the surface of the mandible. The more anterior foramen empties into two closely set, parallel, anterodorsally directed grooves, and indeed the foramen itself is partially constricted into an upper and lower opening. The groove emerging from the posterior mental foramen travels posteroventrally. For *Holmesina septentrionalis*, *Simpson (1930)* illustrates four foramina of varying sizes in the external surface of the mandible anterior to the level of mf4, whereas *James (1957)* describes a single mental foramen beneath mf3. It is not clear if all four of Simpson's openings are mental foramina, or if one or more are nutritive foramina that he chose to illustrate.

The anteroventral edge of the symphysis in *Holmesina floridanus* forms roughly a 27° angle with the toothrow (Fig. 15C). This appears to be similar to the angle in *Holmesina occidentalis* (ROM 4955), but somewhat more acute than in *Holmesina septentrionalis* [roughly 30° as measured in *Simpson (1930*, fig. 4) and *Edmund (1985b*, fig. 6)]. The posteriormost point of the symphysis extends just below the ventral edge of the horizontal ramus in medial view, as in other *Holmesina*, and the anteriormost point forms a very short triangular extension in front of mf1, marked by two small foramina on its dorsal surface. The length of this short mandibular spout is a little longer than the mesiodistal diameter of the mf1 alveolus (spout = 6.3 mm, mf1 alveolus = 6.0 mm), whereas in

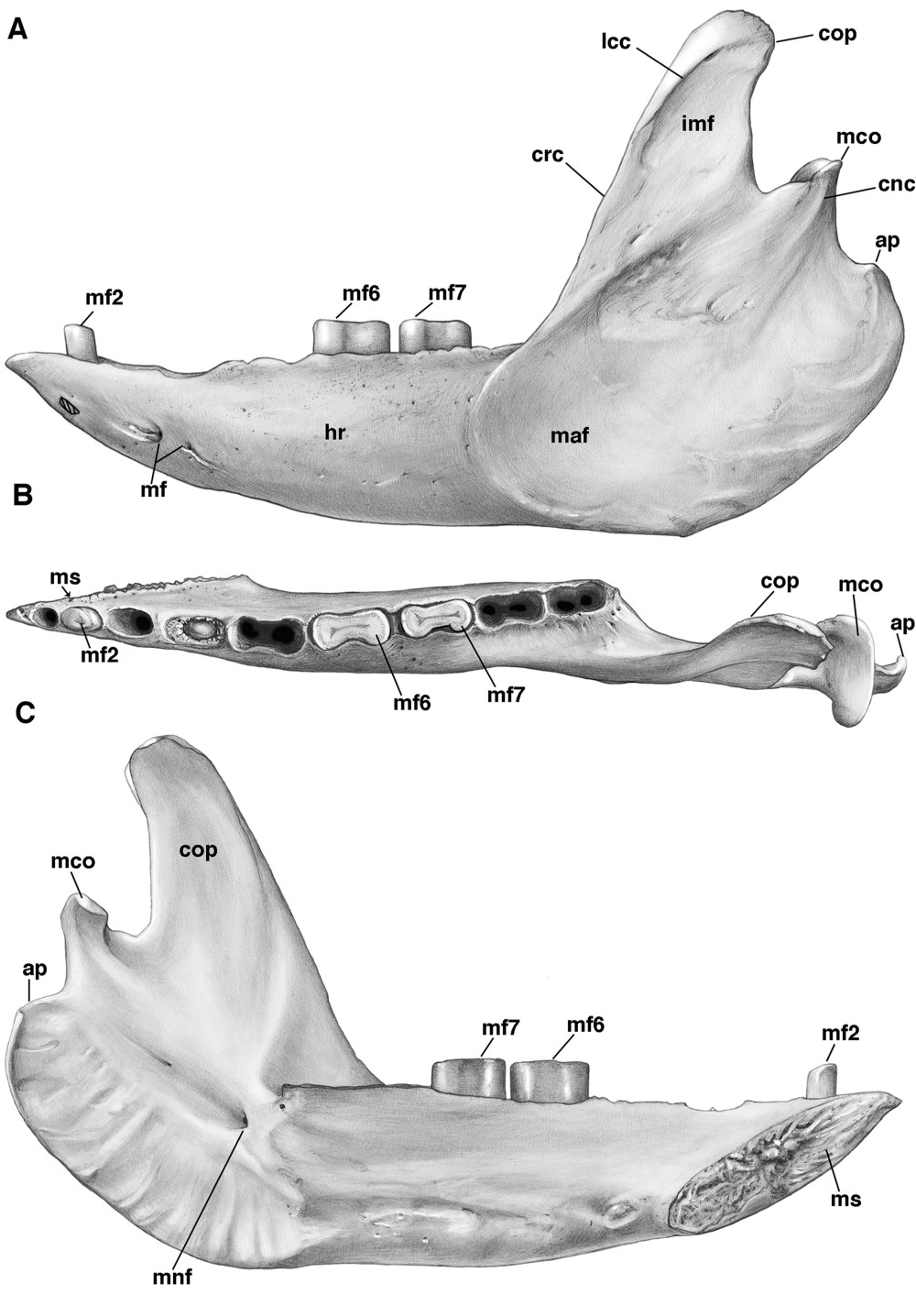

**Figure 15 Left mandible of *Holmesina floridanus* (UF 224450) in (A) lateral; (B) occlusal; and (C) medial views. ap**, angular process; **cnc**, condyloid crest; **cop**, coronoid process; **crc**, coronoid crest; **hr**, horizontal ramus of mandible; **imf**, intermuscular fossa; **lcc**, lateral coronoid crest; **mf2**, second lower molariform tooth; **mf6**, sixth lower molariform tooth; **mf7**, seventh lower molariform tooth; **maf**, masseteric fossa; **mco**, mandibular condyle; **mf**, mental foramen; **mnf**, mandibular foramen; **ms**, mandibular symphysis. Scale bar = 5 cm.

Table 3 Mandibular measurements for *Holmesina floridanus* and related taxa.

| Measurement description | Holmesina floridanus UF 224450 | Holmesina occidentalis ROM 4955 | Vassalia maxima FMNH P14424[a] | Propalaeho-plophorus australis MLP 16–15[b] | Proeutatus oenophorus FMNH P13197 | Euphractus sexcinctus UTCM 1491 |
|---|---|---|---|---|---|---|
| Maximum mandibular length (MML) | 200.5 | 268* | 180* | 139 | 98.3 | 93.0 |
| Max dp of horizontal ramus | 40.0 [0.20] | 52 [0.19] | 47.5[a] [0.26] | 37 [0.27] | 16.1 [0.16] | 13.5 [0.15] |
| Max dp of ascending ramus | 120.5 [0.60] | 155 [0.58] | 132.0[a] [0.73] | 98 [0.71] | 53.8 [0.55] | 49.5 [0.53] |
| Condyle, max wd | 23.4 | – | 31.2[a] | – | 9.0 | 9.9 |
| Condyle, max ln | 7.0 | – | 11.0[a] | – | 4.7 | 5.9 |
| Ratio of condyle wd to ln | 3.34 | – | 2.84 | – | 1.91 | 1.68 |
| Height of condyle above toothrow | 39.3 [0.20] | 84 [0.31] | 60 [0.33] | 59 [0.42] | 37.3 [0.38] | 41.7 [0.45] |
| Symphysis ln | 59.0 [0.29] | 79* [0.29] | 54.0[a] [0.30] | 41 [0.29] | 23.5 [0.24] | 25.7 [0.28] |
| Coronoid process, anteroposterior ln at base | 31.0 | 40 | 33 | 31 | 12.4 | 9.8 |
| Coronoid process, maximum dp | 42.0 | – | 50 | 31 | 16.2 | 15.5 |
| Ratio of coronoid process dp to ln | 1.35 | – | 1.52 | 1.0 | 1.31 | 1.58 |
| Mesiodistal ln/max wd of lower molariforms in *Holmesina floridanus* UF 224450 | mf1: 5.8/5.1**, mf2: 7.5/5.6, mf3: 11.7/7.1**, mf4: 14.9/7.7**, mf5: 17.3/9.3**, mf6: 16.9/8.6, mf7: 16.2/8.0, mf8: 16.7/8.5**, mf9: 12.3/7.6** [Mean ratio of lower molariform ln/wd: 1.72] | | | | | |

Notes:
All measurements reported in millimeters (mm); those reported to the nearest 10th of a millimeter are direct measurements, those rounded to the nearest millimeter are taken from literature sources or from photographs taken by TJG. Numbers in square brackets are scaled to maximum mandibular length (MML).
–, data unavailable; dp, dorsoventral depth; ln, anteroposterior length; Max, maximum; Min, minimum; n, data not applicable; wd, transverse width.
* Estimated due to breakage.
** Estimated from alveolus diameter.
[a] Data from *De Iuliis & Edmund (2002)*.
[b] Estimated from photograph, *Vizcaíno, De Iuliis & Bargo (1998*, Fig. 2*)*.

*Edmund's (1985b*, fig. 6*)* illustration of *Holmesina septentrionalis* the spout is shorter than mf1.

The masseteric fossa of *Holmesina floridanus* (UF 224450) is broad, its anterior terminus marked by a low crest that connects the anteriormost edge of the angular process with the ventralmost edge of the coronoid process (Fig. 15A). This crest continues posteriorly across the lateral surface of the coronoid base. This makes the masseteric fossa of *Holmesina floridanus* deeper than that of *Vassallia*, but shallower than that of *Holmesina occidentalis* (*Vizcaíno, De Iuliis & Bargo, 1998*). There are distinct depressions on either side of this upper masseteric crest. The depression above the crest covers most of the lateral surface of the coronoid process, and is bounded anteriorly by the coronoid crest, i.e., the thickened anterolateral margin of the coronoid process. The coronoid crest is continuous dorsally with a distinct crest that crosses the lateral surface of the coronoid process proximal to its tip, which we are designating the lateral coronoid crest. This lateral coronoid crest is found in all euphractine and eutatine armadillos, as well as pampatheres (*Gaudin & Wible, 2006*: char 21[1], an unambiguous synapomorphy of Node A). Because this lateral coronoid depression lies between the coronoid and lateral coronoid

**Table 4 Mandibular measurements for additional specimens of *Holmesina floridanus*.**

| Measurement description | *Holmesina floridanus* UF 223813 | UF 275497 | UF 275498 | UF 285000 | UF 293000 |
|---|---|---|---|---|---|
| Maximum mandibular length (MML) | 182.2 | 187* | 191* | – | 185* |
| Max dp of horizontal ramus | 38* [0.21] | 41.0 [0.22] | 40.4 [0.21] | 44 | 39.3 [0.21] |
| Max dp of ascending ramus | 106.5 [0.58] | 120.3 [0.64] | – | – | 113* [0.61] |
| Condyle, max wd | 24.0 | 21.4 | 18.3 | 23.0 | 23.8 |
| Condyle, max ln | 7.4 | 9.5 | 7.5 | 10.0 | 9.1 |
| Ratio of condyle wd to ln | 3.24 | 2.25 | 2.44 | 2.30 | 2.62 |
| Height of condyle above toothrow | 45.6 [0.25] | 43.0 [0.23] | – | 30.0 | 37.0 [0.20] |
| Symphysis ln | 54.3 [0.30] | – | 50.3 [0.26] | – | 58.4 [0.32] |
| Coronoid process, anteroposterior ln at base | 31.2 | 33.3 | – | – | 31.7 |
| Coronoid process, maximum dp | 26.6 | 47.7 | – | – | 36.0 |
| Ratio of coronoid process dp to ln | 0.85 | 1.43 | – | – | 1.14 |
| Mesiodistal ln/max wd of lower molariforms in *Holmesina floridanus* UF 223813 | mf1: 5.0/4.8, mf2: 7.2/5.7, mf3: 8.2/6.0, mf4: 11.5/6.8, mf5: 16.0/8.6, mf6: 17.0/8.4, mf7: 15.2/8.0, mf8: 13.6/7.0, mf9: 10.7/5.9 [Mean ratio of lower molariform ln/wd: 1.65] | | | | |

**Notes:**
All measurements reported in millimeters (mm). Numbers in square brackets are scaled to maximum mandibular length (MML).
–, data unavailable; dp, dorsoventral depth; ln, anteroposterior length; Max, maximum; Min, minimum; wd, transverse width.
* Estimated due to breakage.

crests, which serve as insertion points for the temporalis musculature in *Euphractus* (*Smith & Redford, 1990*; *Wible & Gaudin, 2004*) and presumably in *Holmesina* as well (*Vizcaíno, De Iuliis & Bargo, 1998*), and the upper limit of the masseter, we are labeling this area the "intermuscular fossa." The intermuscular fossa is very similar in size and shape in *Holmesina floridanus* and *Holmesina occidentalis* (ROM 4955), ovate and elongated along an anteroventral to posterodorsal axis. In *Holmesina septentrionalis,* it appears to be narrower anteroposteriorly and more elongated posterodorsally (*Cahn, 1922*; *Simpson, 1930*).

There is also a weak, ovate depression below the upper masseteric crest in UF 224450, its long axis oriented in an anteroventral to posterodorsal direction, bounded posteriorly by the lateral condyloid crest. It is unclear if this area serves as part of the attachment for the masseter, although low ridges crossing its surface suggest that it does, and *Smith & Redford (1990)* show that the comparable area in *Euphractus* is covered by the masseter muscle.

The coronoid process itself is generally triangular in *Holmesina floridanus,* but varies rather dramatically in its proportions. The ratio of maximum height to basal anteroposterior length ranges from 0.85 to 1.43, easily encompassing *Holmesina septentrionalis* [as illustrated by *James (1957)* and *Edmund (1985b)*, the ratio is 1.25 or 1.23, respectively] within this range. The process appears to be somewhat more tapered distally in *Holmesina floridanus* than in *Holmesina septentrionalis*. A complete coronoid is not preserved in the specimen of *Holmesina occidentalis* illustrated by *Vizcaíno, De Iuliis & Bargo (1998),* but the preserved portion is more parallel sided than tapered distally, resembling more closely the condition in *Holmesina septentrionalis*. Although the posterior edge of the coronoid process is slightly inclined posterodorsally in both

*Holmesina floridanus* and *Holmesina septentrionalis,* the former taxon possesses an additional small but distinct posterior hook at its distal terminus (Fig. 15), a feature lacking in the latter species (*James, 1957*; *Edmund, 1985b*). Similar to the coronoid crest described above, the coronoid process of *Holmesina floridanus* carries a thickened anterior edge on its medial face, as well (Fig. 15C). This medial crest connects to a second crest near the base of the coronoid. This low crest traces a posteroventrally curved arc, terminating at a point above the space between the last molariform tooth and the mandibular foramen. Anteroposteriorly, the medial surface of the coronoid process is gently concave. The base of the coronoid covers the posterior half of mf8 in lateral view, and hides mf9 entirely, as in other pampatheres (*De Iuliis & Edmund, 2002*).

When viewed laterally or medially (Fig. 15), the condylar process of *Holmesina floridanus* is very short and triangular, closely resembling that of *Holmesina septentrionalis* (*Simpson, 1930*; *James, 1957*; *Edmund, 1985b*) and *Holmesina occidentalis* (*Vizcaíno, De Iuliis & Bargo, 1998*). As noted above, there is a single, short condyloid crest on the lateral side of the condylar process. There are two such crests on the medial side. All are short and extend in an anteroventral direction—the lateral crest is straight, whereas the medial crests are curved in an anteriorly concave fashion. *Edmund (1985b)* illustrates three medial condyloid crests in his specimen of *Holmesina septentrionalis*. The condyle itself is ovate, very broad transversely, and narrow anteroposteriorly, its width two to three times its length (Tables 3 and 4). Its surface is flat anteroposteriorly, but slightly concave transversely, and inclined posterodorsally in lateral view, as it is in *Holmesina occidentalis* (ROM 4955). At its medial extremity, the condyle of *Holmesina floridanus* hooks sharply anteriorly at nearly a right angle, forming a tall medial wall to an ovate fossa. This fossa extends nearly to the midpoint of the condyle, lying anterior to the articular surface. It likely served as the site of insertion for the lateral pterygoid muscle, since the muscle attaches to this region in the extant *Euphractus* (*Wible & Gaudin, 2004*). The condyle in *Holmesina floridanus* is strongly elevated, located high above the level of the toothrow, like it is in *Holmesina septentrionalis* (*Edmund, 1985b*) and *Holmesina occidentalis* (*Vizcaíno, De Iuliis & Bargo, 1998*). Curiously, the condyle of pampatheres is noticeably less elevated than that of *Propalaehoplophorus*, *Proeutatus*, or *Euphractus* (Table 3).

As in the other *Holmesina*, the angular process of *Holmesina floridanus* extends only a short distance posterior to the base of the condylar process, but forms a very broad, posteroventrally convex curved structure that reaches anteriorly nearly to the midpoint of the last molariform tooth (Fig. 15). It extends, at its lowest point, slightly below the ventral edge of the horizontal ramus. The outer surface, part of the very large masseteric fossa, is only slightly convex anteroposteriorly. Similarly, its inner surface is only slightly concave anteroposteriorly, nearly flat dorsoventrally, but strongly scalloped near its margin by the insertion of the medial pterygoid muscle, which attaches to this same region in *Euphractus* (*Wible & Gaudin, 2004*) and other placental mammals (e.g., *Canis*, Evans and Christiansen 1979; *Homo*, *Clemente 1985*). Again, this morphology is virtually identical to that of other *Holmesina* (*Simpson, 1930*; *Edmund, 1985b*; *Vizcaíno, De Iuliis & Bargo, 1998*). The mandibular foramen, for the inferior alveolar nerves and vessels

(*Wible & Gaudin, 2004*), lies just above the inner medial pterygoid fossa, just behind and below the level of the last molarifom and positioned directly above the most ventral portion of the angular process. In *Simpson's (1930)* illustration of *Holmesina septentrionalis*, the foramen is located somewhat more anterior and much further ventrally, but this may be due to postmortem damage. In *Edmund's (1985b)* illustration of the same taxon, there appear to be two mandibular foramina, one in a position like that of Simpson's specimen, the other in roughly the same position as in *Holmesina floridanus* (UF 224450).

There is more variation in lower tooth counts than upper tooth counts among crown-group cingulates (*Gaudin & Wible, 2006*)—e.g., *Proeutatus* (*Scott, 1903*) and *Euphractus* (*Wible & Gaudin, 2004*) both have 10 lowers, and *Propalaehoplophorus* has only eight (*Scott, 1903*). *Holmesina* has nine, as in other pampatheres (*Simpson, 1930*; *Edmund & Theodor, 1997*; *De Iuliis & Edmund, 2002*), and this condition is optimized as a synapomorphy of pampatheres plus glyptodonts in *Gaudin & Wible's (2006*; Node 8*)* phylogenetic analysis.

Only three teeth are preserved in UF 224450: mf2, mf6, and mf7 (Fig. 15). In addition, there appears to be a conical, unerupted mf4, but this is likely a pathological condition, as indicated not only by the shape and position of the tooth itself, but by the spongy bone that occupies much of the volume of the alveolus. The shape of the remaining teeth can only be inferred from the outline of their alveoli. There are lower teeth preserved in other FLMNH *Holmesina floridanus* specimens, though many can only be observed in lateral view because of preservation and degree of preparation. UF 223813 preserves all nine lower molariforms (Table 4), UF 275497 preserves mf1, mf3–7, and mf9, UF 275498 preserves mf1–7, and UF 285000 preserves mf2, mf4–5, and mf7–8. The first three lower molariforms in *Holmesina floridanus* are ovate mesiodistally, with their long axis rotated to a slightly mesiolingual to distolabial orientation. The fourth molarifom is pathological in UF 224450. The alveolus shape indicates a reniform outline, with a slight labial indentation, but there is no visible external groove on the teeth in the other *Holmesina floridanus* specimens, where the tooth takes on almost a rectangular shape, or is perhaps weakly bilobate, in contrast to the reniform mf4 (with a lingual groove) of other *Holmesina* (see below). The remaining lower teeth in *Holmesina floridanus* (mf5–9) appear to be strongly bilobate in outline. The first and last of these (i.e., mf5 and mf9) are substantially shorter mesiodistally than the intervening three teeth in between.

The tooth outlines and proportions in *Holmesina septentrionalis* are quite similar (*Simpson, 1930*; *Edmund, 1985b*), although both mf3 and mf4 are clearly reniform (concave lingually) in this species, in clear contrast to *Holmesina floridanus*, and even mf2 has a lingual groove as illustrated by *Edmund (1985b)*. *Holmesina occidentalis* (*Vizcaíno, De Iuliis & Bargo, 1998*) is even more similar to *Holmesina floridanus*, lacking the reniform anterior teeth of *Holmesina septentrionalis*, although mf5 in this taxon is as large as mf6–8, contrasting with its reduced length (relative to mf6–8) in other *Holmesina*. *Simpson (1930)* notes that mf4 is bilobate in both *Pampatherium* and *Kraglievichia*, and is clearly more elongated mesiodistally than mf3, both features

contrasting with the condition in *Holmesina*. *De Iuliis & Edmund (2002)* describe and illustrate an mf4 for *Vassallia* that resembles that of *Pampatherium* and *Kraglievichia*, whereas *Castellanos (1937, 1946)* attributes a *Holmesina*-like morphology to this taxon. *De Iuliis & Edmund (2002)* suggest the discrepancy may be due to individual variation, and *Edmund & Theodor (1997*, p. 230*)* note that the shape of mf4 in *Scirrotherium* varies "from reniform to elongate-elliptical." In both *Pampatherium* and *Vassallia* (*Simpson, 1930*; *Castellanos, 1937, 1946*), mf5 is reniform rather than bilobate, as it is in *Holmesina*. In *Scirrotherium*, mf5 is described as bilobate but illustrated as reniform (*Edmund & Theodor, 1997*, fig. 14.2). The long axis of mf5–7 in *Vassallia* is obtusely V-shaped with a lingual vertex (FMNH P14424; see illustration in *De Iuliis & Edmund, 2002*), and is rotated so that the posterior lobe extends further labially than the anterior lobe. *Simpson (1930)* illustrates a similar if less well developed condition for mf6, mf7, and mf8 in *Kraglievichia*, and mf5, mf6, and mf8 in *Pampatherium*, whereas in *Holmesina* and *Scirrotherium* (*Edmund & Theodor, 1997*), the long axis of the posterior molariforms is essentially straight.

The lower teeth of Santacrucian glyptodonts (*Propalaehoplophorus* and *Eucinepeltus*; *Scott 1903*) are reminiscent of those in pampatheres in some respects, with the first and second lower molariforms (likely homologous to mf2 and mf3 of pampatheres) ovate or slightly reniform in outline, and the third (=mf4 of pampatheres) clearly reniform, but the remaining lower molariforms show the distinctive trilobate shape characteristic of glyptodonts (*Hoffstetter, 1958*). The lower tooth outlines in *Proeutatus* (FMNH P13197; *Scott, 1903*) also display some pampathere-like features. The anterior teeth (mf1–3) are ovate, but mf4–8 are vaguely heart shaped, with a shallow groove followed by a sharp keel on the lingual surface, with a stronger groove on the labial edge. The long axes of mf4–8 are tilted somewhat posterolabially, as described above for *Vassallia*. The mf9 in *Proeutatus* is weakly bilobate, like that of pampatheres, but the distal lobe is the broader of the two, whereas the mesial lobe is broader in pampatheres. The tooth outlines in *Euphractus* are like those of most other armadillos, i.e., uniformly circular or ovate in cross section (*Wible & Gaudin, 2004*; *Gaudin & Wible, 2006*).

As was the case with the upper dentition, the preserved teeth in UF 224450 possess a raised central region of osteodentine surrounded by more typical orthodentine (*Ferigolo, 1985*; *Kalthoff, 2011*). In mf2, the osteodentine core takes on the shape of a very narrow oval aligned with the long axis of tooth's outline. The osteodentine in mf6 and mf7 is mostly linear, expanding into a short "Y" at its mesial and distal ends, as was the case with the posterior upper molarifoms. The same condition is present in other pampatheres (*Simpson, 1930*; *De Iuliis & Edmund, 2002*; *Kalthoff, 2011*), whereas in glyptodonts the central osteodentine core bear multiple lateral branches (*Scott, 1903*; *Gillette & Ray, 1981*; *Ferigolo, 1985*; *Kalthoff, 2011*), and in *Proeutatus* the osteodentine core takes the form of an obliquely oriented oval (*Scott, 1903*). *Euphractus* and other cingulates lack this osteodentine core, the central regions of their teeth occupied instead by a variably vascularized "modified orthodentine" (*Ferigolo, 1985*; *Gaudin & Wible, 2006*; *Kalthoff, 2011*).

The occlusal surface of the first three lower molariforms in *Holmesina floridanus* is quite variable. In some instances the teeth are nearly flat, contrasting with the more beveled crowns of the anterior upper molariforms—e.g., in mf1 of UF 223813 and 275497 (L only), and mf2 and mf3 of UF 275498. Other are beveled (some only weakly, e.g., mf2 of UF 224450; Fig. 15), with a small, mesioventrally sloping anterior facet, usually occupying less than ¼ of the occlusal surface, and the remaining distal facet sloping distoventrally. As noted above in the description of the premaxillary teeth, there does not appear to be any obvious correlation between variation in wear surface morphology of these beveled anterior molariforms and the chronological age of the specimens. The remaining molariforms (mf4–9) have single, flat occlusal surfaces, as was the case with the upper posterior teeth. These occlusal surface are not horizontal, but inclined distoventrally, giving adjacent tooth crowns a stair-step appearance, as described above for the upper dentition. These occlusal surface patterns are, as far as can be determined, nearly identical in other pampatheres. *Proeutatus* also has beveled wear on the anterior teeth and flat surfaces on the posterior teeth (flat on mf8–10 in FMNH P13197; see also *Scott, 1903*; *Gaudin & Wible, 2006*). In glyptodonts, all teeth are worn flat, whereas in *Euphractus* and other armadillos, all teeth show beveled wear (*Wible & Gaudin, 2004*; *Gaudin & Wible, 2006*).

The long axis of mf2 is oriented anteriorly in lateral view in *Holmesina floridanus*, and nearly vertical or perhaps slightly posteriorly in mf6 and mf7 (UF 224450). In anterior view, mf2 tilts somewhat lingually, like the anteriormost upper teeth (although this may be a preservational artifact, since the anterior alveoli appear to slant labially). Like their counterparts in the upper dentition, mf6 and mf7 are implanted vertically, whereas the alveolus of mf9 seems to clearly be inclined lingually, opposite its counterpart in the upper dentition. The latter condition (i.e., posterior lower molariforms inclined lingually) is identified as a xenarthran synapomorphy by *Gaudin (2004)*.

## DISCUSSION

The present study represents the most detailed and extensively illustrated description of a pampathere skull published to date. This is not to say that everything worthy of note is now known about the cranial osteology of *Holmesina floridanus*. As noted in the descriptive text above, we have yet to identify any ear ossicles, or any ecto- or entotympanic elements, in whole or in part (if the latter indeed exists in pampatheres, as it does in many other cingulates—see *Patterson, Segall & Turnbull, 1989*). CT-scanning of existing, but still unprepared specimens of *Holmesina floridanus* might allow for the digital recovery of these small, often delicate and loosely attached structures. Such structures are frequently lost through more traditional mechanical preparation techniques, even if they exist and remain with the skull, either embedded in matrix, or, in the case of the ossicles, trapped in the vestibule of the inner ear. CT scans might also yield information on endocranial osteology, e.g., on the delicate and hard-to-prepare nasal turbinate elements and paranasal sinuses, as has been done for the glyptodont *Neosclerocalyptus* (*Fernicola et al., 2012*) and the extant armadillo *Dasypus* (*Billet et al., 2017*). Scanning would also allow for the reconstruction of soft tissues, especially the brain

and associated cranial nerves and endocranial vasculature, as has recently been done (in part) for the pampathere *Pampatherium humboldti* (*Tambusso & Fariña, 2015*). Producing and describing detailed CT scans of the skull in *Holmesina floridanus* were deemed beyond the scope of the present study, but are planned for the future. In addition, there is extensive postcranial and carapace material for this species that was not considered in this investigation, but will be the subject of planned future work.

It is particularly fortuitous that this description centers on *Holmesina floridanus*, a taxon represented by such abundant and well preserved material, including at least 10 complete or nearly complete skulls from two sites of similar age in central Florida. As noted by *Wible & Gaudin (2004)* and *De Iuliis et al. (2014)*, and many others, all too often descriptions of fossil species are based on single (or even incomplete) specimens. Whereas this is often due to the limitations of the available material, it makes it difficult to account for intraspecific variation, to distinguish between species level distinctions and sexual dimorphism (*McDonald, 2006*), or to assess the reliability of systematic characters based on fossil taxa. The present study, like other recent detailed analyses of xenarthran skull morphology (*Wible & Gaudin, 2004*; *Gaudin, 2011*; *McAfee & Naples, 2012*; *De Iuliis et al., 2014*; *Hautier et al., 2014*; *Gaudin et al., 2015*), has revealed substantial variation in a variety of cranial features in *Holmesina floridanus*. These features include the number, size and/or position of a variety of cranial foramina (anterior palatal foramen, maxillary foramen, minor palatine foramina, foramen for frontal diploc vein, ethmoid foramen, transverse canal foramen, foramina for rami temporalis, suprameatal foramen, hypoglossal foramen); the presence, size and shape of various processes (anteroventral process on premaxilla, lacrimal tubercle, ventral zygomatic process, postorbital process of jugal, orbito-auricularis crest, medial pterygoid process, circular boss on lateral wall of promontorium, medial shelf of petrosal, coronoid process of mandible) or depressions (digastric fossa, tensor tympani fossa, fossa incudis); and the shape of other cranial (proportions of nasal bone, shape of anterior margin of premaxilla, shape of naso-frontal and jugal/squamosal sutures, shape of external nares and occipital exposure of mastoid) or dental features (e.g., outline of M4 and M5, shape of wear facets on M1).

Whereas the present study reveals a significant amount of intraspecific cranial variation in *Holmesina floridanus*, it has also produced a long list of features that affirm previously hypothesized systematic relationships between this and other purportedly related taxa. Among these are features that are diagnostic of the taxon itself. The only diagnostic feature provided in the original diagnosis of the species by *Robertson (1976)* was the shape and orientation of the fourth upper molariform, and, as noted above, the shape of this tooth is variable among specimens of *Holmesina floridanus*. *Edmund (1987)* distinguished this taxon based almost exclusively on size. *Hulbert & Morgan (1993)* conducted a more extensive analysis, looking at the taxonomic implications of size variation but also a series of qualitative postcranial and dental features for *Holmesina* specimens from Florida only, but they did not list any cranial characteristics that served to distinguish *Holmesina floridanus* from the younger *Holmesina septentrionalis*. The present study recognizes at least 11 distinct, meristic cranial features that may be diagnostic for *Holmesina floridanus*

**Table 5** Listing of cranial features with potential systematic value (i.e., diagnostic features or putative synapomorphies) identified in the description from the present study.

*Holmesina floridanus*: (1) ovate shape of Mf3; (2) absence of vomer/premaxilla contact within nasal cavity; (3) lacrimal foramen situated on the antorbital ridge; (4) medial pterygoid exposure that fails to reach nasopharyngeal roof; (5) inflated pterygoid; (6) presence of a bony bridge connecting the tympanohyal and crista interfenestralis; (7) presence of a raised circular boss on the lateral surface of the promontorium; (8) elongate, narrow basioccipital; presence of well-developed rectus capitis fossae and pharyngeal tubercle on basioccipital; (9) distinct grooves emerging anteriorly from mental foramina; and (10) mandibular spout with anteroposterior ln > mf1; (11) rectangular shape of mf4.

Genus *Holmesina*: (1) nasals become narrower posteriorly; (2) maxillary/premaxillary suture M-shaped in ventral view; (3) maxillary/palatine suture U-shaped in ventral view; (4) presence of prominent lateral maxillary ridge and deep antorbital fossa; (5) reniform Mf4 and bilobate mf5; (6) lack of orbital exposure of palatine; (7) ethmoid foramen entirely within frontal, lacking orbitosphenoid participation in rim; (8) no orbitosphenoid participation in rim of sphenopalatine foramen; (9) fenestra cochleae very low and wide, ratio of wd/dp > 3; (10) triangular stylohyal fossa with distally expanded tympanohyal; (11) strong medial flange of petrosal marked by pits and ridges; (12) low stapedial ratio (<1.4); and (13) ventrally displaced internal acoustic meatus.

Pampatheres: (1) nasal ln > 45% of GSL; (2) presence of median anteroventral processes on premaxilla; (3) incisive foramina open ventrally into single, deeply recessed, midline fossa; (4) infraorbital canal elongate, extending from level of Mf6 to Mf8; (5) reniform anterior molariforms and bilobate posterior molariforms; (6) posterior molariforms with linear, unbranched core of osteodentine; (7) partially closed (anteriorly) upper toothrows; (8) teeth wear in stairstep fashion in lateral view; (9) triangular facial process of lacrimal; (10) triangular glenoid fossa (apex lateral); (11) no horizontal portion of jugal/squamosal suture; (12) ridged, anteroposteriorly expanded ventral zygomatic process formed by maxilla and anterior jugal [mostly the latter]; (13) zygomatic process of squamosal increases in dp anteriorly; (14) reduced midline crest on endocranial exposure of orbitosphenoid; (15) ventral flange of basisphenoid forms portion of the lateral wall of the nasopharynx; (16) elongate groove on petrosal for greater petrosal nerve; (17) broad crista interfenestralis of petrosal; (18) enormously enlarged paroccipital process of petrosal; (19) caudal tympanic process of petrosal forms posterior wall to stapedial fossa; (20) large epitympanic wing of petrosal (as in *Euphractus*), forms shelf above rostral process of promontorium; (21) groove for auricular branch of vagus nerve between caudal tympanic process and triangular shelf [=roof of postpromontorial sinus]; (22) sharp, narrow transverse crest within internal acoustic meatus; (23) low rounded ridges subdivide subarcuate fossa; (24) prefacial commissure enlarged, bulbous; (25) crista petrosa rounded, divided by groove into medial and lateral parts; (26) paracondylar process of exoccipital hooked medially; (27) ventral surface of exoccipital convex transversely; (28) mandibular condyle less elevated above toothrow than glyptodonts, *Proeutatus*, *Euphractus*; and (29) coronoid process covers mf9 and part of mf8 in lateral view.

Pampatheres plus glyptodonts: (1) presence of semicircular notch in anterolateral edge of premaxilla; (2) dp of external nares ≥ wd; (3) teeth with essentially linear core of osteodentine; (4) anterior molariforms slanted lingually in posterior view, posterior molariforms slanted labially; (5) narrow, U-shaped postpalatal margin (also in *Gaudin & Wible (2006)*); (6) pterygoid processes form thickened, rugose bosses (also in *Gaudin & Wible (2006)*); (7) pterygoid participation in hard palate; (8) lacrimal foramen positioned on facial process of lacrimal; (9) lacrimal foramen situated within distinct fossa; (10) presence of an enlarge ventral zygomatic process near anterior terminus of zygomatic arch; (11) sphenopalatine foramen in common fossa with sphenorbital fissure (also in *Gaudin & Wible (2006)*); (12) raised ridge along squamosal/parietal suture; (13) posterior zygomatic root directed laterally (also in *Gaudin & Wible (2006)*); (14) postglenoid foramen visible in ventral view (also in *Gaudin & Wible (2006)*); (15) broad triangular shelf [=roof of postpromontorial sinus]; (16 and 17) enlarged paroccipital process of petrosal and paracondylar process of exoccipital; (18) well-developed external occipital crest; (19) anterior portion of occipital condyle extends lateral to dorsal portion in ventral view; (20) nine lower molariforms present; (21) ratio between maximum depth of mandibular horizontal ramus vs. MML > 0.2 (also in *Gaudin & Wible (2006)*); and (22) maximum wd of mandibular condyle ≥ 3x its ln (also in *Gaudin & Wible (2006)*).

Pampatheres plus glyptodonts plus *Proeutatus*: (1) nasal ln > 35% of GSL (also in *Gaudin & Wible (2006)*); (2) presence of osteodentine core in molariforms (also in *Gaudin & Wible (2006)*); (3) beveled wear on anterior molariforms, posterior molariforms worn flat (also in *Gaudin & Wible (2006)*); (4) lateral portion of frontal/parietal suture even with anterior edge of the glenoid; (5) optic foramen hidden in lateral view (also in *Gaudin & Wible (2006)*); (6) dorsal edge of zygomatic process of squamosal connected to nuchal crest posteriorly; (7) middle of infratemporal crest marked by large boss, the ossified ala hypochiasmata; (8) open groove for vidian nerve in roof of nasopharynx; (9) large entoglenoid process of squamosal; (10) groove for greater petrosal nerve uncovered by anteroventral process of tegmen tympani [=processus crista facialis]; (11) anteroventral process of tegmen tympani reduced in size, only contacts squamosal; (12) tensor tympanic muscle originates on anteroventral promontorium; (13) caudal tympanic process of petrosal lacks contact for ectotympanic; (13) rostral process of petrosal enlarged, promontorium elongated anteromedially; (14) presence of epitympanic recess [as opposed to a sinus]; and (15) triangular shelf of petrosal with raised posterolateral corner.

Note:
GSL, greatest skull length; dp, dorsoventral depth; ln, anteroposterior length; M1...9, upper molariform teeth; MML, maximum mandibular length; wd, transverse width.

(Table 5), further affirming its status as a disctinct pampathere species, currently known only from the late Blancan NALMA of central Florida.

The description has also revealed a large number of characteristics that appear to distinguish the genus *Holmesina* from other pampatheres. As noted in the Introduction

section of the present study, *Holmesina* is not recognized as a separate genus by *McKenna & Bell (1997)*, and other authors have suggested the genus may be invalid (*James, 1957*; *Robertson, 1976*). Our description identifies more than a dozen potential diagnostic cranial features (Table 5), strongly affirming the monophyly of this genus, which includes species from both North and South America.

Perhaps the largest suite of features with systematic value are identified as potential synapomorphies of pampatheres as a group (Table 5). The pampatheres have long been recognized as a distinctive group of cingulates, with their flat-topped, bilobate posterior molariforms that are highly dissimilar to the teeth of other cingulates. However, there is less agreement on how this morphological uniqueness should be treated taxonomically, with debate centered on whether pampatheres should be placed in a subfamily Pampatheriinae, a subgroup of the extinct and extant armadillo family Dasypodidae, as was typically the case in traditional classifications (*Hoffstetter, 1958*; *Patterson, Segall & Turnbull, 1989*), or considered a family in their own right, the Pampatheriidae, as they are listed in *McKenna & Bell (1997)* and in most recent papers (*De Iuliis & Edmund, 2002*; *Tambusso & Fariña, 2015*; *Góis et al., 2015*). It should be noted here that if pampatheres are placed in their own family, and if we accept their close relationship to glyptodonts (discussed below), both morphological (*Gaudin & Wible, 2006*; *Billet et al., 2011*) and molecular phylogenies (*Delsuc et al., 2016*; *Mitchell et al., 2016*) would imply that this clade evolved from within armadillos. This in turn would make the family Dasypodidae, a taxon still widely employed in the mammalogical literature (*Wilson & Reeder, 2005*; *Vaughan, Ryan & Czaplewski, 2015*), a paraphyletic group, necessitating the recognition of multiple armadillo families within "Dasypodidae." *Gibb et al. (2016)* propose dividing the Cingulata into two families, Dasypodidae and Chlamyphoridae, the latter including the glyptodonts as a subfamily. This arrangement reflects the basal split in Cingulata between these two clades, but it results in one of the long-recognized subfamilies of armadillos (and indeed the smallest subfamily in terms of generic level diversity), the Dasypodinae, being accorded family level status by *Gibb et al. (2016)*. The other three extant armadillo subfamilies (Euphractinae, Chlamyphorinae, and Tolypeutinae) are lumped together, along with the extinct glyptodonts (Glyptodontinae), and presumably their close relatives the pampatheres (now Pampatheriinae), into a single, very large, taxonomically, morphologically and ecologically diverse family. In our view, this is a minimalist approach to reordering family level diversity among cingulates that does not adequately reflect the age, morphological disparity, and taxonomic diversity encompassed by cingulates in general, and the Chlamyphoridae in particular. Moreover, it appears inconsistent with the manner in which taxonomic diversity is distributed in the sister taxon to Cingulata, the Pilosa. It is particularly noteworthy that the Vermilingua, the youngest and least diverse of the three main xenarthran clades (including only four living species; *McDonald, Vizcaíno & Bargo, 2008*; *Gaudin & Croft, 2015*), is currently divided into two families. The sloths, which are also a younger radiation than the cingulates (at least as far as they are documented in the fossil record; *Gaudin & Croft, 2015*) are currently arranged in five families (*Gaudin, 2004*). We would therefore advocate recognition of all 4 extant subfamilies of armadillos, as well as the pampatheres and very diverse glyptodonts,
respectively, as family level taxa, so that Cingulata would encompass at least seven families—Dasypodidae (following *Gibb et al., 2016*), Chlamyphoridae (following *Gibb et al., 2016*, but restricted to the members of the subfamily Chlamyphorinae, i.e., the living fairy armadillos), Euphractidae (for living euphractines plus their extinct kin), Tolypeutidae, Glyptodontidae, and Pampatheriidae. We would then group the latter six families into a larger clade Chlamyphoroidea, following the phylogeny of *Gibb et al. (2016)*—note this is also consistent with the morphological phylogenies of *Billet et al. (2011)* and *Gaudin & Wible (2006)*. We believe this is a better way to arrange cingulate diversity; however, we recognize that it does not account for all the taxonomic difficulties posed by the group. For example, it would leave some extinct taxa (e.g., eutatine armadillos, and perhaps some extinct "euphractines" like *Prozaedyus* or *Macroeuphractus*, and the peculiar horned armadillo *Peltephilus*; see *Gaudin & Wible, 2006*; *Billet et al., 2011*) with an unresolved family level status.

The second largest list of putative synapomorphies recognized in this study support the alliance of pampatheres with the other clade of cingulate herbivores, the much more diverse and specialized glyptodonts (Table 5). An alliance of these two groups of large bodied herbivores was most prominently suggested by Bryan Patterson (*Patterson & Pascual, 1972*; *Patterson, Segall & Turnbull, 1989*), and was confirmed by the subsequent cladistic phylogenetic studies of *Engelmann (1985)*, *Gaudin & Wible (2006)*, *Porpino, Fernicola & Bergqvist (2009)*, *Porpino, Fernicola & Bergqvist (2010)*, and *Billet et al. (2011)*. The present study adds to the already extensive list of derived resemblances among these forms (Table 5). The studies by *Gaudin & Wible (2006)* and *Billet et al. (2011)* also suggest that the Miocene armadillo *Proeutatus* (Santacrucian SALMA) is the sister taxon to pampatheres plus glyptodonts. This armadillo has been hypothesized to share the herbivorous diet characteristic of pampatheres and glyptodonts (*Vizcaíno, Fernicola & Bargo, 2012* and references therein). Table 5 confirms that this relationship is supported by cranial features not directly related to mastication, e.g., features from the ear region.

Lastly, it should be noted that the present study identified a number of cranial features which are shared by some, but not all pampathere genera (e.g., Mf4 is bilobate in *Pampatherium* and *Kraglievich* but not *Holmesina*; mf5 is reniform in *Pampatherium* and *Vassallia* but not *Holmesina*) and some features that appear to be apomorphies of pampatheres other than *Holmesina* (e.g., postorbital process of zygomatic arch on squamosal rather than the jugal, and loss of connection between zygomatic arch and nuchal crest in *Vassallia*). Clearly, and unsurprisingly, cranial data has much to contribute to our understanding of pampathere systematics. To our knowledge, no published phylogenetic analysis of pampatheres exists, but we felt that such an analysis was beyond the scope of the present study, especially given the fact that much of the critical material is available only in South American museums. Nevertheless, such a study clearly needs to be produced in the near future if we are to better understand the evolution of this distinctive group of large cingulate herbivores, and their place in the history of Cingulata as a whole. Moreover, given their geographic distribution on both sides of the Isthmus of Panama (*Scillato-Yané et al., 2005*), a better understanding of pampathere internal relationships might also yield insights into their role in the so-called

Great American Biotic Interchange (GABI), the extensive exchange of taxa between North and South America that plays such a central role in the evolution of the mammalian fauna of these two continents.

## CONCLUSION

The present study represents the first detailed, extensively illustrated, bone-by-bone description of pampathere cranial osteology, including reconstructions of sutural patterns and the position and content of the major cranial foramina. Due to the abundance of fossil material available for this late Pliocene—early Pleistocene species from Florida, we have been able to document extensive intraspecific variation in a variety of cranial features. We have also identified a series of new cranial characteristics which appear to be diagnostic for *Holmesina floridanus*. Though the systematics of pampatheres is controversial, our study affirms the monophyly of the genus *Holmesina*, and provides additional characters that support the monophyly of pampatheres as a whole. We advocate the recognition of pampatheres as a distinct family Pampatheriidae within the large clade Cingulata. We also advocate for the recognition of their sister taxon, the glyptodonts, as a family level grouping Glyptodontidae, and for similar family level recognition for the extant cingulate clades historically assigned subfamily status, i.e., the Dasypodidae, Chlamyphoridae, Euphractidae, and Tolypeutidae. Lastly, this analysis highlights the need for further studies of pampatheres in general and *Holmesina floridanus* in particular, including phylogenetic analyses of pampathere interrelationships, studies of *Holmesina floridanus* postcrania and carapaces, and further studies of *Holmesina floridanus* cranial anatomy using CT-scans.

## INSTITUTIONAL ABBREVIATIONS

| | |
|---|---|
| **CM** | Carnegie Museum of Natural History, Pittsburgh, PA, USA |
| **FMNH** | Field Museum, Chicago, IL, USA |
| **UF, FLMNH** | Florida Museum of Natural History, University of Florida, Gainesville, FL, USA |
| **UTCM** | University of Tennessee at Chattanooga Museum of Natural History, University of Tennessee at Chattanooga, Chattanooga, TN, USA |
| **YPM VPPU** | Princeton University Collection, Peabody Museum of Natural History, Yale University, New Haven, CT, USA. |

## OTHER ABBREVIATIONS

| | |
|---|---|
| **c.n.** | cranial nerve |
| **GSL** | greatest skull length |
| **Mf1–9** | upper molariform teeth |
| **mf1–9** | molariform teeth |
| **MML** | maximum mandibular length |
| **NALMA** | North American Land Mammal Age |
| **SALMA** | South American Land Mammal Age. |

## ACKNOWLEDGEMENTS

First and foremost, we thank Richard Hulbert and Jon Bloch of the Florida Museum of Natural History (University of Florida, Gainesville, FL), for all of their help in accessing the wonderful material of *Holmesina floridanus* on which this report is based. For access to other fossil and extant skeletal material used in this study, we thank B. Simpson, J. Flynn and K. Angielczyk of the Field Museum of Natural History (Chicago, IL), J. Wible of the Carnegie Museum of Natural History (Pittsburgh, PA), and W. Joyce and D. Brinkman of the Peabody Museum at Yale University (New Haven, CT). For the exceptional illustrations accompanying this paper, we thank the ever-talented Julia Morgan Scott, and we thank S. Chatzimanolis of the University of Tennessee at Chattanooga for his help in making the stereophotographs in Fig. 10. We thank our anonymous reviewers for their insightful reviews of this manuscript.

### Funding

This work was supported by the Department of Biology, Geology, and Environmental Science at the University of Tennessee at Chattanooga, and by the Bramblett Gift Fund. The funders had no role in study design, data collection and analysis, decision to publish, or preparation of the manuscript.

### Grant Disclosures

The following grant information was disclosed by the authors:
Department of Biology, Geology, and Environmental Science at the University of Tennessee at Chattanooga, and the Bramblett Gift Fund.

### Competing Interests

The authors declare that they have no competing interests.

### Author Contributions

- Timothy J. Gaudin conceived and designed the experiments, contributed reagents/materials/analysis tools, wrote the paper, prepared figures and/or tables, reviewed drafts of the paper.
- Lauren M. Lyon wrote the paper, reviewed drafts of the paper.

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
