# Peer review of "Cranial osteology of the pampathere Holmesina floridanus (Xenarthra: Cingulata; Blancan NALMA), including a description of an isolated petrosal bone"

_PeerJ, doi:10.7717/peerj.4022_

## Round 0.1 · original submission · Minor Revisions

· Academic Editor

Minor Revisions

I concur with all of the reviewers that this is an extremely valuable contribution that is clearly of great importance for understanding this enigmatic group of cingulates, and will no doubt prove to be a landmark study for students of xenarthran cranial anatomy. The manuscript is very well-written and highly detailed, and the figures are beautiful. All three of the reviewers have provided a number of useful comments as annotations on pdfs; please pay close attention to these suggestions as you revise your manuscript. Many of the comments are simple recommendations for additional citations or discussion, labelling of figures, or clarification of meaning, but a few are less trivial and would require more work to incorporate:

I agree with the suggestion of Reviewer 2 (Billet) that comparisons with a chlamyphorine/id and/or tolypeutine/id would be very valuable, given that these two higher taxa together appear to represent the sister clade of glyptodonts and pampatheres, but are not absolutely necessary. Inclusion of such comparisons would make an already excellent study that much better.

I agree with Reviewer 2 that photographs of the original specimens (even as supporting documents) would be very useful, as would an additional image of UF 121742 showing the “double” pterygoid process, and an image showing an oblique posteroventral view of the petrosal UF 248500 in which the cochlear fossula/crista interfenestralis/fenestra vestibuli is visible.

I agree with Reviewer 3 (Ruiz) that it would be useful to include comparisons to PVL 6245, the isolated petrosal from the Geste Formation, and that of Kuntinaru (MNHN-SAL 1024), to the extent possible. I understand that the approach was to focus only on those comparative specimens for which all views were available, but even a short separate paragraph would be valuable.

A few more minor issues:

Reviewer 2 was not supportive of the recommendation that various cingulate subfamilies be elevated to family rank. Your discussion could benefit from more explicit reference to the time trees and divergence dates published by Delsuc et al., Gibb et al., and Mitchell et al. in their various 2016 papers, much in the way that other authors in the recent mammalogical literature have argued that divergence times provide a useful guide for erection or elevation of higher taxa (e.g., Patterson and Upham, 2014, ZJLS, who erected the new family Heterocephalidae within Hystricognathi).

Reviewers 2 and 3 both expressed concern about how subadult status was determined; this issue is deserving of more detailed treatment.

A reference to the 2013 Ph.D. dissertation of Góis (“Análisis morfológico y afinidades de los Pampatheriidae (Mammalia, Xenarthra)”, Universidad Nacional de La Plata) would be appropriate, as noted by two of the reviewers.

There appear to be issues with the call-outs to figures 5 and 6, which are more likely to figures 3 and 4, on line 450, and then the reverse on line 461.

I look forward to seeing the revised manuscript! And thanks again for submitting such an excellent study to PeerJ.

·

Basic reporting

This is a clearly written and exquisitely illustrated manuscript. I believe that there is no need of additional materials besides the informations provided in the tables and figures. I have only minor suggestions and commentaries on certain parts of the manuscript that I have included in the attached annotated PDF file.

Experimental design

This is a well-organized and much needed in depth description of a representative of one of the most interesting groups of fossil cingulates. The results have clear implications for the understanding of the taxonomy and phylogenetic relationships of pampatheres (as well as for other cingulates), and might be useful in future cladistics analysis. The bone-by-bone descriptions are very lengthy at places, but overall there is no excessive details. In fact, I should say that similarly detailed up to date descriptions are lacking for several groups of cingulates. That being said, I suggest that some descriptions may be shortened by reducing or excluding morphological features that are common to all cingulates wherever possible.

Validity of the findings

The findings reported in the manuscript are sound and they can be easily checked by any interested reader with the help of the excellent illustrations.

Additional comments

I have no additional comments in addition to those stated above

·

Basic reporting

The paper is clear and very well-written; I've noted a few typos in the attached annotated PDF.
The literatures references are up to date, and well-cited throughout; I've just made a few suggestions in the annotated PDF, though these additions are not necessary.
The article is very well structured and the figures are generally excellent. As this constitutes an extensive description of cranial anatomy, I feel that some more features described in the text should be labelled on the figures (these are specified in the attached PDF).

Experimental design

no comment

Validity of the findings

no comment

Additional comments

Overall, I found this work (a thorough description of the external cranial anatomy of the pampathere Holmesina floridanus) to be very well executed and very valuable. It will certainly count as a major reference concerning cranial anatomy in fossil cingulates and will be much used as a comparative basis. The quality of the anatomical observations and of the figures is very high.
I have only minor comments for improvement:
1) additional labeling is needed on several figures
2) comparison of cranial anatomy with a chlamyphorine and/or tolypeutine armadillo is also desirable, given the fact that pampatheres, supposedly close to glyptodonts, may in turn be close to chlamyphorine and tolypeutine subfamilies (according to DNA studies);
3) for some features with high variation (eg, teeth outline), it might be interesting to discuss whether or not ontogeny (eg, tooth wear) may explain the observed differences
4) the discussion on taxonomic ranks attributed within Cingulata (family or subfamily) should be mitigated and existing phylogenetic arguments for having two basal families of cingulates should be given better consideration
Please see further details and other minor comments in the attached PDF

·

Basic reporting

Im not an american speaker so I can't check the grammar.
The scientific terms used are actualized and in agreement with the modern terminology of mammals anatomy.
The list of references is adequate, although suggested to quote a few more for specific topics (comments on the text).
The article structure, figs, and tables are adequate. The Figures are fantastic and well planed according to the paper.
The submission is Self-contained. Its a solid manuscript.

Experimental design

This research fills an important gap in the knowledge of xenarthran anatomy. Research questions are well defined and relevant. The investigation was conducted in a very high quality way.
The methods are adequate for this kind of research, although I suggested to explain some of them a little bit more (comments on the text).

Validity of the findings

This paper is an absolute novelty for xenarthrans anatomy.

The data is robust for this kind of anatomical paper.

The discussion and conclusions are well stated.

Additional comments

Was a placer to read this paper.
Most of my comments tried to make the anatomical part of this paper even more informative for a frequent user of this kind of papers.

I made most of the comments on the manuscript.

Most of them refers to little details that in such a long description is usual to find. I suggested to take a look of some non quoted references that I think will improve the quality of the paper (comments on the text)

I specially recommend to check with other published armadillos phylogenies (see comments on the text) if some of the characters consider here as synapomorphies, were discussed of described by other authors.

---

## Round 0.2 · accepted · Accept

· Academic Editor

Accept

Thank you very much for the effort you have put into the revised version, which appears to be in excellent shape. I think that your responses to reviewers' concerns are fair and well-justified. I look forward to seeing this important paper in print!